# Streaming Belief Propagation
# for Community Detection

**Yuchen Wu**
Stanford University
wuyc14@stanford.edu

**Jakab Tardos**
EPFL
jakab.tardos@epfl.ch

**MohammadHossein Bateni**
Google Research
bateni@google.com

**André Linhares**
Google Research
linhares@google.com

**Filipe Miguel Gonçalves de Almeida**
Google Research
filipea@google.com

**Andrea Montanari**
Stanford University
montanari@stanford.edu

**Ashkan Norouzi-Fard**
Google Research
ashkannorouzi@google.com

## Abstract

The community detection problem requires to cluster the nodes of a network into a small number of well-connected 'communities'. There has been substantial recent progress in characterizing the fundamental statistical limits of community detection under simple stochastic block models. However, in real-world applications, the network structure is typically dynamic, with nodes that join over time. In this setting, we would like a detection algorithm to perform only a limited number of updates at each node arrival. While standard voting approaches satisfy this constraint, it is unclear whether they exploit the network information optimally. We introduce a simple model for networks growing over time which we refer to as *streaming stochastic block model* (StSBM). Within this model, we prove that voting algorithms have fundamental limitations. We also develop a streaming belief-propagation (STREAMBP) approach, for which we prove optimality in certain regimes. We validate our theoretical findings on synthetic and real data.

## 1 Introduction

Given a single realization of a network $G = (V, E)$, the community detection problem requires to find a partition of its vertices into a small number of clusters or 'communities' [GN02, MO04, JTZ04, For10, PKVS12, JYL$^+$18].

Numerous methods have been developed for community detection on static networks [GN02, CNM04, GA05, RCC$^+$04, LF09, VLBB08, WZCX18]. However, the network structure evolves over time in most applications. For instance, in social networks new users can join the network; in online commerce new products can be listed; see [WGKM18, GLZ08, KvB$^+$14] for other examples. In such dynamic settings, it is desirable to have algorithms that perform only a limited number of operations each time a node joins or leaves, possibly revising the labels of nodes in a neighborhood of the new node. (These notions will be formalized below.) Several groups have developed algorithms of this type in the recent past [HS12, CSG16, ZWCY19, MKV$^+$20]. The present paper aims at characterizing the fundamental statistical limits of community detection in the dynamic setting, and proposing algorithms that achieve those limits.

35th Conference on Neural Information Processing Systems (NeurIPS 2021).

As usual, establishing fundamental statistical limits requires introducing a statistical model for the network $G = (V, E)$. In the case of static networks, precise characterizations have only been achieved recently, and almost uniquely for a simple network model, namely the stochastic block model (SBM) [HLL83, ABFX08, KN11, RCY+11, DKMZ11, Mas14, ABH15, AS15, MNS18]. In this paper we build on these recent advances and study a dynamic generalization of the SBM, which we refer to as *streaming* SBM (StSBM).

The SBM can be defined as follows. Each vertex $v$ is given a label $\tau(v)$ drawn independently from a fixed distribution over the set $[k] = \{1, 2, \ldots, k\}$. Edges are conditionally independent given vertex labels. Two vertices $u$, $v$ are connected by an edge with probability $W_{\tau(u), \tau(v)}$. The analyst is given a realization of the graph and required to estimate the labels $\tau$. We will assume in addition that the analyst has access to some noisy version of the vertex labels, denoted by $\tilde{\tau} = (\tilde{\tau}(v))_{v \in V}$. This is a mathematically convenient generalization: the special case in which no noisy observations $\tilde{\tau}$ are available can be captured by letting $\tilde{\tau}$ be independent of $\tau$. Further, such a generalization is useful to model cases in which covariate information is available at the nodes [MX16].

Informally, the *streaming* SBM (StSBM) is a version of SBM in which nodes are revealed one at a time in random order (see below for a formal definition). In order to model the notion that only a limited number of updates is performed each time a new node joins the network, we introduce a class of 'local streaming algorithms.' These encompass several algorithms in earlier literature, e.g, [ZGL03, CSG16]. Our definition is inspired and motivated by the more classical definition of local algorithms for static graphs. Local algorithms output estimates for one vertex based only on a small neighborhood around it, and thus scale well to large graphs; see [Suo13] for a survey. A substantial literature studies the behavior of local algorithms for sparse graphs [GT12, HLS14, GS14, Mon15, MX16, FM17].

Our results focus on the sparse regime in which the graph's average degree is bounded. This is the most challenging regime for the SBM, and it is also relevant for real-world applications where networks are usually sparse. We present the following contributions:

*Fundamental limitations of local streaming algorithms.* We prove that, in the absence of side information, in streaming symmetric SBM (introduced in Section 2), local streaming algorithms (introduced in Section 3) do not achieve any non-trivial reconstruction: their accuracy is asymptotically the same as random guessing; see Corollary 1. This holds despite the fact that there exist polynomial time non-local algorithms that achieve significantly better accuracy. From a practical viewpoint, this indicates that methods with a small 'locality radius' (the range over which the algorithm updates its estimates) are ineffective at aggregating information: they perform poorly unless strong local side information is available.

*Optimality of streaming belief propagation.* On the positive side, in Section 4 we define a streaming version of belief propagation (BP), a local streaming algorithm that we call STREAMBP, parameterized by a locality radius $R$. We prove that, for any non-vanishing amount of side information, STREAMBP achieves the same reconstruction accuracy as offline BP; see Theorem 2. The latter is in turn conjectured to be the optimal offline polynomial-time algorithm [DKMZ11, Abb17, HS17]. Under this conjecture, there is no loss of performance in restricting to local streaming algorithms as long as (1) local side information is available; (2) the locality radius is sufficiently large; and (3) information is aggregated optimally via STREAMBP.

Let us emphasize that we do not claim (nor do we expect) STREAMBP to outperform offline BP. We use offline BP as an 'oracle' benchmark (as it has the full graph information available, and is not constrained to act in a streaming fashion).

*Implementation and numerical experiments.* In Section 5 and Appendix A–C, we validate our results both on synthetic data, generated according to the StSBM, and on real datasets. Our empirical results are consistent with the theory; in particular, STREAMBP substantially outperforms simple voting methods. However, we observe that it can behave poorly with large locality radius $R$. In order to overcome this problem, we introduce a 'bounded distance' version of STREAMBP, called STREAMBP*, which appears to be more robust. (STREAMBP* can be shown to enjoy the same theoretical guarantees as STREAMBP.)

## 2 Streaming stochastic block model

In this section we present a formal definition of the proposed model. The streaming stochastic block model is a probability distribution $\text{StSBM}(n, k, p, W, \alpha)$ over triples $(\tau, \tilde{\tau}, \boldsymbol{G})$ where $\tau \in [k]^n$ is a vector of labels (here $[k] \triangleq \{1, \dots, k\}$), $\tilde{\tau} \in [k]^n$ are noisy observations of the labels $\tau$, and $\boldsymbol{G} = (G(0), G(1), \dots, G(n))$ is a sequence of undirected graphs. Here $G(t) = (V(t), E(t))$ is a graph over $|V(t)| = t$ vertices and, for each $0 \le t \le n - 1$, $V(t) \subseteq V(t+1)$ and $E(t) \subseteq E(t+1)$. We will assume, without loss of generality, that $V(n) = [n]$, and interpret $\tau(v)$ as the label associated to vertex $v \in [n]$. For each $0 \le t \le n - 1$, $G(t)$ is the subgraph induced in $G(t+1)$ by $V(t)$; equivalently, all edges in $E(t+1) \setminus E(t)$ are incident to the unique vertex in $V(t+1) \setminus V(t)$.

The distribution $\text{StSBM}(n, k, p, W, \alpha)$ is parameterized by a scalar $\alpha \in [0, \frac{k-1}{k}]$, a probability vector $p = (p_1, \dots, p_k) \in \Delta_k \triangleq \{x \in [0, 1]^k, \langle x, 1 \rangle = 1\}$, and a symmetric matrix $W \in [0, 1]^{k \times k}$. We draw the coordinates of $\tau$ independently with distribution $p$, and set $\tilde{\tau}(v) = \tau(v)$ with probability $1 - \alpha$, and $\tilde{\tau}(v) \sim \text{Unif}([k] \setminus \{\tau(v)\})$ otherwise, independently across vertices:

$$\mathbb{P}\big(\tau(v) = s\big) = p_s, \qquad \mathbb{P}\big(\tilde{\tau}(v) = s_1 \mid \tau(v) = s_0\big) = \begin{cases} 1 - \alpha & \text{if } s_1 = s_0, \\ \alpha/(k-1) & \text{if } s_1 \ne s_0. \end{cases}$$

We then construct $G(n)$ by generating conditionally independent edges, given $(\tau, \tilde{\tau})$, with

$$\mathbb{P}\big((u, v) \in E(n) \mid \tau, \tilde{\tau}\big) = W_{\tau(u), \tau(v)}. \tag{1}$$

Note that the labels $\tilde{\tau}$ provide noisy 'side information' about the true labels $\tau$. This information $\tilde{\tau}$ is conditionally independent of the graph $G(n)$ given $\tau$. Finally we generate the graph sequence $\boldsymbol{G}$ by choosing a uniformly random permutation of the vertices $(v(1), v(2), \dots, v(n))$ and setting $V(t) = \{v(1), \dots, v(t)\}$ and $G(t)$ to the graph induced by $V(t)$. If $v = v(t)$, then we define $t$ as the arrival order of vertex $v$. Note that, for each $t$, $G(t)$ is distributed according to a standard SBM with $t$ vertices: $G(t) \sim \text{SBM}(t, k, p, W)$.

An equivalent description is that (conditional on $\tau, \tilde{\tau}$) $\text{StSBM}(n, k, p, W, \alpha)$ defines a Markov chain over graphs. The new graph $G(t+1)$ is generated from $G(t)$ by drawing the vertex $v(t+1)$ uniformly at random from $[n] \setminus V(t)$, and then the edges $(u, v(t+1))$, $u \in V(t)$, independently with probabilities given by Equation (1).

We are interested in the behavior of large graphs with bounded average degree. In order to focus on this regime, we will consider $n \to \infty$ and $W = W^{(n)} \to 0$ with $W^{(n)} = W_0/n$ for a matrix $W_0 \in \mathbb{R}_{\ge 0}^{k \times k}$ independent of $n$.

A case of special interest is the streaming symmetric SBM, $\text{StSSBM}(n, k, a, b, \alpha)$, which corresponds to taking $p = (1/k, \dots, 1/k)$ and $W_0$ having diagonal elements $a$ and non-diagonal elements $b$. Finally, the case $\alpha = (k-1)/k$ corresponds to pure noise $\tilde{\tau}$: in this case we can drop $\tilde{\tau}$ from the observations and we will drop $\alpha$ from the distribution parameters.

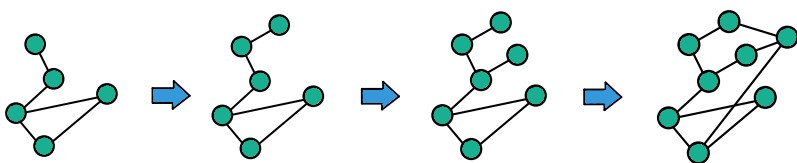

Figure 1: Evolving process of StSBM.

### 2.1 Definitions and notations

For two nodes $v, v' \in V(t)$, we denote by $d_t(v, v')$ their graph distance in $G(t)$, i.e., the length of the shortest path in $G(t)$ connecting $v$ and $v'$, with $d_t(v, v') = \infty$ if no such path exists. We also write $d(v, v') = d_n(v, v')$ for the graph distance in $G(n)$. For $v \in V(n)$ and $R \in \mathbb{N}^+$, let $\text{B}_R^t(v) = (V_R^t(v), E_R^t(v))$ denote the ball of radius $R$ in $G(t)$ centered at $v$, i.e., the subgraph induced in $G(t)$ by nodes $V_R^t(v) \triangleq \{v' \in V(t) : d_t(v, v') \le R\}$ and edges $(v_1, v_2) \in E(t)$ with

$v_1, v_2 \in V_R^t(v)$. Furthermore, let $D_R^t(v) \triangleq \{v' \in V(t) : d(v, v') = R\}$. Throughout the paper, unless otherwise stated, we assume $(v_1, v_2)$ is an undirected edge.

We consider an algorithm $\mathcal{A}$ that takes as input the graph $G(n)$ and side information $\tilde{\tau}$, and for each $v \in V(n)$ outputs $\mathcal{A}(v; G(n), \tilde{\tau}) \in [k]$ as an estimate for $\tau(v)$. Note that we always assume the arrival orders of vertices are observed (i.e., by observing $v \in [n]$ we also observe the unique $t \in [n]$ such that $v = v(t)$), thus $G(n)$ contains the arrival order of its vertices. We define the estimation accuracy of algorithm $\mathcal{A}$ as

$$Q_n(\mathcal{A}) \triangleq \mathbb{E}\left[\max_{\pi \in \mathfrak{S}_k} \frac{1}{n} \sum_{v \in V(n)} \mathbb{1}\left(\mathcal{A}(v; G(n), \tilde{\tau}) = \pi \circ \tau(v)\right)\right]. \tag{2}$$

Here $\mathfrak{S}_k$ is the group of permutations over $[k]$ and the expectation is with respect to $G(n), \tau, \tilde{\tau}$, and the randomness of the algorithm (if $\mathcal{A}$ is randomized).

## 3 Local streaming algorithms

In this section we introduce local streaming algorithms, which are a generalization of local algorithms in the dynamic network setting. An *R-local streaming algorithm* is an algorithm that at each vertex keeps some information available to that vertex. As a new vertex $v(t)$ joins, information within the $R$-neighborhood $\mathsf{B}_R^t(v(t))$ is pulled. An estimate for $\tau(v)$ is constructed based on information available to $v$. In order to accommodate randomized algorithms we assume that random variables $(\xi_v)_{v \in V(n)} \overset{iid}{\sim} \mathsf{Unif}([0, 1])$, independent of the graph, are part of the local information available to the algorithm.

As an example, we can consider a simple voting algorithm. At each step $t$, this algorithm keeps in memory the current estimates $\hat{\tau}(v) \in [k]$ for all $v \in V(t)$. As a new vertex $v(t)$ joins, its estimated label is determined according to

$$\hat{\tau}(v(t)) = \operatorname{argmax}_{s \in [k]} \pi_t(s) \qquad \pi_t(s) = \delta \mathbb{1}(s = \tilde{\tau}(v(t))) + \sum_{(v(t), u) \in E(t)} \mathbb{1}(s = \hat{\tau}(u)). \tag{3}$$

In words, the estimated label at $v(t)$ is the winner of a voting procedure, where the neighbors of $v(t)$ contribute one vote each, while the side information at $v(t)$ contributes $\delta$ votes.

For $v \in V(n)$ and $t \in [n]$, we denote the subgraph accessible to $v$ at time $t$ by $\mathcal{G}_v^t = (\mathcal{V}_v^t, \mathcal{E}_v^t)$, with initialization $\mathcal{G}_v^0 = (\{v\}, \emptyset)$. At time $t$, we conduct the following updates:

$$\mathcal{V}_v^t \triangleq \begin{cases} \bigcup_{v' \in V_R^t(v(t))} \mathcal{V}_{v'}^{t-1} & \text{for } v \in V_R^t(v(t)), \\ \mathcal{V}_v^{t-1} & \text{for } v \notin V_R^t(v(t)). \end{cases}$$

We let $\mathcal{G}_v^t$ be the subgraph induced in $G(t)$ by $\mathcal{V}_v^t$, and denote by $\bar{\mathcal{G}}_v^t = (\bar{\mathcal{V}}_v^t, \bar{\mathcal{E}}_v^t)$ the corresponding labeled graph with vertex labels $\tilde{\tau}$ and randomness $\xi$. Namely $\bar{\mathcal{V}}_v^t \triangleq \{(v', \tilde{\tau}(v'), \xi_{v'}) : v' \in \mathcal{V}_v^t\}$, $\bar{\mathcal{E}}_v^t \triangleq \mathcal{E}_v^t$. At time $t$, all nodes in the $R$ neighborhood of $v(t)$ share the same updated subgraph $\mathcal{G}_{v(t)}^t$ while the rest of the subgraphs stay unchanged. Let us emphasize that the 'neighborhoods' $\mathcal{G}_v^t$ are not symmetric, in the sense that we can have $v_1 \in \mathcal{G}_{v_2}^t$ but $v_2 \notin \mathcal{G}_{v_1}^t$.

**Definition 1** (*R*-local streaming algorithm). *An algorithm $\mathcal{A}$ is an R-local streaming algorithm if, at each time $t$ and for each vertex $v \in V(t)$, it outputs an estimate of $\tau(v)$ denoted by $\mathcal{A}^t(v; G(t), \tilde{\tau}) \in [k]$, which is a function uniquely of $\bar{\mathcal{G}}_v^t$.*

Note that this class includes as special cases voting algorithms (which correspond to $R = 1$) but also a broad class of other approaches. We will compare $R$-local streaming algorithms with $R$-local algorithms (non-streaming). In order to define the latter, given a neighborhood $\mathsf{B}_R^t(v)$, we define the corresponding labeled graph as $\bar{\mathsf{B}}_R^t(v) \triangleq (\bar{V}_R^t(v), E_R^t(v))$, with $\bar{V}_R^t(v) \triangleq \{(v', \tilde{\tau}(v'), \xi_{v'}) : v' \in V_R^t(v(t))\}$.

**Definition 2** (*R*-local algorithm). *An algorithm $\mathcal{A}$ is an R-local algorithm if, at each time $t$ and for each vertex $v \in V(t)$, it outputs an estimate of $\tau(v)$ denoted by $\mathcal{A}^t(v; G(t), \tilde{\tau}) \in [k]$, which is a function uniquely of $\bar{\mathsf{B}}_R^t(v)$.*

For simplicity, define the final output of an algorithm $\mathcal{A}$ by $\mathcal{A}(v; G(n), \tilde{\tau}) \triangleq \mathcal{A}^n(v; G(n), \tilde{\tau})$. The next theorem states that, under StSSBM, any local streaming algorithm with fixed radius behaves asymptotically as a local algorithm. Here we focus on StSSBM—extension to asymmetric cases is straightforward.

**Theorem 1.** *Let $G$ be distributed according to $\mathrm{StSSBM}(n, k, a, b, \alpha)$, and $v_0 \sim \mathrm{Unif}([n])$ be a vertex chosen independently of $G$. Then, for any $\epsilon > 0$, there exist $n_\epsilon, r_\epsilon \in \mathbb{N}^+$, such that for every $n \geq n_\epsilon$ with probability at least $1 - \epsilon$, the following properties hold: (1) $\mathcal{G}_{v_0}^n$ is a subgraph of $\mathsf{B}_{r_\epsilon}^n(v_0)$; and (2) $v_0$ does not belong to $\mathcal{G}_v^n$ for any $v \in V(n) \backslash V_{r_\epsilon}^n(v_0)$.*

Under the symmetric SBM, local algorithms without side information cannot achieve non-trivial estimation accuracy as defined in (2) [KMS16]. Therefore we have the following corollary of the first part of Theorem 1.

**Corollary 1.** *Under $\mathrm{StSSBM}(n, k, a, b)$ with no side information, no $R$-local streaming algorithm $\mathcal{A}$ can achieve non-trivial estimation accuracy. That is, $\lim_{n\to\infty} Q_n(\mathcal{A}) = 1/k$.*

**Remark 1.** *Corollary 1 does not hold if side information is available. As we will see below, an arbitrarily small amount of side information (any $\alpha < (k-1)/k$) can be boosted to ideal accuracy using $R$-local streaming algorithms with sufficiently large $R$. On the other hand, for a fixed small $R$, a small amount of side information has only limited impact on accuracy. Our numerical simulations illustrate this for voting algorithms, which are $R$-local for $R = 1$: they do not provide substantial boost over the use of only side information (i.e., the estimated label $\hat{\tau}(v) = \tilde{\tau}(v)$ at all vertices).*

**Remark 2.** *In practice we can imagine keeping a small memory containing global information and updating it each time a new vertex joins. This global information would be available at each node. For example, we could keep track of the estimated size of each community. For a suitable class of algorithms of this type, it can be shown that they cannot achieve non-trivial reconstruction in the symmetric model $\mathrm{StSSBM}(n, k, a, b)$ either. Due to space constraints we do not present this result here. Interested readers are referred to Section G in the appendix.*

## 4 Streaming belief propagation

In this section we focus on the symmetric model StSSBM. Notice that this model makes community detection more difficult compared to the asymmetric model, as in the latter case average degrees for vertices are different across communities, and one can obtain non-trivial estimation accuracy by simply using the degree of each vertex. For the symmetric model $\mathrm{StSSBM}(n, k, a, b, \alpha)$, we proved that local streaming algorithms cannot provide any non-trivial reconstruction of the true labels if no side information is provided, i.e., if $\alpha = (k-1)/k$. We also comment that, for a fixed (small) $R$, accuracy achieved by any algorithm is continuous in $\alpha$, and hence a small amount of side information will have a small effect.

In contrast, for any non-vanishing side information $\alpha < (k-1)/k$, we conjecture that information-theoretically optimal reconstruction is possible using a local streaming algorithm, under two conditions: $(i)$ the locality radius $R$ is large enough; and $(ii)$ the following Kesten-Stigum (KS) condition is met:

$$\lambda \triangleq \frac{(a-b)^2}{a + (k-1)b} > 1\,. \tag{4}$$

We will refer to $\lambda$ as to the 'signal-to-noise ratio' (SNR).

We provide evidence towards this conjecture by proposing a streaming belief propagation algorithm (STREAMBP) and showing that it achieves asymptotically the same accuracy as the standard offline BP algorithm. The latter is believed to achieve information-theoretically optimal reconstruction above the KS threshold [DKMZ11, MNS14]. We will describe STREAMBP in the setting of the symmetric model $\mathrm{StSSBM}(n, k, a, b, \alpha)$, but its generalization to the asymmetric case is immediate.

The algorithm has a state which is given by a vector of messages indexed by directed edges in $G(t)$, $\boldsymbol{m}^t = \{\boldsymbol{m}_{u\to v}^t, \boldsymbol{m}_{v\to u}^t : (u, v) \in E(t)\}$. Note that $G(t)$ is an undirected graph, and each edge $(u, v)$ corresponds to two messages indexed by $u \to v$ and $v \to u$. Each message is a probability distribution over $[k]$:

$$\boldsymbol{m}_{u\to v} = (m_{u\to v}(1), m_{u\to v}(2), \ldots, m_{u\to v}(k)) \in \Delta_k\,.$$

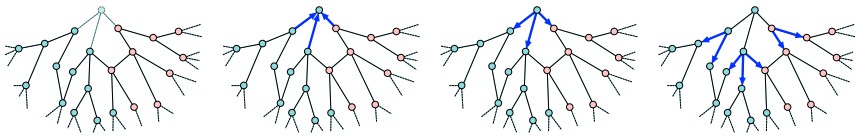

Figure 2: Update schedule of STREAMBP. Upon the arrival of a new vertex (shown in the leftmost figure), STREAMBP performs the belief propagation updates corresponding to the blue edges in the three other figures, in the order from left to right.

The BP update is a map BP $: (\Delta_k)^* \times [k] \to \Delta_k$, where $(\Delta_k)^*$ denotes the finite sequences of elements of $\Delta_k$:

$$\mathrm{BP}(\{\boldsymbol{m}_i\}_{i \leq \ell}; \tilde{\tau})(s) := \frac{\mathrm{BP}_0(\tilde{\tau})(s)}{Z} \prod_{i=1}^{\ell} \big(b + (a - b)m_i(s)\big) . \tag{5}$$

Here $\mathrm{BP}_0(\tilde{\tau})(s) \triangleq \big(\alpha + (k - 1 - k\alpha)\mathbb{1}_{\tilde{\tau}=s}\big)/(k - 1)$ and the constant $Z = Z(\{\boldsymbol{m}_i : i \leq \ell\}; \tilde{\tau})$ is defined implicitly by the normalization condition $\sum_{s \in [k]} \mathrm{BP}(\{\boldsymbol{m}_i : i \leq \ell\}; \tilde{\tau})(s) = 1$. When a message $v \to u$ is updated, we compute its new value by applying the function (5) to the incoming messages into vertex $v$, with the exception of $u \to v$ (non-backtracking property):

$$\boldsymbol{m}_{v \to u} \leftarrow \mathrm{BP}(\{\boldsymbol{m}_{w \to v} : w \in \partial v \setminus \{u\}\}; \tilde{\tau}(v)) . \tag{6}$$

Here $\partial v$ denotes the set of neighbors of vertex $v$ in the current graph. When a new vertex $v(t)$ joins at time $t$, we use the above rule to: (1) update all the messages incoming into $v(t)$, i.e., $w \to v(t)$, for $w$ a neighbor of $v(t)$ in $G(t)$, and (2) update all messages at distance $1 \leq \ell \leq R$ from $v(t)$ in $G(t)$, along paths outgoing from $v(t)$, in order of increasing distance $\ell$. The pseudocode for STREAMBP is given in Algorithm 1, and an illustration in Figure 2. For the sake of simplicity, we analyze this algorithm in the two-group symmetric model StSSBM$(n, 2, a, b, \alpha)$. We believe that the extension of this analysis to other cases is straightforward, but we leave it out of this presentation.

Our main result is that STREAMBP achieves asymptotically at least the same accuracy as offline BP, as originally proposed in [DKMZ11] and analyzed, e.g., in [MX16]. Offline BP performs the updates via Equation (6) in parallel on all the edges of $G(n)$ for $R - 1$ iterations, and then computes vertex estimates using $\boldsymbol{m}_u \leftarrow \mathrm{BP}(\{\boldsymbol{m}_{v \to u}\}_{v \in \partial u}; \tilde{\tau}(u))$, $\hat{\tau}(u) := \arg\max_{s \in [k]} m_u(s)$. Note that, for each $R$, this defines an $R$-local algorithm, and hence we will refer to $R$ as its radius.

**Theorem 2.** *For* $v \in V(n)$, *let* $\mathcal{A}_R(v; G(n), \tilde{\tau}) \in [k]$ *be the estimate of* $\tau(v)$ *given by Algorithm 1 (*STREAMBP*), and* $\mathcal{A}_R^{\mathrm{off}}(v; G(n), \tilde{\tau}) \in [k]$ *be the estimate given by offline BP with radius* $R$ *(equivalently, BP with parallel updates, stopped after* $R$ *iterations). Under the model* StSSBM$(n, 2, a, b, \alpha)$, STREAMBP *performs at least as well as offline BP:*

$$\liminf_{n \to \infty} \big(Q_n(\mathcal{A}_R) - Q_n(\mathcal{A}_R^{\mathrm{off}})\big) \geq 0.$$

**Algorithm 1** Streaming $R$-local belief propagation

1: **for** $t = 1, 2, \ldots, n$ **do**
2:     *// Update the incoming messages:*
3:     **for** $w \in D_1^t(v(t))$ **do**
4:         $\boldsymbol{m}_{w \to v(t)} \leftarrow \mathrm{BP}(\{\boldsymbol{m}_{u \to w} : u \in \partial w \setminus \{v(t)\}\}; \tilde{\tau}(w))$
5:     **end for**
6:     *// Update the outgoing messages:*
7:     **for** $r = 1, 2, \ldots, R$ **do**
8:         **for** $v \in D_r^t(v(t))$ **do**
9:             Let $v' \in D_1^t(v)$ on a shortest path connecting $v$ and $v(t)$.
10:             $\boldsymbol{m}_{v' \to v} \leftarrow \mathrm{BP}(\{\boldsymbol{m}_{u \to v'} : u \in \partial v' \setminus \{v\}\}; \tilde{\tau}(v'))$
11:         **end for**
12:     **end for**
13: **end for**
14: **for** $u \in V$ **do**
15:     $\boldsymbol{m}_u \leftarrow \mathrm{BP}(\{\boldsymbol{m}_{v \to u}\}_{v \in \partial u}; \tilde{\tau}(u))$
16:     Output $\hat{\tau}(u) := \arg\max_{s \in [k]} m_u(s)$ as an estimate for $\tau(u)$.
17: **end for**

It is conjectured that, in the presence of side information, i.e., for $\alpha < (k - 1)/k$, offline BP is optimal among all polynomial-time algorithms [DKMZ11] (provided $R$ can be taken arbitrarily

large). Whenever this is the case, the above theorem implies that STREAMBP is optimal as well. In the case of the symmetric model, it is also believed that under the KS condition (Equation 4), and for $\alpha < (k-1)/k$, offline BP does indeed achieve the information-theoretically optimal accuracy. This claim has been proven in certain cases by [MX16]: we can use the results of [MX16] in conjunction with Theorem 2 to obtain conditions under which STREAMBP is information-theoretically optimal.

**Corollary 2.** *Suppose one of the following conditions holds (for a sufficiently large absolute constant $C$) in the two-group symmetric model* StSSBM$(n, k = 2, a, b, \alpha)$: (1) $|a - b| < 2$ and $\alpha \in (0, 1/2)$; (2) $(a - b)^2 > C(a + b)$ and $\alpha \in (0, 1/2)$; or (3) $\alpha \in (0, 1/C)$. Then Algorithm 1 achieves optimal estimation accuracy:

$$\limsup_{R \to \infty} \limsup_{n \to \infty} \left( Q_n(\mathcal{A}_R) - \sup_{\mathcal{A}} Q_n(\mathcal{A}) \right) = 0.$$

*(Here the supremum is taken over all algorithms, not necessarily local or online.)*

## 5 Empirical evaluation

In this section, we compare the empirical performance of two versions of our streaming belief propagation algorithm (Algorithm 1) with some baselines. We consider the following streaming algorithms:

- STREAMBP: proposed algorithm in this work with radius $R$, outlined in Algorithm 1.
- STREAMBP$^*$: proposed algorithm in this work, which is a 'bounded-distance' version of STREAMBP. It is described in more detail in Section 5.1, and its pseudocode is given in Algorithm 2.
- VOTE1X, VOTE2X, VOTE3X: simple plurality voting algorithms that give a weight $\delta$ to the side information, as defined in Equation (3) (the numbering corresponds to $\delta \in \{1, 2, 3\}$). Despite looking somewhat naïve, voting algorithms are common in industrial applications.

We also compare the streaming algorithms above with ORACLEBP, which is the *offline* belief propagation algorithm, parameterized by its radius (number of parallel iterations) $R$. Note that in contrast to the streaming algorithms, to which we reveal one vertex at a time (along with its side information and the edges connecting it to previously revealed vertices), ORACLEBP has access to *the entire graph* and the side information *of all the vertices* from the beginning.

Our experiments are based both on synthetic datasets (Section 5.2) and on real-world datasets (Section 5.3). Because the model considered in this paper assumes *undirected* graphs, in a preprocessing step we convert the input graphs of the real-world datasets into undirected graphs by simply ignoring edge directions. Table 1 shows statistics of the datasets used (after making the graphs undirected); the values used for $a$ and $b$ for the real-world datasets are discussed in Section 5.3.

|           | $|V|$            | $|E|$               | $k$     | $a$        | $b$         |
|-----------|------------------|---------------------|---------|------------|-------------|
| Citeseer  | 3,264            | 4,536               | 6       | 11.47      | 0.89        |
| Cora      | 2,708            | 5,278               | 7       | 17.62      | 0.90        |
| Polblogs  | 1,490            | 16,715              | 2       | 40.69      | 4.23        |
| Synthetic | [10,000–50,000]  | [20,000–700,000]    | [2–5]   | [2.5–18]   | [0.05–1]    |

Table 1: Statistics of the synthetic and real-world datasets. For the synthetic datasets, various experiments with a range of parameters are performed.

As the measure of accuracy, we use the empirical fraction of labels correctly recovered by each algorithm, that is

$$\text{Acc} = \hat{\mathbb{E}} \left[ \max_{\pi \in \mathfrak{S}_k} \frac{1}{n} \sum_{v \in V(n)} \mathbb{1} \left( \mathcal{A}(v; G(n), \tilde{\tau}) = \pi \circ \tau(v) \right) \right].$$

In synthetic data we observe that, as soon as $\alpha < (k-1)/k$, the maximization over $\pi$ is not necessary (as expected from theory), and hence we drop it.

## 5.1 Bounded-distance streaming BP

In our experimental results presented below, we observed that the simple implementation in Algorithm 1 exhibits undesirable behaviors for certain graphs. We believe this is caused by two factors. First, unlike ORACLEBP, we do not have an upper bound on the radius of influence of each vertex in STREAMBP; it may indeed use long paths. Second, the number of cycles in the graph increases as $a, b$ grow. Large values of $a$ and $b$ can result in many paths being cycles, which negatively affects the performance of the algorithm.

In order to overcome these problems, we use two modifications. The first one is standard: we constrain messages so that $m_{u \to v}(s) \in [\varepsilon, 1 - \varepsilon]$ for some fixed small $\varepsilon > 0$ (we essentially constrain the log-likelihood ratios to be bounded).

The second modification defines a variant, presented in Algorithm 2, which we call STREAMBP*. Here the estimate at node $v$ is guaranteed to depend only on the graph structure and side information within $\mathsf{B}_R^n(v)$, and not on the information outside this ball. This constraint can be implemented in a message-passing fashion, by keeping, on each edge, $R + 1$ distinct messages $\boldsymbol{m}_{u \to v}^0, \ldots, \boldsymbol{m}_{u \to v}^R$, corresponding to different locality radii.

## 5.2 Synthetic datasets

Figure 3 illustrates the effect of the radius on the performance of the algorithms.[1] We use various settings for $k, a, b, \alpha$. We observe that voting algorithms do not perform significantly better than the baseline $1 - \alpha$ (dashed line). This is due both to the very small radius $R = 1$ of these algorithms, and to the specific choice of the update rule. For $R = 1$, STREAMBP and STREAMBP* perform significantly better than voting, showing that their update rule is preferable. Their accuracy improves with $R$, and is often close to the optimal accuracy (i.e. the accuracy of ORACLEBP for large $R$) already for $R \approx 5$.

In Figure 4, we study the effect of the SNR parameter $\lambda$, defined in Equation (4), on the performance of the BP algorithms. The accuracy of the algorithms improves as $\lambda$ increases. It is close to the baseline $1 - \alpha$ when the SNR is close to the KS threshold at $\lambda = 1$, and then it improves for large $\lambda$. This is a trace of the phase transition at the KS threshold which is blurred because of side information. For large values of $R$, the accuracy of our streaming algorithms STREAMBP and STREAMBP* nearly matches that of the optimal *offline* algorithm ORACLEBP.

Further experiments on synthetic datasets are reported in Appendix A. We then present in Appendix B the result of experiments where no side information is provided to the algorithms. We observe that in the streaming setting, and for small radius ($R$), neither STREAMBP nor STREAMBP* achieves high accuracy (above $1/k$) in the absence of side information, as suggested by our theoretical results.

---

**Algorithm 2** STREAMBP*: Bounded-distance streaming BP

---

1: **for** $t = 1, 2, \ldots, n$ **do**
2:    **for** $v \in D_1^t(v(t))$ **do**
3:       $\boldsymbol{m}_{v \to v(t)}^0 \leftarrow \mathbf{1}/k$
4:       **for** $i = 1, 2, \ldots, R$ **do**
5:          $\boldsymbol{m}_{v \to v(t)}^i \leftarrow$
           $\mathrm{BP}(\{\boldsymbol{m}_{v' \to v}^{i-1}\}_{v' \in \partial v \setminus \{v(t)\}}; \tilde{\tau}(v))$
6:       **end for**
7:    **end for**
8:    **for** $v \in D_1^t(v(t))$ **do**
9:       $\boldsymbol{m}_{v(t) \to v}^0 \leftarrow \mathbf{1}/k$
10:      **for** $i = 1, 2, \ldots, R$ **do**
11:         $\boldsymbol{m}_{v(t) \to v}^i \leftarrow$
          $\mathrm{BP}(\{\boldsymbol{m}_{v' \to v(t)}^{i-1}\}_{v' \in \partial v(t) \setminus \{v\}}; \tilde{\tau}(v(t)))$
12:      **end for**
13:    **end for**
14:    **for** $r = 2, 3, \ldots, R$ **do**
15:      **for** $v \in D_r^t(v(t))$ **do**
16:         Let $v' \in D_1^t(v)$ on a shortest path connecting $v$ and $v(t)$.
17:         **for** $i = 1, 2, \ldots, R$ **do**
18:            $\boldsymbol{m}_{v' \to v}^i \leftarrow$
          $\mathrm{BP}(\{\boldsymbol{m}_{u \to v'}^{i-1}\}_{u \in \partial v' \setminus \{v\}}; \tilde{\tau}(v'))$
19:         **end for**
20:      **end for**
21:    **end for**
22: **end for**
23: *// Compute the estimates:*
24: **for** $u \in V$ **do**
25:    $\boldsymbol{m}_u \leftarrow \mathrm{BP}(\{\boldsymbol{m}_{v \to u}^R\}_{v \in \partial u}; \tilde{\tau}(u))$
26:    Output $\hat{\tau}(u) := \arg\max_s m_u(s)$
27: **end for**

---

[1]Notice that the three voting algorithms do not depend on the radius, resulting in horizontal lines in the diagram.

## 5.3 Real-world datasets

We further investigate the performance of our algorithms on three real-world datasets: Cora [RA15], Citeseer [RA15], and Polblogs [AG05]. Cora and Citeseer are academic citation networks: vertices represent scientific publications partitioned into $k = 7$ and $k = 6$ classes respectively; directed edges represent citations of a publication by another. The Polblogs dataset represents the structure of the political blogosphere in the United States around its 2004 presidential election: vertices correspond to political blogs, and directed edges represent the existence of hyperlinks from one blog to another. The blogs are partitioned into $k = 2$ classes based on their political orientation (liberal or conservative).

As mentioned in the beginning of Section 5, in a preprocessing step we convert the input graphs of the real-world datasets into undirected graphs by simply ignoring edge directions. Also, since the graphs do not stem from the models of Section 2, we use oracle estimates of parameters $a$ and $b$, obtained by matching the density of intra- and inter-community edges with those in the model $\text{StSSBM}(n, k, \tilde{a}, \tilde{b}, \alpha)$. Namely, for a graph $G = (V, E)$, letting $V_1, \dots, V_k \subseteq V$ be the ground-truth communities, we set

$$\tilde{a} = |V| \cdot \frac{\sum_{i \in [k]} \sum_{(u,v) \in E} \mathbb{1}\{u, v \in V_i\}}{\sum_{i \in [k]} \binom{|V_i|}{2}}, \quad \tilde{b} = |V| \cdot \frac{\sum_{\{i,j\} \in \binom{[k]}{2}} \sum_{(u,v) \in E} \mathbb{1}\{u \in V_i, v \in V_j\}}{\sum_{\{i,j\} \in \binom{[k]}{2}} |V_i||V_j|}.$$

For each dataset, we run the streaming algorithms STREAMBP and STREAMBP* using the parameters $a = \tilde{a}$ and $b = \tilde{b}$. Notice that these estimates cannot be implemented in practice because we do not know the communities $V_j$ to start with. However, the performances appear not to be too sensitive to these estimates; we provide empirical evidence of this in Appendix A.

Figure 5 shows the accuracy of different algorithms on these datasets, for selected values of $\alpha$. Although the graphs in these datasets are not random graphs generated according to StSBM, the empirical results generally align with the theoretical results we proved for that model. Specifically, we see that our streaming algorithm STREAMBP* is approximately as accurate as the *offline* algorithm ORACLEBP, and significantly better than the voting algorithms. STREAMBP produces high-quality results for Cora and Citeseer datasets, but behaves erratically on the Polblogs dataset. As Polblogs is relatively dense compared with Cora and Citeseer thus contains more cycles, such phenomenon is likely due to the issues discussed in Section 5.1.

Appendix C provides additional details on these experiments, as well as results for other values of $\alpha$.

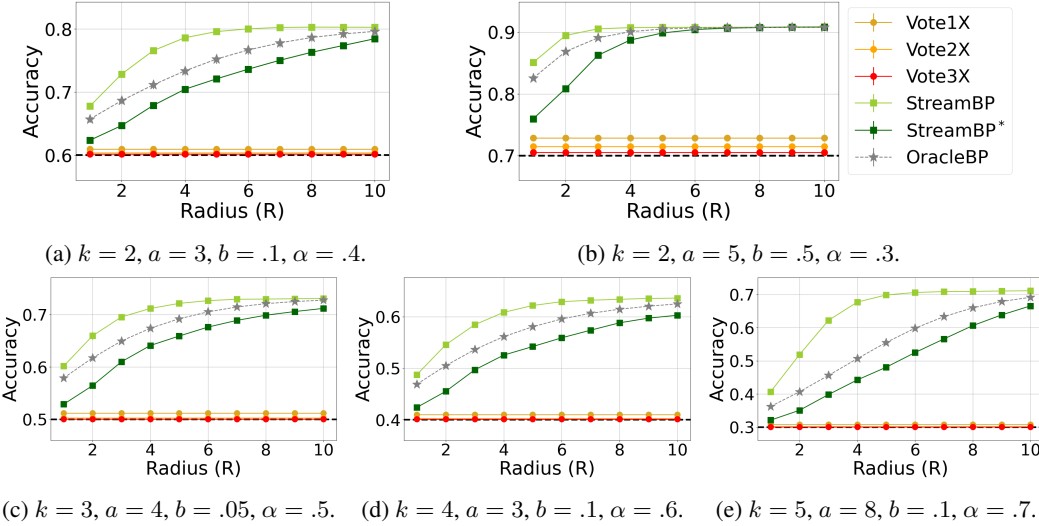

(a) $k = 2, a = 3, b = .1, \alpha = .4.$

(b) $k = 2, a = 5, b = .5, \alpha = .3.$

(c) $k = 3, a = 4, b = .05, \alpha = .5.$  (d) $k = 4, a = 3, b = .1, \alpha = .6.$  (e) $k = 5, a = 8, b = .1, \alpha = .7.$

Figure 3: Results of the experiments on synthetic datasets with 50,000 vertices. The black dashed line represents the accuracy of the noisy side information (without using the graph at all), namely $1 - \alpha$.

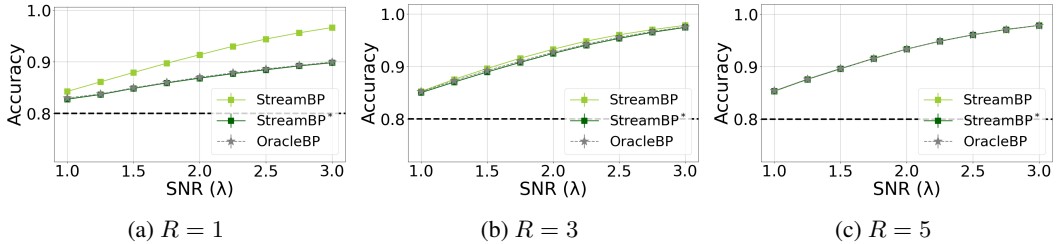

Figure 4: Results of the experiments on synthetic datasets with 50,000 vertices, with $k = 2, a + b = 8, \alpha = 0.2$, as we increase the ratio $\lambda$ from 1.0 to 3.0. The black dashed line represents the accuracy of the noisy side information (without using the graph at all), namely $1 - \alpha$.

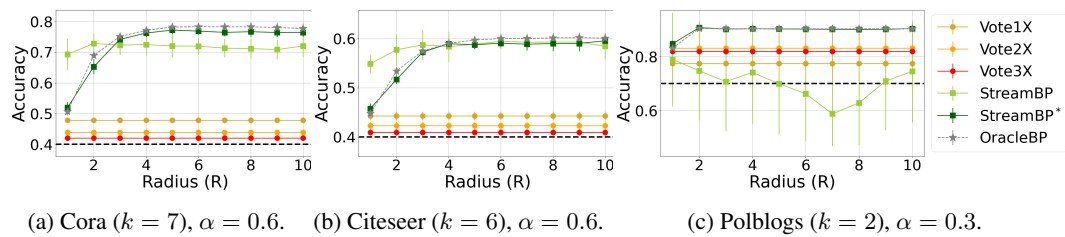

(a) Cora ($k = 7$), $\alpha = 0.6$.    (b) Citeseer ($k = 6$), $\alpha = 0.6$.    (c) Polblogs ($k = 2$), $\alpha = 0.3$.

Figure 5: Results of experiments on real-world datasets. The black dashed line represents the accuracy of the noisy side information (without using the graph at all), namely $1 - \alpha$.

## Acknowledgments and Disclosure of Funding

A.M. and Y.W. were partially supported by NSF grants CCF-2006489, IIS-1741162 and the ONR grant N00014-18-1-2729.

J.T. has received funding from the European Research Council (ERC) under the European Union's Horizon 2020 research and innovation program (grant agreement No 759471).

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
