# A    Further experiments with synthetic datasets

**Results with different sets of parameters.**    In Figures 6–9, we report more comprehensively the dependence of our algorithms' performance on the radius, where each figure corresponds to one choice of $k = 2, 3, 4, 5$. Similarly to Figure 3, we can see that as the radius increases, the performance of the three belief propagation algorithms (STREAMBP, STREAMBP*, and ORACLEBP) improves and converges to the same value. For some parameter settings, we observe that STREAMBP performs poorly when the radius is above a certain threshold; that is likely due to the issues discussed in Section 5.1.

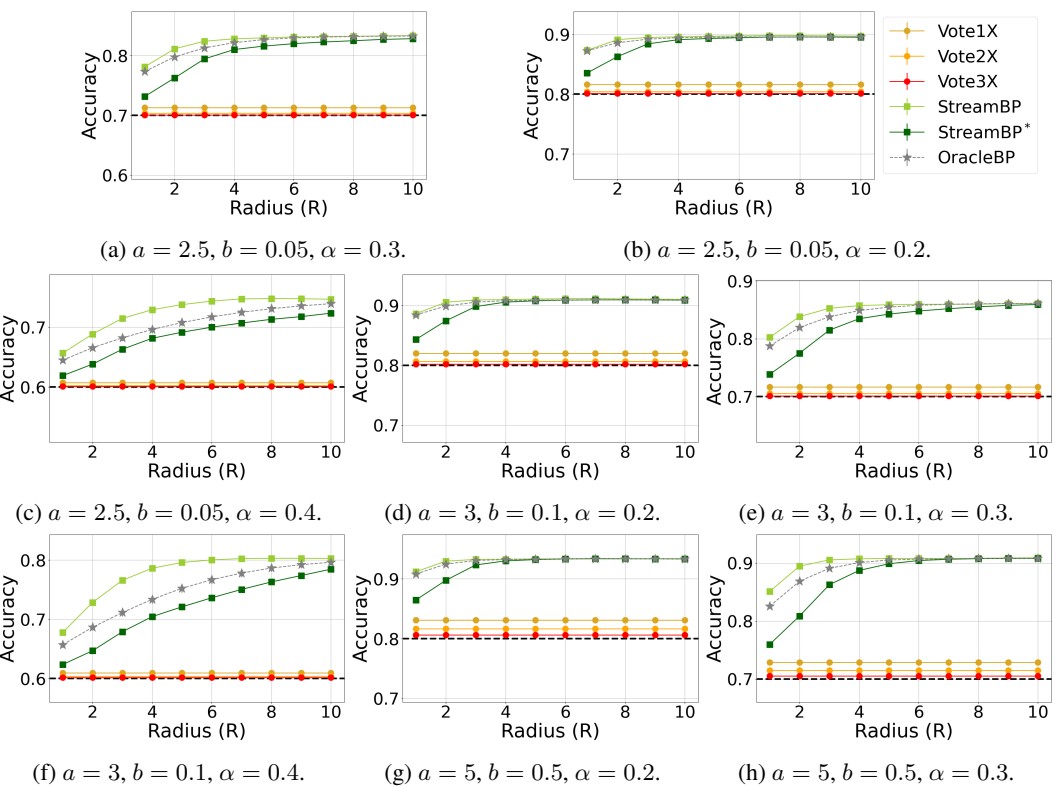

Figure 6: Results of the experiments on synthetic datasets with 50,000 vertices, and $k = 2$. The black dashed line represents the accuracy of the noisy side information (without using the graph at all), namely $1 - \alpha$.

**Variation of the accuracy of STREAMBP and STREAMBP* during their execution.**    We also investigate how the accuracy of the streaming algorithms STREAMBP and STREAMBP* varies during their execution on a given graph, as new vertices arrive one at a time. Figure 10 shows the results obtained for graphs generated according to the distribution $\mathrm{StSSBM}(n, a, b, k, \alpha)$, for various parameter settings.

We generally observe a very high accuracy (close to 1.0) for the first very few vertices, and then a sharp decline. This is not surprising given our use of the estimation accuracy $Q_n(\mathcal{A})$ defined in Section 2; in particular, the accuracy upon arrival of the first vertex is always equal to 1.0. After this initial sharp decline, the arrival of more vertices almost steadily improves the accuracy, which eventually stabilizes. Note that this improvement accelerates at some point; this is a trace of the phase transition at the KS threshold which is blurred because of side information. Also note that the truncation of a triple $(\tau, \tilde{\tau}, \boldsymbol{G}) \sim \mathrm{StSSBM}(n, k, a, b, \alpha)$ to the first $n' < n$ vertices induces the distribution $\mathrm{StSSBM}(n', k, \frac{n'}{n}a, \frac{n'}{n}b, \alpha)$, whose signal-to-noise ratio is $\lambda(n') \triangleq \frac{n'}{n}\lambda$; i.e., the signal-to-noise ratio increases linearly with $n'$. Each plot in Figure 10 has a vertical line showing the point where $\lambda(n')$ crosses the threshold 1.0.

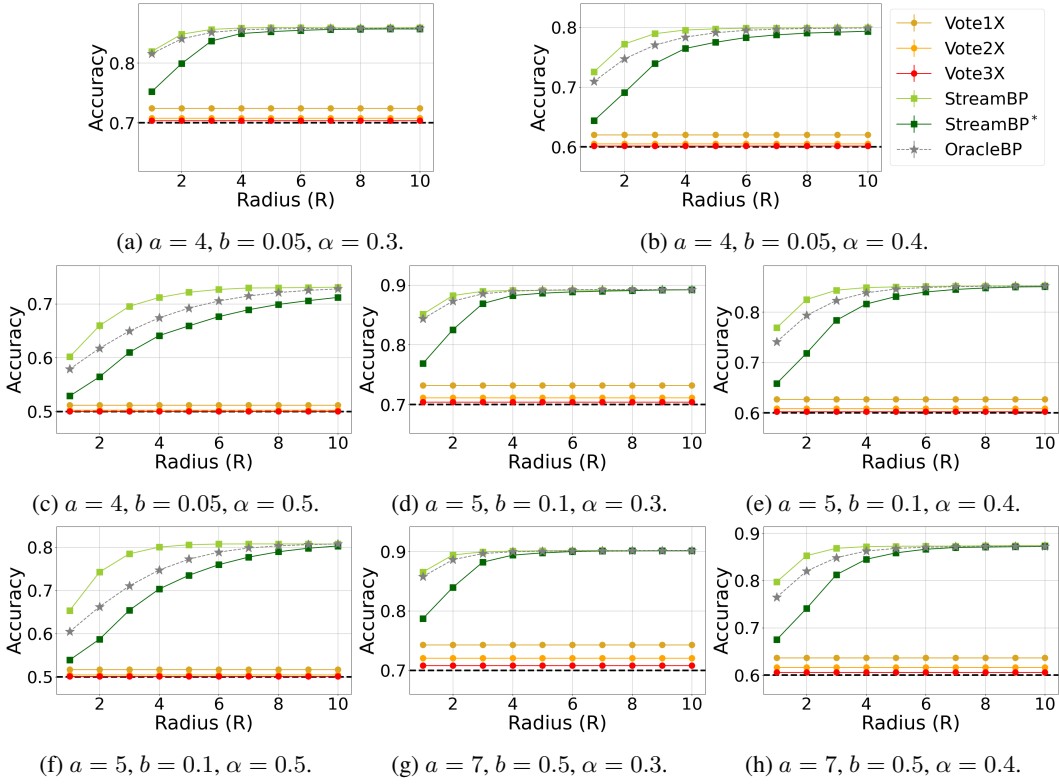

Figure 7: Results of the experiments on synthetic datasets with 50,000 vertices, and $k = 3$. The black dashed line represents the accuracy of the noisy side information (without using the graph at all), namely $1 - \alpha$.

**Robustness of STREAMBP and STREAMBP$^*$ w.r.t. the parameters $a$ and $b$.** Note that our proposed algorithms use $a$ and $b$ as input parameters. When applying these algorithms to real-world datasets that do not conform to the StSSBM model, we must approximate these parameters. Indeed in Section 5 we used empirical estimates of $a$ and $b$.

In this section we provide some evidence that our algorithms' behavior is robust to the choice of $a$ and $b$. In Figures 11 and 12 we present some results of experiments on synthetic data, where the algorithm and the model generating the input graph are given different $a$ or $b$ parameters. We observe that even with a relatively high discrepancy between the approximate and true parameters, STREAMBP$^*$ still achieves results comparable to the optimal setting of the parameters. Similar observations can be made about STREAMBP.

Various settings of $k$, $a$, $b$, and $\alpha$ are given as the caption to the individual plots. Here $a$ and $b$ signify the true parameters, according to which the input graph was generated. The performance of STREAMBP$^*$ is then plotted with various input parameters. For instance, the label "a+200%" in Figure 11 indicates that the algorithm receives a parameter $a$ that is three times greater than the true value. Similarly "b-67%" in Figure 12 indicates that the algorithm receives a parameter $b$ that is 67% less than the true value.

# B   Experiments without side information

In this section, we provide details of our experiments on synthetic data, when no side information is provided to the algorithms. Here $\alpha$ is set to $1 - 1/k$, that is, $\tilde{\tau}$ is completely independent from $\tau$. Note that in this setting we cannot hope to have large overlap between the output labels $\hat{\tau}$ and the true labels $\tau$. Therefore, we must evaluate the algorithms based on the best possible permutation of the output labels, as described in Section 2.1 in the definition of $Q_n(\mathcal{A})$:

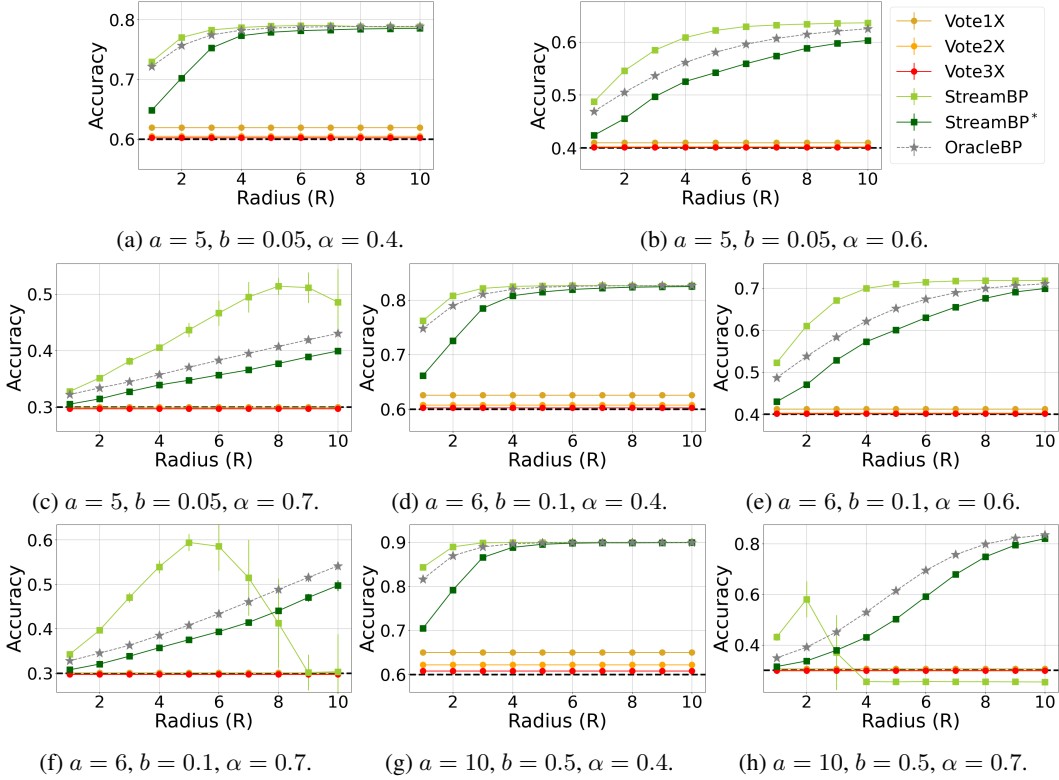

(a) $a = 5, b = 0.05, \alpha = 0.4$.  (b) $a = 5, b = 0.05, \alpha = 0.6$.

(c) $a = 5, b = 0.05, \alpha = 0.7$.  (d) $a = 6, b = 0.1, \alpha = 0.4$.  (e) $a = 6, b = 0.1, \alpha = 0.6$.

(f) $a = 6, b = 0.1, \alpha = 0.7$.  (g) $a = 10, b = 0.5, \alpha = 0.4$.  (h) $a = 10, b = 0.5, \alpha = 0.7$.

Figure 8: Results of the experiments on synthetic datasets with 50,000 vertices, and $k = 4$. The black dashed line represents the accuracy of the noisy side information (without using the graph at all), namely $1 - \alpha$.

$$\max_{\pi \in \mathfrak{S}_k} \frac{1}{n} \sum_{v \in V(n)} \mathbb{1}\left(\hat{\tau}(v) = \pi \circ \tau(v)\right).$$

Our observations confirm the theoretical result from Corollary 1. As predicted, when $r$ is small, no algorithm—neither the baselines nor our algorithms—can achieve any meaningful improvement over the trivial $1/k$ accuracy. We further observe that for sufficiently large $r$, both the offline algorithm ORACLEBP and our algorithm STREAMBP* perform significantly better than $1/k$. However, this occurs when $r$ is comparable to the diameter of the graph, when an online algorithm is no longer efficient. STREAMBP is unable to get any non-trivial result due to issues described in Section 5.1 which become especially problematic for large values of $a$, $b$, and $r$.

We present the results of our experiments for various settings of $k$, $a$, and $b$ in Figure 13. Note that the settings always satisfy the Kesten-Stigum condition from Equation (4). Without side information, this is necessary to see any non-trivial behavior, even for large radii.

## C  Additional results on real-world datasets

Figure 14 summarizes the results obtained by running the different algorithms on the datasets Cora, Citeseer, and Polblogs. For each dataset and choice of the radius (used by the algorithms STREAMBP, STREAMBP*, and ORACLEBP) and of the parameter $\alpha$, we run each algorithm 9 times; each run independently chooses an arrival order of the vertices (uniformly at random among all permutations of the vertices) and the side information (as described in Section 2). We note that the accuracy of our streaming algorithm STREAMBP* is comparable to that of the *offline* algorithm ORACLEBP, and significantly superior to the accuracy of the voting algorithms. STREAMBP produces high-quality results for the datasets Cora and Citeseer, but behaves erratically in the Polblogs dataset, generally worse than the voting algorithms; that is likely due to the issues discussed in Section 5.1.

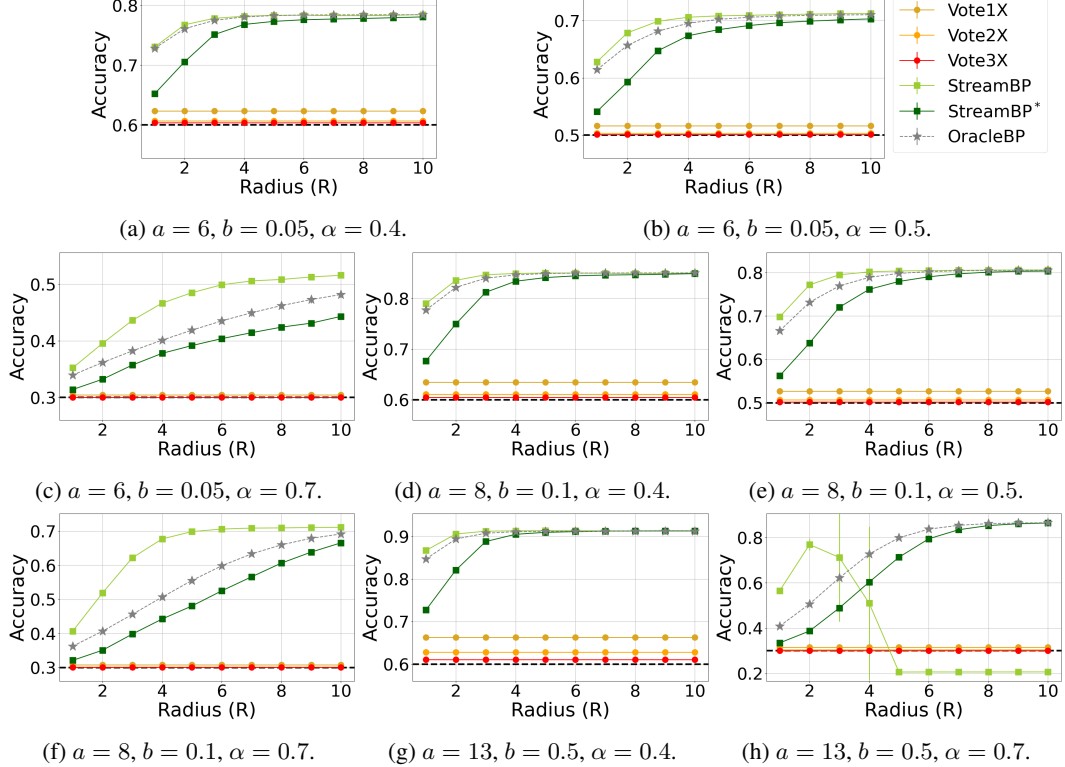

(a) $a = 6$, $b = 0.05$, $\alpha = 0.4$.      (b) $a = 6$, $b = 0.05$, $\alpha = 0.5$.

(c) $a = 6$, $b = 0.05$, $\alpha = 0.7$.    (d) $a = 8$, $b = 0.1$, $\alpha = 0.4$.    (e) $a = 8$, $b = 0.1$, $\alpha = 0.5$.

(f) $a = 8$, $b = 0.1$, $\alpha = 0.7$.    (g) $a = 13$, $b = 0.5$, $\alpha = 0.4$.    (h) $a = 13$, $b = 0.5$, $\alpha = 0.7$.

Figure 9: Results of the experiments on synthetic datasets with 50,000 vertices, and $k = 5$. The black dashed line represents the accuracy of the noisy side information (without using the graph at all), namely $1 - \alpha$.

## D    Definitions and technical lemmas

For completeness, we reproduce a standard lemma establishing that sparse graphs from SBM are locally tree-like.

**Lemma 1** ([MNS15]). *Let* $(X, G) \sim \mathrm{SSBM}(n, 2, a/n, b/n)$ *and* $R = R(n) = \lfloor \frac{1}{10} \log(n)/\log(2(a+b)) \rfloor$. *Let* $B_R := \{v \in [n] : d_G(1, v) \leq R\}$ *be the set of vertices at graph distance at most* $R$ *from vertex 1,* $G_R$ *be the restriction of* $G$ *on* $B_R$, *and let* $X_R = \{X_u : u \in B_R\}$. *Let* $T_R$ *be a Galton-Watson tree with offspring Poisson$(a+b)/2$ and* $R$ *generations, and let* $\tilde{X}^{(t)}$ *be the labelling on the vertices at generation* $t$ *obtained by broadcasting the bit* $\tilde{X}^{(0)} := X_1$ *from the root with flip probability* $b/(a+b)$. *Let* $\tilde{X}_R = \{\tilde{X}_u^{(t)} : t \leq R\}$. *Then, there exists a coupling between* $(G_R, X_R)$ *and* $(T_R, \tilde{X}_R)$ *such that*

$$\lim_{n \to \infty} \mathbb{P}\left( (G_R, X_R) = (T_R, \tilde{X}_R) \right) = 1.$$

For $A \subseteq V$, let $\tau(A)$ be the vector containing all true labels of vertices in $A$, and $\tilde{\tau}(A)$ be the vector containing all noisy labels of vertices in $A$.

**Definition 3** (Labeled branching tree). *For* $r \in \mathbb{N}^+$, $d > 0$, $p \in (0, 1)$, *let* $\mathbb{P}_{r,d,p}^T$ *denote the law of a labeled Galton Watson branching tree: this tree has* $r$ *generations, with the offspring distribution being Poisson with expectation* $d$. *Each vertex in the tree is associated with a label taking values in* $\{1, 2\}$. *The label at the root node is uniformly distributed over* $\{1, 2\}$, *and labels on the rest of the vertices are obtained by broadcasting the label from the root node with flip probability* $p$.

Furthermore, we denote the optimal estimation accuracy by

$$Q_n^* \triangleq \sup_{\mathcal{A}} \mathbb{E}\left[ \max_{\pi \in \mathfrak{S}_k} \frac{1}{n} \sum_{v \in V(n)} \mathbb{1}\left( \mathcal{A}(v; G(n), \tilde{\tau}) = \pi \circ \tau(v) \right) \right]. \tag{7}$$

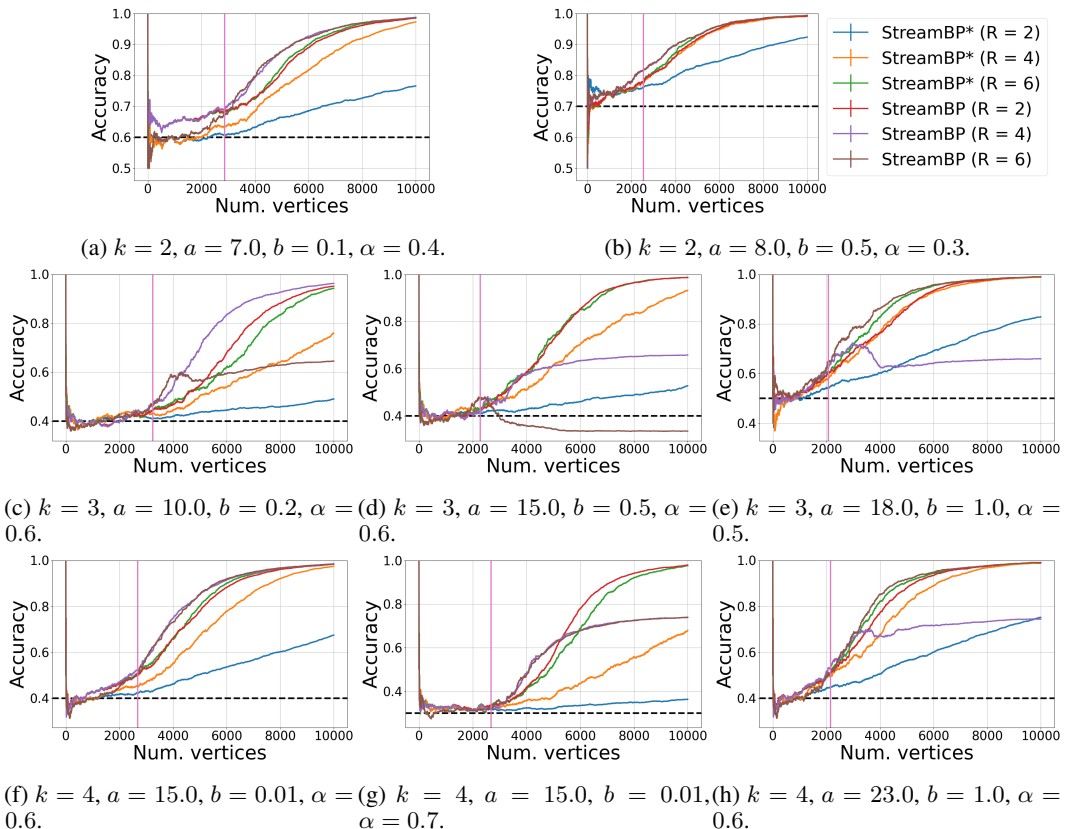

(a) $k = 2$, $a = 7.0$, $b = 0.1$, $\alpha = 0.4$.

(b) $k = 2$, $a = 8.0$, $b = 0.5$, $\alpha = 0.3$.

(c) $k = 3$, $a = 10.0$, $b = 0.2$, $\alpha = 0.6$.

(d) $k = 3$, $a = 15.0$, $b = 0.5$, $\alpha = 0.6$.

(e) $k = 3$, $a = 18.0$, $b = 1.0$, $\alpha = 0.5$.

(f) $k = 4$, $a = 15.0$, $b = 0.01$, $\alpha = 0.6$.

(g) $k = 4$, $a = 15.0$, $b = 0.01$, $\alpha = 0.7$.

(h) $k = 4$, $a = 23.0$, $b = 1.0$, $\alpha = 0.6$.

Figure 10: Accuracy per iteration obtained by running STREAMBP and STREAMBP* on a graph sampled from $\mathrm{StSSBM}(n, k, a, b, \alpha)$ with $n = 10,000$. The black dashed line represents the accuracy of the noisy side information (without using the graph at all), namely $1 - \alpha$. The pink vertical line indicates the point where the signal-to-noise ratio crosses the threshold $1.0$.

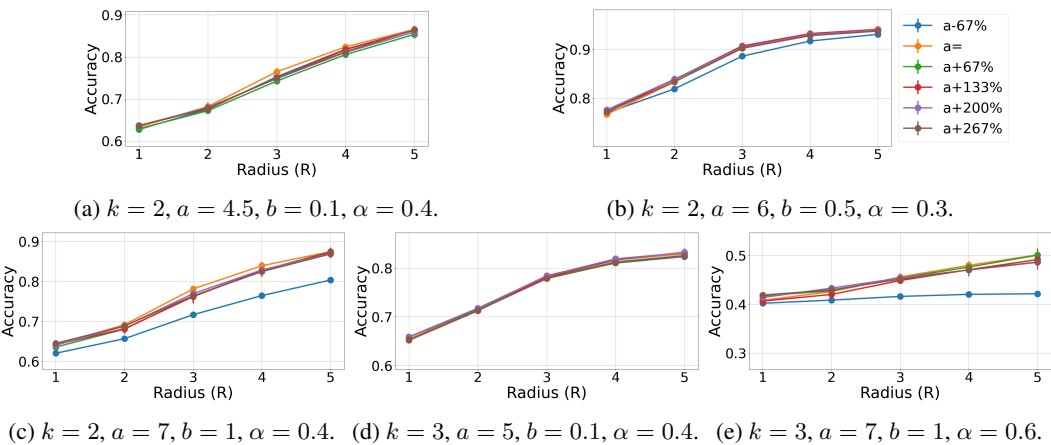

(a) $k = 2$, $a = 4.5$, $b = 0.1$, $\alpha = 0.4$.

(b) $k = 2$, $a = 6$, $b = 0.5$, $\alpha = 0.3$.

(c) $k = 2$, $a = 7$, $b = 1$, $\alpha = 0.4$.

(d) $k = 3$, $a = 5$, $b = 0.1$, $\alpha = 0.4$.

(e) $k = 3$, $a = 7$, $b = 1$, $\alpha = 0.6$.

Figure 11: Robustness of STREAMBP* w.r.t. to the choice of $a$.

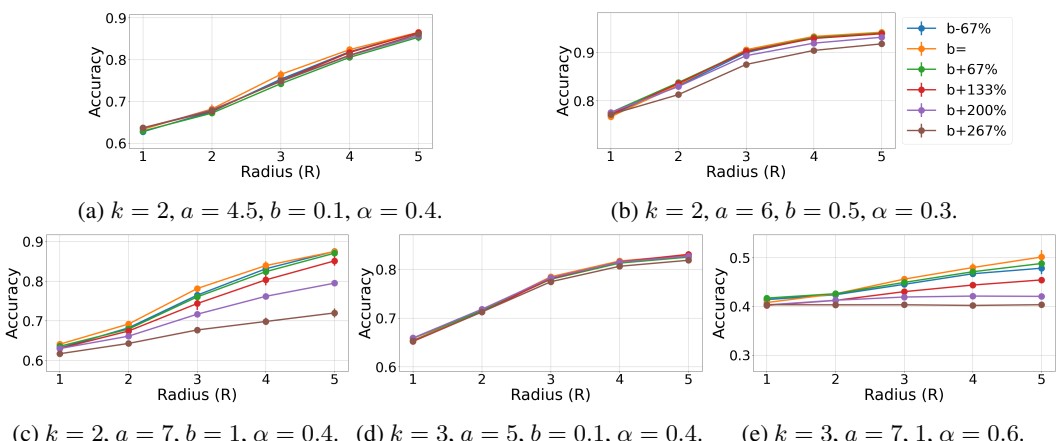

(a) $k = 2$, $a = 4.5$, $b = 0.1$, $\alpha = 0.4$.  (b) $k = 2$, $a = 6$, $b = 0.5$, $\alpha = 0.3$.

(c) $k = 2$, $a = 7$, $b = 1$, $\alpha = 0.4$.  (d) $k = 3$, $a = 5$, $b = 0.1$, $\alpha = 0.4$.  (e) $k = 3$, $a = 7$, $1$, $\alpha = 0.6$.

Figure 12: Robustness of STREAMBP* w.r.t. to the choice of $b$.

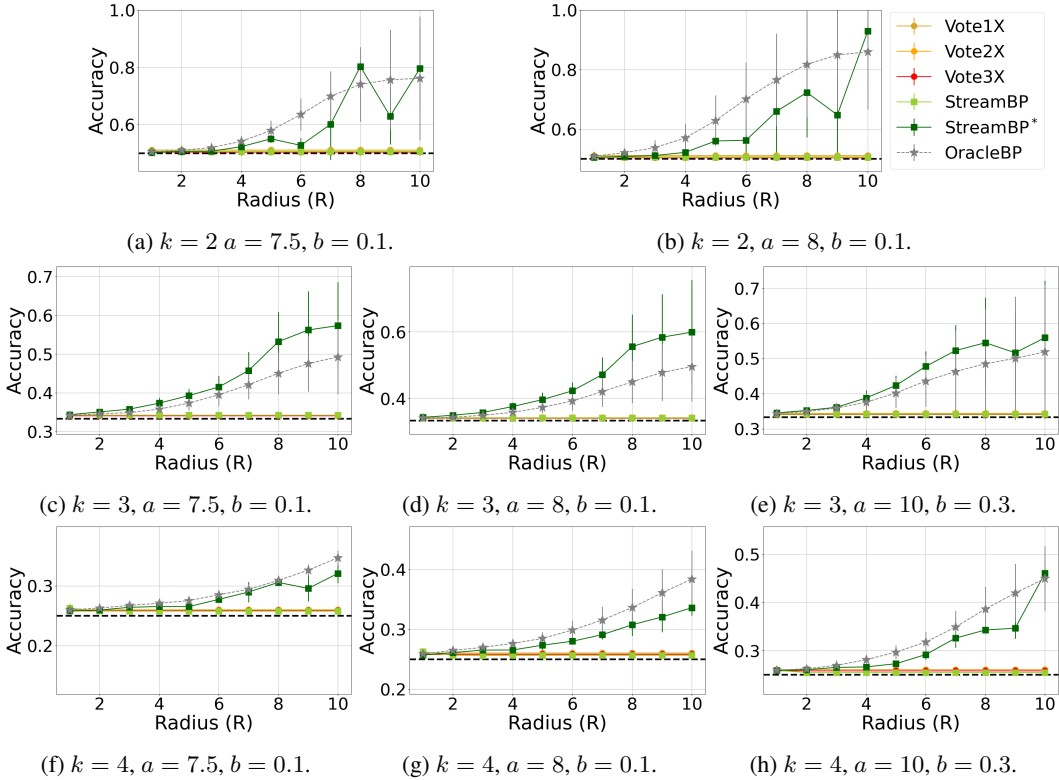

(a) $k = 2$ $a = 7.5$, $b = 0.1$.  (b) $k = 2$, $a = 8$, $b = 0.1$.

(c) $k = 3$, $a = 7.5$, $b = 0.1$.  (d) $k = 3$, $a = 8$, $b = 0.1$.  (e) $k = 3$, $a = 10$, $b = 0.3$.

(f) $k = 4$, $a = 7.5$, $b = 0.1$.  (g) $k = 4$, $a = 8$, $b = 0.1$.  (h) $k = 4$, $a = 10$, $b = 0.3$.

Figure 13: Results of the experiments on synthetic datasets with 10,000 vertices, and no side information (i.e. $\alpha = 1 - 1/k$).

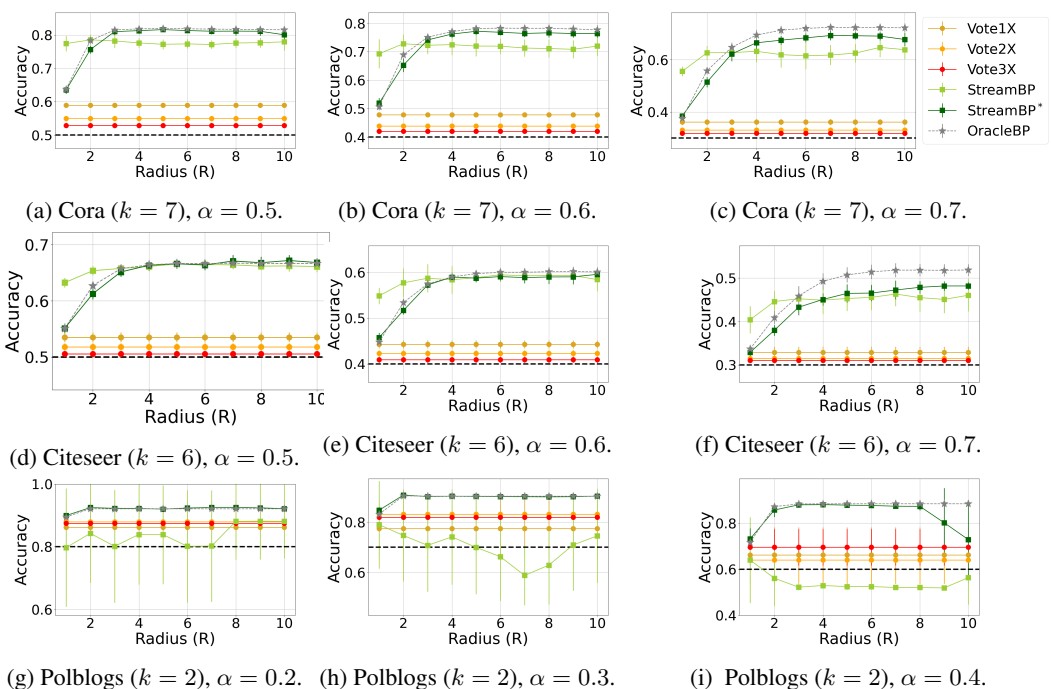

Figure 14: Results of the experiments on real-world datasets. The black dashed line represents the accuracy of the noisy side information (without using the graph at all), namely $1 - \alpha$.

## E   Analysis of local streaming algorithms

### E.1   Proof of Theorem 1

With a slight abuse of notations, in this part of the proof, when we refer to $V_r^t(v), E_r^t(v), \mathsf{B}_r^t(v)$ and $\mathcal{G}_r^t(v)$, we do not assume we know the revealing orderings of vertices within. For the sake of simplicity, we consider $k = 2$, cases with $k > 2$ can be proven similarly. Let $d = (a + b)/2$ be the average degree, $r \in \mathbb{N}^+$ satisfying

$$r \geq 2^{2R+3} R \left( 1 + (1 - e^{-d})^{-1} \sum_{i=0}^{R} d^i \right) d^{2R} e.$$

Let $(\lambda_{r+R}, T_{r+R}) \sim \mathbb{P}_{r+R,d,\frac{b}{a+b}}^T$ as in Definition 3, with $T_{r+R}$ being the graph and $\lambda_{r+R}$ being the set of labels. By Lemma 1, for any $\epsilon > 0$, there exists $n_\epsilon \in \mathbb{N}^+$ which is a function of $r, R$ and $\epsilon$, such that for $n \geq n_\epsilon$, there exists a coupling of $(\lambda_{r+R}, T_{r+R})$ and $(\tau(V_{r+R}^n(v_0)), \mathsf{B}_{r+R}^n(v_0))$ preserving $u_0$ paired up with $v_0$, and satisfies

$$\mathbb{P}\left( (\lambda_{r+R}, T_{r+R}) \neq (\tau(V_{r+R}^n(v_0)), \mathsf{B}_{r+R}^n(v_0)) \right) \leq \epsilon/2.$$

If $\mathcal{G}_{v_0}^n$ is not a subgraph of $\mathsf{B}_r^n(v_0)$, then there must exist $v_b \in D_r^n(v_0)$ such that $v_b$ belongs to $\mathcal{G}_{v_0}^n$. This is equivalently saying that there exists an "information flow" starting at $v_b$, proceeds as vertices are gradually revealed, and finally could reach $v_0$ by the end.

**Definition 4** (Information flow). *Given the graph $G(n) = (V(n), E(n))$, an information flow with origin at $v_b$ and end at $v_0$ is defined as a sequence of vertices $p_1, p_2, \cdots, p_l \in V(n)$, such that*

1. *$t^*(p_i) = t_i$, and $t_i < t_{i+1}$ for all $i \in [l-1]$.*

2. *$V_R^n(p_i) \cap V_R^n(p_{i+1}) \neq \emptyset$. Furthermore, $\min\{d(v_0, v) : v \in V_R^n(p_{i+1})\} < \min\{d(v_0, v) : v \in V_R^n(p_i)\}$, for all $i \in [l-1]$.*

3. *$v_b \in V_R^n(p_1), v_0 \in V_R^n(p_l), v_0 \notin V_R^n(p_i)$ for all $i \in [l-1]$.*

Notice that on the event $\mathsf{B}^n_{r+R}(v_0)$ is a tree, a necessary condition for $v_b \in \mathcal{G}^n_{v_0}$ is that there exists an information flow $u_1, u_2, \cdots, u_l$ with origin at $v_b$ and end at $v_0$. For $i \in [l]$, among all vertices on the shortest path connecting $v_0$ and $v_b$, let $v_i$ be a vertex with the smallest graph distance to $u_i$. A moment of thought reveals that we can find an eligible information flow such that $v_1, \cdots, v_l$ are distinct.

Denote the set of vertices on the path connecting $v_0$ and $v_b$ by $V_p$. Given $l$ and the graph, the number of (unordered) vertex combinations $\{v_1, v_2, \cdots, v_l\}$ is upper bounded by $\binom{r}{l}$. For each $x \in \{v_1, v_2, \cdots, v_l\}$, we define the following set:

$$V_p^x := \{v \in V_R^n(x) : \nexists v' \in V_p, v' \neq x, d(v, v') < d(v, x)\}.$$

Given $\{v_1, v_2, \cdots, v_l\}$, the total number of possible unordered vertex combinations $\{u_1, u_2, \cdots, u_l\}$ is upper bounded by $\prod_{i=1}^l |V_p^{v_i}|$. Furthermore, given unordered set $\{u_1, u_2, \cdots, u_l\}$, by definition one can specify their relative ordering $u_1, u_2, \cdots, u_l$ by sorting the following distances: $\min\{d(v_0, v) : v \in V_R^n(u_i)\}$. Finally, for such combinations to exist, one must have $\lfloor r/2R \rfloor \leq l \leq r$.

As a result, given $\mathsf{B}^n_{r+R}(v_0)$, if it is a tree, then the conditional probability of $\mathcal{G}^n_{v_0}$ not being a subgraph of $\mathsf{B}^n_r(v_0)$ is upper bounded by

$$\sum_{v_b \in V(n)} \mathbb{1}\{v_b \in D^n_r(v_0)\} \sum_{l=\lfloor r/2R \rfloor}^r \sum_{\{v_1, v_2, \cdots, v_l\}} \prod_{i=1}^l |V_p^{v_i}| \times \frac{1}{l!}.$$

Let $\mathbb{P}_{bt}(\cdot)$ denote the probability distribution over $(\lambda_{r+R}, T_{r+R})$, and let $\mathbb{E}_{bt}(\cdot)$ be the corresponding notation for taking expectation under that distribution. Then we have

$\mathbb{P}(\mathcal{G}^n_{v_0}$ is not a subgraph of $\mathsf{B}^n_r(v_0))$

$\leq \mathbb{P}((\lambda_{r+R}, T_{r+R}) \neq (\tau(V^n_{r+R}(v_0)), \mathsf{B}^n_{r+R}(v_0))) +$

$\quad \mathbb{P}\left(\mathcal{G}^n_{v_0}$ is not a subgraph of $\mathsf{B}^n_r(v_0), (\lambda_{r+R}, T_{r+R}) = (\tau(V^n_{r+R}(v_0)), \mathsf{B}^n_{r+R}(v_0))\right)$

$\leq \epsilon/2 + \sum_{v_b \in V(n)} \mathbb{E}_{bt} \left[ \mathbb{1}\{v_b \in D^n_r(v_0)\} \sum_{l=\lfloor r/2R \rfloor}^r \sum_{\{v_1, v_2, \cdots, v_l\}} \prod_{i=1}^l |V_p^{v_i}| \times \frac{1}{l!} \right]$

$\overset{(i)}{\leq} \epsilon/2 + \sum_{v_b \in V(n)} \mathbb{E}_{bt} \left[ \mathbb{1}\{v_b \in D^n_r(v_0)\} \sum_{l=\lfloor r/2R \rfloor}^r \sum_{\{v_1, v_2, \cdots, v_l\}} \frac{1}{l!} \times \underbrace{\left(1 + (1-e^{-d})^{-1} \sum_{i=1}^R d^i\right)^l}_{C(d,R)^l} \right]$

$$\leq \epsilon/2 + \sum_{l=\lfloor r/2R \rfloor}^r \binom{r}{l} \frac{1}{l!} C(d,R)^l d^r, \tag{8}$$

where $(i)$ uses the fact that conditioning on $v_b \in D^n_r(v_0)$, the path connecting $v_0$ and $v_b$, and the choice of $v_1, \cdots, v_l$, the conditional expectation of $\prod_{i=1}^l |V_p^{v_i}|$ is upper bounded by $C(d,R)^l$. By Stirling's formula, there exists numerical constants $C_1$ and $C_2$, such that for any positive integer $m$,

$$C_2 \sqrt{m} \left(\frac{m}{e}\right)^m \leq m! \leq C_1 \sqrt{m} \left(\frac{m}{e}\right)^m.$$

Then application of Stirling's formula gives us

$$\sum_{l=\lfloor r/2R \rfloor}^r \binom{r}{l} \frac{1}{l!} C(d,R)^l d^r$$

$$\leq \sum_{l=\lfloor r/2R \rfloor}^r \frac{2^r d^r}{C_2} \left(\frac{e}{l}\right)^l C(d,R)^l$$

$$\leq \frac{2^{2R} d^{2R}}{C_2} \sum_{l=\lfloor r/2R \rfloor}^\infty \left(\frac{2^{2R+2} R C(d,R) d^{2R} e}{r}\right)^l. \tag{9}$$

where we use the fact that $r/2R - 1 \leq l \leq r/2R$ and $2R/(r - 2R) \leq 4R/r$ for all $r \geq 2^{2R+3}RC(d,R)d^{2R}e$. Furthermore, with $r$ having value exceeding this threshold, $\left(\frac{2^{2R+2}RC(d,R)d^{2R}e}{r}\right) \leq \frac{1}{2}$, thus

$$\frac{2^{2R}d^{2R}}{C_2}\sum_{l=\lfloor r/2R \rfloor}^{\infty}\left(\frac{2^{2R+2}RC(d,R)d^{2R}e}{r}\right)^l \leq \frac{2^{2R+1}d^{2R}}{C_2}\left(\frac{2^{2R+2}RC(d,R)d^{2R}e}{r}\right)^{\lfloor r/2R \rfloor}$$

$$= C_R\left(\frac{2^{2R+2}RC(d,R)d^{2R}e}{r}\right)^{\lfloor r/2R \rfloor}, \qquad (10)$$

where $C_R$ is a constant that depends only on $d, R$. Then there exists an $r_\epsilon \in \mathbb{N}^+$, such that for $r \geq r_\epsilon$,

$$C_R\left(\frac{2^{R+2}RC(d,R)d^{2R}e}{r}\right)^{\lfloor r/2R \rfloor} \leq \epsilon/2. \qquad (11)$$

Combining equations (8), (9), (10) and (11) gives us $\mathbb{P}(\mathcal{G}_{v_0}^n \text{ is a subgraph of } \mathsf{B}_r^n(v_0)) \leq \epsilon$ for large enough $n$ and $r$. Note that the choice of $r_\epsilon$ indeed only depends on $R, d$ and $\epsilon$. Having decided the value of $r_\epsilon$, what remains to be done is to select $n_\epsilon$ large enough to accommodate with this choice of $r_\epsilon$ such that for $n \geq n_\epsilon$, the $r_\epsilon + R$ neighborhood of $v_0$ is locally tree-like with high probability as in Lemma 1. Note that since the choice of $r_\epsilon$ depends only on $R, d$ and $\epsilon$, the choice of $n_\epsilon$ essentially depends only on $R, d$ and $\epsilon$. This finishes the proof of the first part of Theorem 1.

As for the second part of the theorem, we simply reverse the direction of information flow analysis and everything else remains the same.

### E.2 Proof of Corollary 1

By Theorem 1, for any $\epsilon > 0$, there exists $n_\epsilon, r_\epsilon \in \mathbb{N}^+$, such that for all $n \geq n_\epsilon$,

$$\mathbb{E}\left[\max_{\pi \in \mathfrak{S}_k}\frac{1}{n}\sum_{i=1}^{n}\mathbb{1}\{\mathcal{A}(i; G(n)) = \pi \circ \tau(i)\}\right]$$

$$\leq \mathbb{E}[\max_{\pi \in \mathfrak{S}_k}\frac{1}{n}\sum_{i=1}^{n}(\mathbb{1}\{\mathcal{A}(i; G(n)) = \pi \circ \tau(i), \mathcal{G}_i^n \text{ is a subgraph of } \mathsf{B}_{r_\epsilon}^n(i)\}$$

$$+ \mathbb{1}\{\mathcal{G}_i^n \text{ is not a subgraph of } \mathsf{B}_{r_\epsilon}^n(i)\})]$$

$$\leq \epsilon + \sup_{\mathcal{A}'}\mathbb{E}\left[\max_{\pi \in \mathfrak{S}_k}\frac{1}{n}\sum_{i=1}^{n}\mathbb{1}\{\mathcal{A}'(i; G(n)) = \pi \circ \tau(i)\}\right],$$

where in the last line $\mathcal{A}'$ is taken over the family of $r_\epsilon$-local algorithms. Lemma 1 implies that the last line above has limiting supremum no larger than $1/k + \epsilon$ as $n \to \infty$ (for detailed arguments, see [KMS16]). Since $\epsilon$ is arbitrary, then the corollary directly follows.

## F Analysis of streaming belief propagation with side information

In this section we prove Theorem 2. Let $F(x) = \frac{1}{2}\log\left(\frac{e^{2x}a+b}{e^{2x}b+a}\right)$, $h_1 = \frac{1}{2}\log\frac{1-\alpha}{\alpha}$, and $h_2 = \frac{1}{2}\log\frac{\alpha}{1-\alpha}$. With $k = 2$, notice that Algorithm 1 can be equivalently reduced to the following form, with which we will continue our proof:

---

**Algorithm 3** $R$-local streaming belief propagation with $k = 2$

---

1: Initialization: $V(0) = E(0) = G(0) = \emptyset$.
2: **for** $t = 1, 2, \cdots, n$ **do**
3:     $V(t) \leftarrow V(t-1) \cup \{v(t)\}$
4:     $E(t) \leftarrow E(t-1) \cup \{(v(t), v) : v \in V(t-1), (v(t), t) \in E\}$
5:     $G(t) \leftarrow (V(t), E(t))$
6:     **for** $v \in D_1^t(v(t))$ **do**

7:      $M_{v \to v(t)} \leftarrow h_{\tilde{\tau}(v)} + \sum\limits_{v' \in D_1^{t-1}(v)} F(M_{v' \to v})$

8:      **end for**
9:      **for** $r = 1, 2, \cdots, R$ **do**
10:        **for** $v \in D_r^t(v(t))$ **do**
11:          Let $v' \in D_1^t(v)$ be a vertex which is on a shortest path connecting $v$ and $v(t)$.
12:          $M_{v' \to v} \leftarrow h_{\tilde{\tau}(v')} + \sum\limits_{v'' \in D_1^t(v') \backslash \{v\}} F(M_{v'' \to v'})$

13:        **end for**
14:      **end for**
15: **end for**
16: **for** $u \in V(n)$ **do**
17:      $M_u \leftarrow h_{\tilde{\tau}(u)} + \sum\limits_{u' \in D_1^n(u)} F(M_{u' \to u})$

18:      Output $-\mathbb{1}\{M_u \geq 0\} + 2$ as an estimate for $\tau(u)$.
19: **end for**

---

We start the proof by introducing the following definition and lemmas:

**Definition 5** (Output of belief propagation). *Let $T = (V(T), E(T))$ be a tree rooted at $u$. Let $L_T$ be the set of leaves of $T$: $L_T = \{v : v \text{ has degree } 1 \text{ in } T\}$. For each $v \in \partial T$, assume we are given $M_v^{input} \in \mathbb{R}$ which we refer to as the* input belief *into $T$ at $v$. For each $v \in V(T)$, suppose we observe a noisy label $\tilde{\tau}(v) \in \{1, 2\}$, and denote the set of children of $v$ in $T$ by $\mathcal{C}(v)$. Given the model parameters $a, b$ and $\alpha$,*

1. *For $v \in \partial T$, set $\tilde{M}_{v \to pa(v)} = M_v^{input}$, where $pa(v)$ is the parent vertex of $v$ in $T$.*

2. *Denote the depth of tree $T$ by $R$. For $r = R - 1, R - 2, \cdots, 1$, sequentially conduct the following updates: for any $v \in D_r^n(v_0)$, let*

$$\tilde{M}_{v \to pa(v)} = h_{\tilde{\tau}(v)} + \sum_{v' \in \mathcal{C}(v)} F(\tilde{M}_{v' \to v}).$$

3. *We define the output of belief propagation $\mathrm{BPROP}(u; T, \{M_v^{input} : v \in \partial T\}, \tilde{\tau}, \alpha, a, b)$ on the tree $T$ as follows:*

$$\mathrm{BPROP}(u; T, \{M_v^{input} : v \in \partial T\}, \tilde{\tau}, \alpha, a, b) = h_{\tilde{\tau}(u)} + \sum_{u' \in \mathcal{C}(u)} F(\tilde{M}_{u' \to u}).$$

**Lemma 2.** *Let $(M_u)_{u \in V(n)}$ be the output of Algorithm 3, under $\mathrm{StSSBM}(n, 2, a, b, \alpha)$, for any $\epsilon > 0$, there exists $r_\epsilon, n_\epsilon \in \mathbb{N}^+$, such that for any $u \in V(n)$, $n \geq n_\epsilon$, with probability at least $1 - \epsilon$, the following holds:*

$$M_u = \mathrm{BPROP}(u; T, \{M_v^{input} = h_{\tilde{\tau}(v)} : v \in \partial T\}, \tilde{\tau}, \alpha, a, b)$$

*for some (random) tree $T$ rooted at $u$, with the depth of $T$ no larger than $r_\epsilon$.*

*Proof.* By Lemma 1 and Theorem 1, for any $\epsilon > 0$, there exists $r_\epsilon, n_\epsilon \in \mathbb{N}^+$, such that with probability no less than $1 - \epsilon$, (1) $\mathcal{G}_u^n$ is a subgraph of $\mathsf{B}_{r_\epsilon}^n(u)$, so $M_u$ is a function of $\bar{\mathcal{G}}_u^n$ and (2) $\mathsf{B}_{r_\epsilon}^n(u)$ is a tree. The result then follows by observing the iterating formulas of Algorithm 3. Furthermore, if the event just described occurs, then the depth of $T$ is no larger than $r_\epsilon$. $\square$

**Lemma 3.** *Consider Algorithm 3, denote the value of $M_{v \to v'}$ before vertex $v(t+1)$ arrives by $M_{v \to v'}^t$. Then the following equation holds for some random time indices $\{t(v) \in [n] : v \in D_R^n(u)\}$:*

$$M_u = \mathrm{BPROP}(u; \mathsf{B}_R^n(u), \{M_v^{input} = M_{v \to pa(v)}^{t(v)} : v \in D_R^n(u)\}, \tilde{\tau}, \alpha, a, b).$$

*Proof.* We conduct this proof by induction. Actually, we will show a stronger result. Instead of focusing only on $R$-local streaming belief propagation, we will show a more general result for $N$-local streaming belief propagation with any $N \geq R$. Specifically, we will consider the following algorithm:

1. At each time $t$, a new vertex $v(t)$ is revealed. For $v \in D_1^t(v(t))$, define $M_{v \to v(t)}(N) = h_{\tilde{\tau}(v)} + \sum\limits_{v' \in D_1^t(v) \setminus \{v(t)\}} F(M_{v' \to v}(N))$.

2. For $r = 1, 2, \cdots, N$, sequentially conduct the following updates: for each $v \in D_r^t(v)$, let $v' \in D_1^t(v)$ be a vertex which is on one of the shortest path connecting $v(t)$ and $v$, update or initialize the value of $M_{v' \to v}(N) = h_{\tilde{\tau}(v')} + \sum\limits_{v'' \in D_1^t(v') \setminus \{v\}} F(M_{v'' \to v'}(N))$. Since the graph is with high probability locally tree-like, such $v'$ is with high probability unique.

3. Repeat step 1 and step 2 for all $1 \leq t \leq n$.

For $(v, v') \in E(t)$, let $M_{v \to v'}^t(N)$ denote the value of $M_{v \to v'}(N)$ after t-th iteration obtained by the $N$-local streaming belief propagation algorithm. Let $t_R(u)$ be the first time such that all vertices in $V_R^n(u)$ have been revealed:

$$t_R(u) := \inf\{t \in [n] : V_R^n(u) = V_R^t(u)\}.$$

Let $M_u^t(N) = h_{\tilde{\tau}(u)} + \sum\limits_{u' \in D_1^t(u)} F(M_{u' \to u}^t(N))$. Now instead of proving Lemma 3, we show that for any $N \geq R$, and any $t \geq t_R(u)$, if $\mathsf{B}_R^n(u)$ is a tree, then for some random time indices $\{t(v) \in [n] : v \in D_R^t(u)\}$, we have

$$M_u^t(N) = \text{BPROP}(u; \mathsf{B}_R^n(u), \{M_v^{input} = M_{v \to \text{pa}(v)}^{t(v)}(N) : v \in D_R^n(u)\}, \tilde{\tau}, \alpha, a, b). \tag{12}$$

Note that for all $t \geq t_R(u)$, $M_u^t(R) = M_u^n(R) = M_u$, therefore the result we have just described is indeed a stronger version of Lemma 3. Then we will prove this stronger result by performing induction on $R$. Equation (12) obviously holds for $R = 1$. Now suppose equation (12) holds for $R = r$, we will show it holds for $R = r + 1$ by induction. If $\mathsf{B}_{r+1}^n(u)$ is a tree rooted at $u$, for any $N \geq r + 1$, $u' \in D_1^n(u)$, let $\bar{T}_{u'}^r$ be the depth $r$ subtree consisting of $u'$ and its descendants in $\mathsf{B}_{r+1}^n(u)$. Let

$$\bar{t}_r(u') = \inf\{t : \bar{T}_{u'}^r \text{ is a subgraph of } \mathsf{B}_{r+1}^t(u)\}.$$

For $u_1, u_2 \in D_1^n(u)$, $u_1 \neq u_2$, we have $\bar{t}_r(u_1) \neq \bar{t}_r(u_2)$, and $t_*(u) \neq \bar{t}_r(u_1)$, where $t_*$ is defined in Section G. For $u' \in D_1^n(u)$, by the induction hypothesis, for $t \geq t_*(u) \vee \bar{t}_r(u')$, there exists random time indices $\{t(v) \in [n] : v \text{ being a leaf vertex in } \bar{T}_{u'}^r\}$, such that

$$M_{u' \to u}^t(N) = \text{BPROP}(u'; \bar{T}_{u'}^r, \{M_v^{input} = M_{v \to pa(v)}^{t(v)}(N) : v \text{ being a leaf vertex in } \bar{T}_{u'}^r\}, \tilde{\tau}, \alpha, a, b).$$

Since $t_{r+1}(u) \geq \bar{t}_r(u') \vee t_*(u)$ for all $u' \in D_1^n(u)$, then for all $t \geq t_{r+1}(u)$, there exists random time indices $\{t(v) \in [n] : v \in D_{r+1}^t(u)\}$, such that

$$M_u^t(N) = h_{\tilde{\tau}(u)} + \sum\limits_{u' \in D_1^t(u)} F(M_{u' \to u}^t(N))$$
$$= \text{BPROP}(u; \mathsf{B}_{r+1}^n(u), \{M_v^{input} = M_{v \to \text{pa}(v)}^{t(v)}(N) : v \in D_{r+1}^t(u)\}, \tilde{\tau}, \alpha, a, b),$$

which finishes the proof of this lemma.

### F.1 Proof of Theorem 2

For $u \in V$, let $(\lambda_{r_\epsilon}, T_u^{r_\epsilon}) \sim \mathbb{P}_{r_\epsilon, (a+b)/2, b/(a+b)}^T$ where $T_u^{r_\epsilon}$ is the tree and $\lambda_{r_\epsilon}$ is the set of labels. For vertices in $T_u^{r_\epsilon}$, we denote the set of noisy labels generated independently with incorrect probability $\alpha$ by $\tilde{\lambda}_{r_\epsilon}$. For any $\epsilon > 0$, by Theorem 1 and Lemma 1, there exists $r_\epsilon, n_\epsilon \in \mathbb{N}^+$, such that for all $n \geq n_\epsilon$, with probability at least $1 - \epsilon$: (1) $\mathcal{G}_u^n$ is a subgraph of $\mathsf{B}_{r_\epsilon}^n(u)$, (2) $(\lambda_{r_\epsilon}, \tilde{\lambda}_{r_\epsilon}, T_u^{r_\epsilon}) = (\tau(V_{r_\epsilon}^n(u)), \tilde{\tau}(V_{r_\epsilon}^n(u)), \mathsf{B}_{r_\epsilon}^n(u))$. Further there exists a coupling of $(\lambda_{r_\epsilon}, \tilde{\lambda}_{r_\epsilon}, T_u^{r_\epsilon})$ and $(\tau(V_{r_\epsilon}^n(u)), \tilde{\tau}(V_{r_\epsilon}^n(u)), \mathsf{B}_{r_\epsilon}^n(u))$ such that defining

$$S_\epsilon := \left\{ (\lambda_{r_\epsilon}, \tilde{\lambda}_{r_\epsilon}, T_u^{r_\epsilon}) = (\tau(V_{r_\epsilon}^n(u)), \tilde{\tau}(V_{r_\epsilon}^n(u)), \mathsf{B}_{r_\epsilon}^n(u)), \mathcal{G}_u^n \text{ is a subgraph of } \mathsf{B}_{r_\epsilon}^n(u) \right\},$$

then $\mathbb{P}(S_\epsilon) \geq 1 - \epsilon$ for all $n \geq n_\epsilon$. In the following parts of the analysis, we always assume $S_\epsilon$ occurs. Let $\mathbb{P}_{bt}$ denote the probability distribution of $(\lambda_{r_\epsilon}, \tilde{\lambda}_{r_\epsilon}, T_u^{r_\epsilon})$. According to Lemma 2 and 3, conditioning on $S_\epsilon$, there exists $T_u$ being a tree rooted at $u$, such that $\mathsf{B}_R^n(u)$ is a subgraph of $T_u$ and $T_u$ is a subgraph of $\mathsf{B}_{r_\epsilon}^n(u)$. Furthermore, $M_u$ can be expressed as:

$$M_u = \mathrm{BPROP}(u; T_u, \{M_v^{input} = h_{\tilde{\tau}(v)} : v \in \partial T_u\}, \tilde{\tau}, \alpha, a, b) = \frac{1}{2} \log \frac{\mathbb{P}_{bt}\left(\tau(u) = 1 \,|\, T_u, \tilde{\tau}(T_u)\right)}{\mathbb{P}_{bt}\left(\tau(u) = 2 \,|\, T_u, \tilde{\tau}(T_u)\right)},$$

where $\partial T_u$ is the set of leaf vertices in $T_u$, $\tilde{\tau}(T_u)$ refers to the set of noisy labels of vertices in $T_u$. Then for all $n \geq n_\epsilon$, we have

$$Q_n(\mathcal{A}_R) \geq \mathbb{P}(\{\mathcal{A}_R(u; G(n), \tilde{\tau}) = 1, \tau(u) = 1\} \cap S_\epsilon) + \mathbb{P}(\{\mathcal{A}_R(u; G(n), \tilde{\tau}) = 2, \tau_u = 2\} \cap S_\epsilon)$$

$$\geq \frac{1}{2} \mathbb{P}_{bt}(M_u \geq 0 \mid \tau(u) = 1) + \frac{1}{2} \mathbb{P}_{bt}(M_u < 0 \mid \tau(u) = 2) - \epsilon$$

$$= \frac{1}{2} + \mathbb{E}_{bt}\left[\left|\mathbb{P}_{bt}(\tau(u) = 1 \mid T_u, \tilde{\tau}(T_u)) - \frac{1}{2}\right|\right] - \epsilon.$$

We further have

$$\mathbb{E}_{bt}\left[\mathbb{P}_{bt}(\tau(u) = 1 | T_u, \tilde{\tau}(T_u)) \mid \mathsf{B}_R^n(u), \tilde{\tau}(V_R^n(u))\right] = \mathbb{P}_{bt}(\tau(u) = 1 | \mathsf{B}_R^n(u), \tilde{\tau}(V_R^n(u))).$$

Since $x \mapsto |x - \frac{1}{2}|$ is convex, therefore, by Jensen's inequality, for all $n \geq n_\epsilon$,

$$Q_n(\mathcal{A}_R) \geq \frac{1}{2} + \mathbb{E}_{bt}\left[\left|\mathbb{P}_{bt}(\tau(u) = 1 \,|\, \mathsf{B}_R^n(u), \tilde{\tau}(V_R^n(u))) - \frac{1}{2}\right|\right] - \epsilon.$$

Since $\epsilon$ is arbitrary, we have

$$\liminf_{n \to \infty} \left(Q_n(\mathcal{A}_R) - \frac{1}{2} - \mathbb{E}_{bt}\left[\left|\mathbb{P}_{bt}(\tau(u) = 1 \,|\, \mathsf{B}_R^n(u), \tilde{\tau}(V_R^n(u))) - \frac{1}{2}\right|\right]\right) \geq 0. \tag{13}$$

According to Lemma 3.7 in [MX16], we have

$$\lim_{n \to \infty} \left| Q_n(\mathcal{A}_R^{\mathrm{off}}) - \frac{1}{2} - \mathbb{E}_{bt}\left[\left|\mathbb{P}_{bt}(\tau(u) = 1 | \mathsf{B}_R^n(u), \tilde{\tau}(V_R^n(u))) - \frac{1}{2}\right|\right]\right| = 0. \tag{14}$$

Combining equations (13) and (14), we have

$$\liminf_{n \to \infty} \left(Q_n(\mathcal{A}_R) - Q_n(\mathcal{A}_R^{\mathrm{off}})\right) \geq 0,$$

Thus finishes the proof of Theorem 2.

## F.2 Proof of Corollary 2

According to Theorem 2.3 in [MX16], under the three regimes listed in this theorem, we have

$$\lim_{R \to \infty} \limsup_{n \to \infty} \left(Q_n^* - Q_n(\mathcal{A}_R^{\mathrm{off}})\right) = 0. \tag{15}$$

Combining (15) and Theorem 2, we have

$$\limsup_{n \to \infty} \left(Q_n^* - Q_n(\mathcal{A}_R)\right) \leq \limsup_{n \to \infty} \left(Q_n^* - Q_n(\mathcal{A}_R^{\mathrm{off}})\right) + \limsup_{n \to \infty} \left(Q_n(\mathcal{A}_R^{\mathrm{off}}) - Q_n(\mathcal{A}_R)\right)$$

$$\leq \limsup_{n \to \infty} \left(Q_n^* - Q_n(\mathcal{A}_R^{\mathrm{off}})\right),$$

thus

$$\limsup_{R \to \infty} \limsup_{n \to \infty} \left(Q_n^* - Q_n(\mathcal{A}_R)\right) \leq \lim_{R \to \infty} \limsup_{n \to \infty} \left(Q_n^* - Q_n(\mathcal{A}_R^{\mathrm{off}})\right) = 0.$$

Since $Q_n^* \geq Q_n(\mathcal{A}_R)$, the other direction naturally holds, thus finishes the proof of Corollary 2. $\quad\square$

# G  Local streaming algorithms with summary statistics

The class of local algorithms is somewhat restrictive. In practice we can imagine keeping a small memory containing global information and updating it each time a new vertex joins. We will not consider general streaming algorithms under a memory constraint; we instead consider a subclass that we name 'local streaming algorithms with summary statistics'. For instance, in social network setting, such algorithm has access not only to local information as in Definition 1, but also global statistics like the average number of friends.

Formally speaking, the state of the algorithm at time $t$ is encoded in two vectors $\boldsymbol{w}^t = (w_i^t)_{i \in V(t)} \in (\mathbb{R}^m)^{V(t)}$, $\boldsymbol{e}^t = (e_{ij}^t)_{(i,j) \in E(t)} \in (\mathbb{R}^m)^{E(t)}$, indexed respectively by the vertices and edges of $G(t)$. Here $m$ is a fixed integer independent of $n$. These are initialized to independent random variables $w_i^{t_*(i)-1} \overset{iid}{\sim} P_w$, $e_{ij}^{t_*(i) \vee t_*(j)-1} \overset{iid}{\sim} P_e$, where $t_*(i)$ is the time at which vertex $i$ joins the graph $(v(t_*(i)) = i)$, $t_*(i) \vee t_*(j) \triangleq \max\{t_*(i), t_*(j)\}$, and $P_w, P_e$ are probability distributions over $\mathbb{R}^m$.

At each $t \in [n]$, a new vertex $v(t)$ joins the graph, and a 'range of action' $(\mathsf{V}_t^{\mathsf{act}}, \mathsf{E}_t^{\mathsf{act}})$ is decided, with $\mathsf{V}_t^{\mathsf{act}} \subseteq V(t)$ a vertex set and $\mathsf{E}_t^{\mathsf{act}} \subseteq E(t)$ an edge set. We assume $(\mathsf{V}_t^{\mathsf{act}}, \mathsf{E}_t^{\mathsf{act}})$ to depend uniquely on the $R$-neighborhood of $v(t)$, $\mathsf{B}_R^t(v(t))$, and to be such that: $(i)$ the range of action is a subset of the neighborhood: $\mathsf{V}_t^{\mathsf{act}} \subseteq V_R^t(v(t))$, $\mathsf{E}_t^{\mathsf{act}} \subseteq E_R^t(v(t))$; and $(ii)$ the range of action has bounded size: $|\mathsf{V}_t^{\mathsf{act}}| + |\mathsf{E}_t^{\mathsf{act}}| \le C_{act} = C_{act}(R)$ which does not scale with $n$. Notice that the second condition is only required because the maximum degree in $G(n)$ is $\log n$, and it is to avoid pathological behavior due to high-degree vertices; we believe it should be possible to avoid it at the cost of extra technical work.

At each time $t$, the algorithm updates the quantities $w_i^t, e_{ij}^t$ in the range of action:

$$w_i^t = F_w^t(\boldsymbol{w}^{t-1}(\mathsf{V}_t^{\mathsf{act}}), \boldsymbol{e}^{t-1}(\mathsf{E}_t^{\mathsf{act}}), \bar{w}^{t-1}, \bar{e}^{t-1}|i), \qquad \forall i \in \mathsf{V}_t^{\mathsf{act}},$$
$$e_{ij}^t = F_e^t(\boldsymbol{w}^{t-1}(\mathsf{V}_t^{\mathsf{act}}), \boldsymbol{e}^{t-1}(\mathsf{E}_t^{\mathsf{act}}), \bar{w}^{t-1}, \bar{e}^{t-1}|i,j), \qquad \forall (i,j) \in \mathsf{E}_t^{\mathsf{act}}.$$

Here $\boldsymbol{w}^{t-1}(\mathsf{V}_t^{\mathsf{act}}), \boldsymbol{e}^{t-1}(\mathsf{E}_t^{\mathsf{act}})$ are the restrictions of $\boldsymbol{w}^{t-1}, \boldsymbol{e}^{t-1}$ to the range of action sets, and $\bar{w}^{t-1}, \bar{e}^{t-1}$ are summary statistics, updated according to:

$$\bar{w}^t = \frac{1}{|V(t)|} \sum_{v \in V(t)} w_v^t, \quad \bar{e}^t = \frac{1}{|E(t)|} \sum_{(i,j) \in E(t)} e_{ij}^t.$$

Finally, vertex labels are estimated using a function $\hat{\tau} : \mathbb{R}^m \to [k]$. Namely, label at vertex $v$ is estimated at time $t$ as $\hat{\tau}(w_v^t)$. If $w_i^t = $ number of friends user $i$ has at time $t$, then $m = 1$, and the corresponding summary statistics is the average number of friends per user in the current network.

We next establish that, under mild assumptions, local streaming algorithms with summary statistics cannot achieve non-trivial reconstruction in the symmetric model $\mathrm{StSSBM}(n, k, a, b)$. Notice that this claim cannot hold for a general algorithm in this class. Indeed, each node $i$ could encode the structure of (a bounded-size subgraph) $\mathsf{B}_R^t(i)$ in the decimal expansion of $w_i^t$, in such a way that distinct vertices use non-overlapping sets of digits. Then the summary statistics $\bar{w}^t$ would contain the structure of the whole graph. We avoid this by requiring the update functions to be bounded Lipschitz, and adding a small amount $\varepsilon$ of noise to $w_i^n$, before taking a decision. Informally speaking, this means that we assume small change in the quantitative result can not lead to a big difference in the qualitative outcome, thus adding small perturbation will not have huge affect on the final output.

**Theorem 3.** *Assume that there exist numerical constant $L_F$, independent of $n$, such that for all $t \in [n]$, all $i \in [t]$ and all $1 \le j < l \le t$, we have $\|F_w^t(\cdot|i)\|_\infty, \|F_e^t(\cdot|j,l)\|_\infty \le L_F$ and $\|F_w^t(\cdot|i)\|_{\mathrm{Lip}}, \|F_e^t(\cdot|j,l)\|_{\mathrm{Lip}} \le L_F$. (Here $\|f\|_{\mathrm{Lip}}$ denotes the Lipschitz modulus of function $f$.) Let $\{w_i^t : t \le n, i \in V(t)\}$ be the vertex variables generated by the local streaming algorithm with summary statistics defined by functions $F_w, F_e$. Let $\hat{\tau} : \mathbb{R}^m \to [k]$ and $(U_{ij})_{i \le n, j \le m} \overset{iid}{\sim} \mathsf{Unif}([-1,1])$ independent of the other randomness. Let $U_i = (U_{ij})_{j \le m} \in \mathbb{R}^m$. Then under $\mathrm{StSSBM}(n, k, a, b)$, for any $\varepsilon > 0$,*

$$\limsup_{n \to \infty} \mathbb{E}\left[ \max_{\pi \in \mathfrak{S}_k} \frac{1}{n} \sum_{i=1}^n \mathbb{1}(\hat{\tau}(w_i^n + \varepsilon U_i) = \pi \circ \tau(i)) \right] = \frac{1}{k}.$$

# H Analysis of local streaming algorithms with summary statistics

For the sake of simplicity, we assume $m = 1$. We point out that extension to $m \geq 2$ is straightforward.

## H.1 An auxiliary algorithm

To prove Theorem 3, we first introduce an auxiliary algorithm (Algorithm 4) which is almost a local algorithm. Then we show that the original algorithm can be well approximated by the proposed auxiliary algorithm (Lemma 4). Finally, we show that the auxiliary algorithm can not achieve non-trivial estimation accuracy (Lemma 5).

Let $\delta \in (0, 1)$ be a small constant independent of $n$, and let $n_h = |\{v \in V(n) : \tau(v) = h\}|$, $h \in [k]$. We shall perform a global algorithm up to time $\lceil \delta n \rceil$, followed by a local algorithm. To represent the information up to time $\lceil \delta n \rceil$ and the size of communities at time $n$, we introduce the following sigma-algebra:

$$\mathcal{F}_a := \sigma\{G(\lceil \delta n \rceil), (v(i), \xi_{v(i)}, \tau(v(i))), w_{v(i)}^{i-1}, e_{v(j)v(s)}^{j \vee s - 1}, n_h :$$
$$1 \leq i, j, s \leq \lceil \delta n \rceil, (v(j), v(s)) \in E(\lceil \delta n \rceil), h \in [k]\}.$$

In the auxiliary algorithm, we attach a number $b_i^t$ to node $i$ at time $t$, with the same initialization as $w$: $b_i^{t_*(i)-1} = w_i^{t_*(i)-1}$. Similarly, we attach to edge $(v(i), v(j))$ at time $t$ a number $c_{ij}^t = c_{ji}^t$ with initialization $c_{ij}^{t_*(i) \vee t_*(j) - 1} = e_{ij}^{t_*(i) \vee t_*(j) - 1}$. Note that $w_i^{t_*(i)-1}$ and $e_{ij}^{t_*(i) \vee t_*(j) - 1}$ are as defined in the original algorithm. Denote the vector containing all $b$'s attached to vertices in $\mathsf{V}_s^{\mathsf{act}}$ at time $t$ by $\boldsymbol{b}_s^t$, and the vector containing all $c$'s attached to edges in $\mathsf{E}_s^{\mathsf{act}}$ at time $t$ by $\boldsymbol{c}_s^t$. Similarly, let $\boldsymbol{w}_s^t$ and $\boldsymbol{e}_s^t$ denote the restrictions of $\boldsymbol{w}^t$ and $\boldsymbol{e}^t$ to $\mathsf{V}_s^{\mathsf{act}}$ and $\mathsf{E}_s^{\mathsf{act}}$, respectively. For $\lceil \delta n \rceil \leq t \leq \lceil n \rceil$, let

$$\mathsf{average}_b(t) = \frac{1}{t} \sum_{i=1}^{t} b_{v(i)}^t, \qquad \mathsf{average}_c(t) = \frac{1}{|E(t)|} \sum_{(j,k) \in E(t)} c_{jk}^t.$$

Then we introduce the following auxiliary algorithm:

---

**Algorithm 4** Auxiliary algorithm with summary statistics

---

1: **for** $i \in V(\lceil \delta n \rceil)$ **do**
2: $\quad b_i^{\lceil \delta n \rceil} = w_i^{\lceil \delta n \rceil}$
3: **end for**
4: **for** $(i, j) \in E(\lceil \delta n \rceil)$ **do**
5: $\quad c_{ij}^{\lceil \delta n \rceil} = c_{ji}^{\lceil \delta n \rceil} = e_{ij}^{\lceil \delta n \rceil}$
6: **end for**
7: $\bar{b}^{\lceil \delta n \rceil} \leftarrow \mathsf{average}_b(\lceil \delta n \rceil)$
8: $\bar{c}^{\lceil \delta n \rceil} \leftarrow \mathsf{average}_c(\lceil \delta n \rceil)$
9: **for** $t = \lceil \delta n \rceil + 1, \lceil \delta n \rceil + 2, \cdots, n$ **do**
10: $\quad$ **for** $i \in \mathsf{V}_t^{\mathsf{act}}$ **do**
11: $\quad\quad b_i^t \leftarrow F_w^t(\boldsymbol{b}_t^{t-1}, \boldsymbol{c}_t^{t-1}, \mathbb{E}[\bar{b}^{t-1} \mid \mathcal{F}_a], \mathbb{E}[\bar{c}^{t-1} \mid \mathcal{F}_a] \mid i)$
12: $\quad$ **end for**
13: $\quad$ **for** $(i, j) \in \mathsf{E}_t^{\mathsf{act}}$ **do**
14: $\quad\quad c_{ij}^t \leftarrow F_e^t(\boldsymbol{b}_t^{t-1}, \boldsymbol{c}_t^{t-1}, \mathbb{E}[\bar{b}^{t-1} \mid \mathcal{F}_a], \mathbb{E}[\bar{c}^{t-1} \mid \mathcal{F}_a] \mid i, j)$
15: $\quad$ **end for**
16: $\quad \bar{b}^t \leftarrow \mathsf{average}_b(t)$
17: $\quad \bar{c}^t \leftarrow \mathsf{average}_c(t)$
18: **end for**
19: **for** $t = 1, 2, \cdots, n$ **do**
20: $\quad$ Output $\hat{\tau}(b_t^n)$ as an estimate for $\tau(t)$.
21: **end for**

---

We can prove the following lemmas regarding Algorithm 4:

**Lemma 4.** *Under the conditions stated in Theorem 3, Algorithm 4 and the original algorithm proposed in Section G are asymptotically equivalent, in the sense that as $n \to \infty$,*

$$\frac{1}{n} \sum_{i=1}^{n} |w_i^n - b_i^n| + \frac{1}{|E(n)|} \sum_{(i,j) \in E(n)} |e_{ij}^n - c_{ij}^n| \xrightarrow{P} 0.$$

**Lemma 5.** *Under the conditions stated in Theorem 3, for any $\epsilon > 0$, the following holds:*

$$\limsup_{\delta \to 0^+} \limsup_{n \to \infty} \mathbb{E}\left[\max_{\pi \in \mathfrak{S}_k} \frac{1}{n} \sum_{i=1}^{n} \mathbb{1}\{\hat{\tau}(b_i^n + \epsilon U_i) = \pi \circ \tau(i)\}\right] = \frac{1}{k}.$$

We defer the proofs of Lemma 4 and 5 to later parts of the appendix. With these two lemmas, we are able to prove Theorem 3.

### H.2 Proof of Theorem 3

For any $\delta \in (0,1)$, using Lemma 4, we conclude that for any $\zeta \in (0,1)$, $n$ large enough, with probability at least $1 - \zeta$, $\frac{1}{n} \sum_{i=1}^{n} |w_i^n - b_i^n| \le \zeta^2$. If this happens, then $\#\{i : |w_i^n - b_i^n| \ge \zeta\} \le \zeta n$. For $\epsilon$ given in the theorem, let $\psi_\epsilon(\zeta) := \mathrm{TV}(\epsilon Z_1, \zeta + \epsilon Z_2)$ where $Z_1, Z_2 \sim \mathsf{Unif}[-1,1]$ and TV stands for the total variation distance between probability distributions. One can verify that $\psi_\epsilon(\zeta) \to 0$ as $\zeta \to 0$. Then we have

$$\mathbb{E}\left[\max_{\pi \in \mathfrak{S}_k} \frac{1}{n} \sum_{i=1}^{n} \mathbb{1}\{\hat{\tau}(w_i^n + \epsilon U_i) = \pi \circ \tau(i)\}\right]$$

$$\le \mathbb{E}\left[\max_{\pi \in \mathfrak{S}_k} \frac{1}{n} \sum_{i=1}^{n} \mathbb{1}\{\hat{\tau}(w_i^n + \epsilon U_i) = \pi \circ \tau(i)\} \mathbb{1}\{\frac{1}{n} \sum_{i=1}^{n} |w_i^n - b_i^n| \le \zeta^2\}\right] + \zeta$$

$$\le \mathbb{E}\left[\max_{\pi \in \mathfrak{S}_k} \frac{1}{n} \sum_{i=1}^{n} \mathbb{1}\{\hat{\tau}(w_i^n + \epsilon U_i) = \pi \circ \tau(i), |w_i^n - b_i^n| \le \zeta\} \mathbb{1}\{\frac{1}{n} \sum_{i=1}^{n} |w_i^n - b_i^n| \le \zeta^2\}\right] + 2\zeta \tag{16}$$

$$:= \triangle. \tag{17}$$

Conditioning on given values of $w_i^n$ and $b_i^n$, we may bound the total variation distance between $w_i^n + \epsilon U_i$ and $b_i^n + \epsilon U_i$. Specifically, conditional on all $w_i^n$ and $b_i^n$, there exists $U_i' \overset{iid}{\sim} \mathsf{Unif}[-1,1]$ for $i \in [n]$, independent of $w_i^n$ and $b_i^n$, such that with probability at least $1 - \psi_\epsilon(|w_i^n - b_i^n|)$, we have $w_i^n + \epsilon U_i = b_i^n + \epsilon U_i'$. Note that $U_i'$ is independent of $w_i^n, b_i^n$, but dependent on $U_i$. Then we have

$$\triangle \le \mathbb{E}\left[\max_{\pi \in \mathfrak{S}_k} \frac{1}{n} \sum_{i=1}^{n} \mathbb{1}\{\hat{\tau}(b_i^n + \epsilon U_i') = \pi \circ \tau(i), |w_i^n - b_i^n| \le \zeta\} \mathbb{1}\{\frac{1}{n} \sum_{i=1}^{n} |w_i^n - b_i^n| \le \zeta^2\}\right]$$

$$\quad + 2\zeta + \psi_\epsilon(\zeta)$$

$$\le \mathbb{E}\left[\max_{\pi \in \mathfrak{S}_k} \frac{1}{n} \sum_{i=1}^{n} \mathbb{1}\{\hat{\tau}(b_i^n + \epsilon U_i') = \pi \circ \tau(i)\}\right] + 2\zeta + \psi_\epsilon(\zeta).$$

Since $\zeta$ can be arbitrarily small,

$$\limsup_{n \to \infty} \mathbb{E}\left[\max_{\pi \in \mathfrak{S}_k} \frac{1}{n} \sum_{i=1}^{n} \mathbb{1}\{\hat{\tau}(w_i^n + \epsilon U_i) = \pi \circ \tau(i)\}\right] \le$$

$$\limsup_{n \to \infty} \mathbb{E}\left[\max_{\pi \in \mathfrak{S}_k} \frac{1}{n} \sum_{i=1}^{n} \mathbb{1}\{\hat{\tau}(b_i^n + \epsilon U_i') = \pi \circ \tau(i)\}\right].$$

This holds for any value of $\delta$. Taking $\delta \to 0^+$ then using Lemma 5 finishes the proof of Theorem 3.

## H.3  Proof of Lemma 4

For simplicity of presentation, in the proof of this lemma we drop the edge attachments (i.e., setting them to zero), and consider only the vertex attachments. We point out that proof involving edge attachments can be conducted almost identically. By the Lipschitz continuity assumption, as the $(t+1)$-th vertex joins we have

$$\text{for all } l \in \mathsf{V}_{t+1}^{\mathsf{act}}, \qquad |w_l^{t+1} - b_l^{t+1}| \leq L_F \sum_{i \in \mathsf{V}_{t+1}^{\mathsf{act}}} |w_i^t - b_i^t| + L_F |\bar{w}^t - \bar{b}^t| + L_F |\bar{b}^t - \mathbb{E}[\bar{b}^t | \mathcal{F}_a]|, \tag{18}$$

$$\text{for all } l \notin \mathsf{V}_{t+1}^{\mathsf{act}}, \qquad |w_l^{t+1} - b_l^{t+1}| = |w_l^t - b_l^t|. \tag{19}$$

Using equations (18) and (19), we have

$$|\bar{w}^{t+1} - \bar{b}^{t+1}| \leq |\bar{w}^t - \bar{b}^t| + \frac{1}{t+1} \sum_{i \in \mathsf{V}_{t+1}^{\mathsf{act}}} |w_i^{t+1} - b_i^{t+1}| + \frac{1}{t+1} \sum_{i \in \mathsf{V}_{t+1}^{\mathsf{act}}} |w_i^t - b_i^t|$$

$$\leq \left(1 + \frac{L_F |\mathsf{V}_{t+1}^{\mathsf{act}}|}{t+1}\right) |\bar{w}^t - \bar{b}^t| + \frac{L_F |\mathsf{V}_{t+1}^{\mathsf{act}}| + 1}{t+1} \sum_{i \in \mathsf{V}_{t+1}^{\mathsf{act}}} |w_i^t - b_i^t| + \frac{L_F |\mathsf{V}_{t+1}^{\mathsf{act}}|}{t+1} |\bar{b}^t - \mathbb{E}[\bar{b}^t | \mathcal{F}_a]|. \tag{20}$$

For $t \in [n]$, let $\boldsymbol{d}_t \in \mathbb{R}^{n+1}$ with entries indexed from 0 to $n$. The first entry is set to $|\bar{w}^t - \bar{b}^t|$, and for $i \in [n]$, the entry with index $i$ is set to $|w_i^t - b_i^t|$. By definition for all $1 \leq s \leq \lceil \delta n \rceil$, $\boldsymbol{d}_s$ has only zero entries.

For $S_1, S_2, S \subseteq \{0\} \cup [n]$, let $\mathbf{1}_S \in \mathbb{R}^{n+1}$ be a vector with entries indexed by 0 to $n$, and the entry with index $\alpha$ is 1 if and only if $\alpha \in S$ otherwise it is zero. For simplicity, let $\mathbf{1}_t = \mathbf{1}_{\{t\}}$, and let $Q(S_1, S_2) := \mathbf{1}_{S_1} \mathbf{1}_{S_2}^T \in \mathbb{R}^{(n+1) \times (n+1)}$. Then define the following matrices:

$$A_t := Q(\mathsf{V}_t^{\mathsf{act}}, \{0\} \cup \mathsf{V}_t^{\mathsf{act}}) + \frac{|\mathsf{V}_t^{\mathsf{act}}| + 1}{t} Q(\{0\}, \{0\} \cup \mathsf{V}_t^{\mathsf{act}}),$$

$$A_{t_1, t_2} := Q(\mathsf{V}_{t_1}^{\mathsf{act}}, \{0\} \cup \mathsf{V}_{t_2}^{\mathsf{act}}) + \frac{|\mathsf{V}_{t_1}^{\mathsf{act}}| + 1}{t_1} Q(\{0\}, \{0\} \cup \mathsf{V}_{t_2}^{\mathsf{act}}).$$

Without loss of generality we may assume $L_F \geq 1$, then combining equations (18), (19) and (20) gives

$$\boldsymbol{d}_{t+1} \leq (I + L_F A_{t+1}) \boldsymbol{d}_t + L_F (|\bar{b}^t - \mathbb{E}[\bar{b}^t | \mathcal{F}_a]|) A_{t+1} \mathbf{1}_{v(t+1)},$$

here "$\leq$" refers to element-wise comparison. Let

$$\tilde{\boldsymbol{d}}_{\lceil \delta n \rceil} = \boldsymbol{d}_{\lceil \delta n \rceil} = \vec{0}, \qquad \tilde{\boldsymbol{d}}_{t+1} = (I + L_F A_{t+1}) \tilde{\boldsymbol{d}}_t + (|\bar{b}^t - \mathbb{E}[\bar{b}^t | \mathcal{F}_a]|) L_F A_{t+1} \mathbf{1}_{v(t+1)},$$

then we have $\boldsymbol{d}_t \leq \tilde{\boldsymbol{d}}_t$ for all $t \geq \lceil \delta n \rceil$. Furthermore, we have the following decomposition with $h_t$ defined in equation (22):

$$\langle \boldsymbol{d}_n, \vec{1} \rangle \leq \langle \tilde{\boldsymbol{d}}_n, \vec{1} \rangle = \sum_{t = \lceil \delta n \rceil + 1}^{n-1} h_t |\bar{b}^t - \mathbb{E}[\bar{b}^t | \mathcal{F}_a]|. \tag{21}$$

Before elaborating on the definition of $h_t$, we state the following lemma without proof. Notice that the proof is nothing but basic linear algebra.

**Lemma 6.** *For $n \geq t_m > t_{m-1} > \cdots > t_1 \geq \lceil \delta n \rceil + 1$, we have*

$$A(t_m, t_{m-1}, \cdots, t_1) := A_{t_m} A_{t_{m-1}} \cdots A_{t_1} = \prod_{k=1}^{m-1} \left( \frac{|\mathsf{V}_{t_k}^{\mathsf{act}}| + 1}{t_k} + |\mathsf{V}_{t_k}^{\mathsf{act}} \cap \mathsf{V}_{t_{k+1}}^{\mathsf{act}}| \right) A_{t_m, t_1}.$$

Applying Lemma 6, we have for all $\lceil \delta n \rceil + 1 \leq t \leq n - 1$,

$$h_t = \sum_{n \geq t_m > \cdots > t_1 = t+1} \langle L_F^m A(t_m, t_{m-1}, \cdots, t_1) \mathbf{1}_{v(t+1)}, \vec{1} \rangle$$

$$= \sum_{n \geq t_m > \cdots > t_1 = t+1} L_F^m \left( |\mathsf{V}_{t_m}^{\mathsf{act}}| + \frac{|\mathsf{V}_{t_m}^{\mathsf{act}}| + 1}{t_m} \right) \times \prod_{k=1}^{m-1} \left( |\mathsf{V}_{t_k}^{\mathsf{act}} \cap \mathsf{V}_{t_{k+1}}^{\mathsf{act}}| + \frac{|\mathsf{V}_{t_k}^{\mathsf{act}}| + 1}{t_k} \right). \quad (22)$$

For $\lceil \delta n \rceil + 1 \leq s \leq n - 1$, let $H(s) = \sum_{t=s}^{n-1} h_t$, and let

$$C(s) = \sum_{n \geq t_m > \cdots > t_1 = s+1} L_F^m \left( |\mathsf{V}_{t_m}^{\mathsf{act}}| + \frac{|\mathsf{V}_{t_m}^{\mathsf{act}}| + 1}{t_m} \right) \times$$

$$\prod_{k=1}^{m-1} \left( |\mathsf{V}_{t_k}^{\mathsf{act}} \cap \mathsf{V}_{t_{k+1}}^{\mathsf{act}}| + \frac{|\mathsf{V}_{t_k}^{\mathsf{act}}| + 1}{t_k} \right) \mathbb{1} \left\{ |\mathsf{V}_{t_k}^{\mathsf{act}} \cap \mathsf{V}_{t_{k+1}}^{\mathsf{act}}| > 0 \right\}.$$

Then we can provide an upper bound for $H(s)$ using $C(s)$ and $H(s+1)$:

$$H(s) \leq \left( 1 + \frac{1}{\lceil \delta n \rceil} C(s) \right) H(s+1) + C(s).$$

By induction, and applying the fact that $\log(1 + x) \leq x$ for all $x > -1$, we have

$$H_{\lceil \delta n \rceil + 1} \leq \sum_{s = \lceil \delta n \rceil + 1}^{n-1} C(s) \prod_{s = \lceil \delta n \rceil + 1}^{n-1} \left( 1 + \frac{C(s)}{\lceil \delta n \rceil} \right) \leq \sum_{s = \lceil \delta n \rceil + 1}^{n-1} C(s) \exp \left( \frac{1}{\lceil \delta n \rceil} \sum_{s = \lceil \delta n \rceil + 1}^{n-1} C(s) \right).$$
$$(23)$$

Using equation (21), we can prove Lemma 4 if we can prove the following two lemmas.

**Lemma 7.** *Under the assumptions stated in Theorem 3, for all $\delta > 0$, we have $H_{\lceil \delta n \rceil + 1} = O_p(n)$.*

**Lemma 8.** *Under the assumptions stated in Theorem 3, we have*

$$\sup_{\lceil \delta n \rceil + 1 \leq t \leq n} |\bar{b}^t - \mathbb{E}[\bar{b}^t | \mathcal{F}_a]| = o_p(1).$$

The proof of Lemma 7 and 8 will be deferred to later parts of the appendix. Using Lemma 7 and 8, then apply equation (21), we have

$$\frac{1}{n} \sum_{i=1}^{n} |w_i^t - b_i^t| \leq \frac{1}{n} \langle \boldsymbol{d}_n \vec{1} \rangle \leq \frac{1}{n} H(\lceil \delta n \rceil) \sup_{\lceil \delta n \rceil + 1 \leq t \leq n-1} |\bar{b}^t - \mathbb{E}[\bar{b}^t | \mathcal{F}_a]| = o_p(1),$$

thus finishes the proof of Lemma 4.

### H.4 Proof of Lemma 7

To prove Lemma 7 we shall first provide a uniform upper bound for the expectation of $C(s)$ which is independent of $n, \delta$ and $s$. If this holds, then by Markov's inequality $\sum_{s=\lceil \delta n \rceil + 1}^{n} C(s) = O_p(n)$. Plugging this into equation (23) gives $H_{\lceil \delta n \rceil + 1} = O_p(n)$, which finishes the proof of this lemma. Let $\delta_1 := \frac{1}{2} \left( \frac{(a+(k-1)b)\delta}{k} \wedge 1 \right) \delta$. For $n$ large enough, we have $\frac{|\mathsf{V}_{t_k}^{\mathsf{act}}| + 1}{t_k} < 1$ for all $\lceil \delta n \rceil + 1 \leq t_k \leq n$, then we have

$$C(s) \leq \sum_{n \geq t_m > \cdots > t_1 = s+1} 2^m L_F^m C_{act}^m \mathbb{1} \left\{ |\mathsf{V}_{t_k}^{\mathsf{act}} \cap \mathsf{V}_{t_{k+1}}^{\mathsf{act}}| > 0 \; k = 1, 2, \cdots, m-1 \right\}$$

$$\leq \sum_{n \geq t_m > \cdots > t_1 = s+1} 2^m L_F^m C_{act}^m \mathbb{1} \left\{ d(v(t_k), v(t_{k+1})) \leq 2R, \; k = 1, 2, \cdots, m-1 \right\}. \quad (24)$$

Note that in equation (24), $m$ can be any positive integer, therefore, taking the expectation of $C(s)$ gives

$$\mathbb{E}[C(s)] \leq \sum_{m=1}^{\infty} \sum_{n \geq t_m > \cdots > t_1 = s+1} 2^m L_F^m C_{act}^m \mathbb{P}(d(v(t_k), v(t_{k+1})) \leq 2R, \; k = 1, 2, \cdots, m-1)$$

$$\leq \sum_{m=1}^{\infty} \sum_{n \geq t_m > \cdots > t_1 = s+1} 2^m L_F^m C_{act}^m \mathbb{E} \left[ \prod_{k=1}^{m-1} \frac{|V_{2R}^n(v(t_k))|}{n-k} \right]$$

$$\leq \sum_{m=1}^{\infty} \frac{2^m L^m C_{act}^m}{(m-1)!} \mathbb{E}[|V_{2R}^n(v)|^m],$$

where $v$ is an arbitrary vertex in $V(n)$. The following lemma provides an upper bound on $\mathbb{E}[|V_{2R}^n(v)|^m]$.

**Lemma 9.** *Under the assumptions of Theorem 3, we have*

$$\limsup_{n\to\infty} \sum_{m=1}^{\infty} \frac{2^m L_F^m C_{act}^m}{(m-1)!} \mathbb{E}[|V_{2R}^n(v)|^m] < \infty.$$

The proof of Lemma 9 is deferred to next subsection. With Lemma 9, we can apply Markov's inequality to prove $\sum_{s=\lceil \delta n \rceil + 1}^{n} C(s) = O_p(n)$, then Lemma 7 follows from equation (23).

### H.5   Proof of Lemma 9

To prove lemma 9, we introduce the following branching process:

1. $X_0 = 1$.
2. Let $\{Z(i,t) : i, t \in \mathbb{Z}^+\}$ be an array of i.i.d. $\mathbb{Z}^+$-valued random variables with distribution Binomial$(n, \frac{a \vee b}{n})$. For $t \geq 1$, define $X_t = \sum_{1 \leq i \leq X_{t-1}} Z(i,t)$.

Then we have

$$\mathbb{E}[|V_{2R}^n(v)|^m] \leq \mathbb{E}\left[\left(\sum_{k=0}^{2R} X_k\right)^m\right]. \tag{25}$$

Let $f_j(t) := \mathbb{E}[\exp(tX_j)]$, $\mu_m := a \vee b$. By the proof of Theorem 2.3.1 in [Ver18], we have $f_1(t) \leq \exp(\mu_m(e^t - 1))$. For $\gamma > 0$, let $t_1^\gamma := \log\left(\frac{\gamma}{\mu_m} + 1\right)$, and $t_{j+1}^\gamma =: \log\left(\frac{t_j^\gamma}{\mu_m} + 1\right)$ for $j \in \mathbb{N}^+$. Then by Proposition 5.2 in [LP17], we have

$$f_{j+1}(t_{j+1}^\gamma) \leq f_j(t_j^\gamma) \leq \cdots \leq f_1(t_1^\gamma) \leq \exp(\gamma).$$

Then for all $1 \leq j \leq 2R$ and $\gamma_0 > 0$, we have

$$\begin{aligned}
\mathbb{E}[|X_j^m|] &= \int_0^\infty m\gamma^{m-1} \mathbb{P}(X_j \geq \gamma) d\gamma \\
&\leq \int_0^\infty m\gamma^{m-1} \exp(-\gamma t_j^\gamma + \gamma) d\gamma \\
&= \int_0^{\gamma_0} m\gamma^{m-1} \exp(-\gamma t_j^\gamma + \gamma) d\gamma + \int_{\gamma_0}^\infty m\gamma^{m-1} \exp(-\gamma t_j^\gamma + \gamma) d\gamma \\
&\leq \gamma_0^m \exp(\gamma_0) + \frac{m!}{(t_j^{\gamma_0} - 1)^m}. \tag{26}
\end{aligned}$$

By equation (26), for any $\gamma_0 > 0$, we have

$$\mathbb{E}\left[\left(\sum_{k=0}^{2R} X_k\right)^m\right] \leq (2R+1)^m \gamma_0^m \exp(\gamma_0) + (2R+1)^{m-1} \sum_{j=1}^{2R} \frac{m!}{(t_j^{\gamma_0} - 1)^m} + (2R+1)^{m-1}. \tag{27}$$

Notice that we can choose $\gamma_0$ large enough, such that $\min_{1 \leq j \leq 2R} t_j^{\gamma_0} - 1 \geq 4L_F C_{act}(2R+1)$. Using equations (25) and (27), we have

$$\sum_{m=1}^{\infty} \frac{2^m L_F^m C_{act}^m}{(m-1)!} \mathbb{E}[|V_{2R}^n(v)|^m]$$

$$\leq \sum_{m=1}^{\infty} \underbrace{\frac{2^m L_F^m C_{act}^m}{(m-1)!}(2R+1)^m(\gamma_0^m \exp(\gamma_0)+1)}_{\text{I}} + \sum_{m=1}^{\infty} \underbrace{\frac{2^m L_F^m C_{act}^m (2R+1)^m m}{(\min_{1\leq j\leq 2R} t_j^{\gamma_0} - 1)^m}}_{\text{II}}.$$

The choice of $\gamma_0$ gives us equation II $\leq \sum_{m=1}^{\infty} \frac{m}{2^m} < \infty$. One can also derive equation I $< \infty$ for any value of $\gamma_0$. Note that the derived upper bounds for equations I and II are independent of $n$. Thus we have finished proving Lemma 9.

## H.6    Proof of Lemma 8

We first show a weaker result. As $n \to \infty$, we want to show $\sup_{\lceil \delta n \rceil + 1 \leq t \leq n} \mathbb{E}[|\bar{b}^t - \mathbb{E}[\bar{b}^t | \mathcal{F}_a]|^2] \to 0$. Since the $|F_w|$ functions are uniformly bounded by $L_F$, we only need to show as $n \to \infty$,

$$\sup_{\lceil \delta n \rceil + 1 \leq t \leq n} \sup_{1 \leq x < y \leq t} \left| \mathbb{E}\left[(b_{v(x)}^t - \mathbb{E}[b_{v(x)}^t | \mathcal{F}_a])(b_{v(y)}^t - \mathbb{E}[b_{v(y)}^t | \mathcal{F}_a])\right] \right| \to 0. \qquad (28)$$

For $1 \leq x < y \leq n$, $r \in \mathbb{N}^+$, we introduce the following sets in the probability space we consider:

$$A_x^r = \left\{ \mathcal{G}_{v(x)}^n \text{ is a subgraph of } \mathsf{B}_r^n(v(x)) \right\}, \qquad B_{x,y}^r = \left\{ V_{r+1}^n(v(x)) \cap V_{r+1}^n(v(y)) = \emptyset \right\}.$$

Then we have

$$|\mathbb{E}[(b_{v(x)}^t - \mathbb{E}[b_{v(x)}^t | \mathcal{F}_a])(b_{v(y)}^t - \mathbb{E}[b_{v(y)}^t | \mathcal{F}_a])]| \qquad (29)$$

$$\leq 4L_F^2 \left( \mathbb{P}((B_{x,y}^r)^c) + \mathbb{P}((A_x^r)^c) + \mathbb{P}((A_y^r)^c) \right) \qquad (30)$$

$$+ \underbrace{|\mathbb{E}[(b_{v(x)}^t - \mathbb{E}[b_{v(x)}^t | \mathcal{F}_a])(b_{v(y)}^t - \mathbb{E}[b_{v(y)}^t | \mathcal{F}_a]) \mathbb{1}_{A_x^r \cap A_y^r \cap \mathbb{1}_{B_{x,y}^r}}]|}_{C(x,y,t,n,r)}.$$

Then we provide an upper bound for $C(x,y,t,n,r)$. The following procedures are conducted **conditioning only on** $\mathcal{F}_a$. We point out that $\mathsf{B}_{r+1}^n(v(x))$ can be generated via the following procedure, which is equivalent to our proposed model $\text{StSSBM}(n,k,a,b)$.

1. If $x \geq \lceil \delta n \rceil$, then arbitrarily pick $v(x)$ in $V \backslash V(\lceil \delta n \rceil)$. Otherwise, $v(x)$ is already specified by $\mathcal{F}_a$.

2. Conditioning on $\mathcal{F}_a$ and $v(x)$, generate $\mathsf{B}_{r+1}^n(v(x))$ following conditional distribution of $\text{StSSBM}(n,k,a,b)$.

3. For $v \in V_{r+1}^n(v(x))$, if $t_*(v)$ is not yet specified, then assign it a value based on the current conditional distribution.

Given $\mathsf{B}_{r+1}^n(v(x))$ and the orderings of vertices within, we are able to judge whether $\mathcal{G}_{v(x)}^n$ is a subgraph of $\mathsf{B}_r^n(v(x))$. If $\mathcal{G}_{v(x)}^n$ is not a subgraph of $\mathsf{B}_r^n(v(x))$, then formula inside the expectation of $C(x,y,t,n,r)$ is zero. If $\mathcal{G}_{v(x)}^n$ is a subgraph of $\mathsf{B}_r^n(v(x))$, then we consider another graph $G_{new}$ which also follows $\text{StSSBM}(n,k,a,b)$:

1. We assume that conditioning on $\mathcal{F}_a$, $G_{new}$ and $G(n)$ are exactly the same.

2. Again conditioning on $\mathcal{F}_a$, we can find a bijection $\phi$ mapping vertices in $G(n)$ to vertices in $G_{new}$: if $v \in V(\lceil \delta n \rceil)$, then $\phi$ maps $v$ to the vertex in $G_{new}$ with reveal ordering $t_*(v)$; otherwise, map $v$ to an arbitrary vertex in $G_{new}$ with the same community label. For $S \subseteq V(n)$, denote the image set of $S$ obtained via mapping $\phi$ by $\phi(S)$.

3. Among all unordered vertices in $G_{new}$, we randomly pick $\tilde{v}(y)$ and assign it the revealing ordering $y$.

4. Generate the $r+1$ neighborhood of $\tilde{v}(y)$ in $G_{new}$, following the conditional distribution of $\text{StSSBM}(n,k,a,b)$ conditioning on $\mathcal{F}_a$, and denote it by $\mathsf{B}_{r+1}^n(\tilde{v}(y))$. For vertices in $\mathsf{B}_{r+1}^n(\tilde{v}(y))$ which have not been assigned orderings, we randomly assign them one.

In both $G(n)$ and $G_{new}$, we denote the $r$ neighborhood of $v$ with the information about revealing orderings of vertices within encoded by $O_r^n(v)$. Notice that conditioning on $\mathcal{F}_a$, $O_{r+1}^n(\tilde{v}(y))$ is conditionally independent of $O_{r+1}^n(v(x))$. Let

$\mathsf{NonOverlap}(x,y) =$

$\{$revealing orderings in $O_{r+1}^n(\tilde{v}(y))$ does not overlap with revealing orderings in $O_{r+1}^n(v(x))\}$.

Then we have

$$\mathcal{L}\left(O_{r+1}^n(v(y))|O_{r+1}^n(v(x)), V_{r+1}^n(v(x)) \cap V_{r+1}^n(v(y)) = \emptyset, \mathcal{F}_a\right)$$
$$=\mathcal{L}(O_{r+1}^n(\tilde{v}(y))|O_{r+1}^n(v(x)), V_{r+1}^n(\tilde{v}(y)) \cap \phi(V_{r+1}^n(v(x))) = \emptyset, \mathsf{NonOverlap}(x,y), \mathcal{F}_a).$$

where $\mathcal{L}(\cdot|\cdot)$ denotes the conditional probability law. Furthermore, the marginal probability distributions are equal: $\mathcal{L}(O_{r+1}^n(v(y))|\mathcal{F}_a) = \mathcal{L}(O_{r+1}^n(\tilde{v}(y))|\mathcal{F}_a)$. Define the following sets:

$$\tilde{A}_y^r = \left\{ \mathcal{G}_{\tilde{v}(y)}^n \text{ is a subgraph of } \mathsf{B}_r^n(\tilde{v}(y)) \right\},$$
$$\tilde{C}_{x,y}^r = \left\{ V_{r+1}^n(\tilde{v}(y)) \cap \phi(V_{r+1}^n(v(x))) = \emptyset \right\} \cap \mathsf{NonOverlap}(x,y).$$

In $G_{new}$ we can still run Algorithm 4, and denote the obtained quantity at time $t$ for vertex $\tilde{v}(y)$ by $\tilde{b}_{\tilde{v}(y)}^t$. Then we have

$$|\mathbb{E}[(b_{v(x)}^t - \mathbb{E}[b_{v(x)}^t|\mathcal{F}_a])\mathbb{1}_{A_x^r}\mathbb{1}_{B_{x,y}^r}(b_{v(y)}^t - \mathbb{E}[b_{v(y)}^t|\mathcal{F}_a])\mathbb{1}_{A_y^r}|\mathcal{F}_a]|$$
$$=|\mathbb{E}[(b_{v(x)}^t - \mathbb{E}[b_{v(x)}^t|\mathcal{F}_a])\mathbb{1}_{A_x^r}\mathbb{1}_{B_{x,y}^r}\mathbb{E}[(b_{v(y)}^t - \mathbb{E}[b_{v(y)}^t|\mathcal{F}_a])\mathbb{1}_{A_y^r}|O_{r+1}^n(v(x)), B_{x,y}^r, \mathcal{F}_a]|\mathcal{F}_a]|$$
$$=|\mathbb{E}[(b_{v(x)}^t - \mathbb{E}[b_{v(x)}^t|\mathcal{F}_a])\mathbb{1}_{A_x^r}\mathbb{1}_{B_{x,y}^r}\mathbb{E}[(\tilde{b}_{\tilde{v}(y)}^t - \mathbb{E}[\tilde{b}_{\tilde{v}(y)}^t|\mathcal{F}_a])\mathbb{1}_{\tilde{A}_y^r}|O_{r+1}^n(v(x)), \tilde{C}_{x,y}^r, \mathcal{F}_a]|\mathcal{F}_a]|$$
$$=\left|\mathbb{E}\left[(b_{v(x)}^t - \mathbb{E}[b_{v(x)}^t|\mathcal{F}_a])\mathbb{1}_{A_x^r}\mathbb{1}_{B_{x,y}^r}\frac{\mathbb{E}[(\tilde{b}_{\tilde{v}(y)}^t - \mathbb{E}[\tilde{b}_{\tilde{v}(y)}^t|\mathcal{F}_a])\mathbb{1}_{\tilde{A}_y^r}\mathbb{1}_{\tilde{C}_{x,y}^r}|O_{r+1}^n(v(x)), \mathcal{F}_a]}{\mathbb{P}(\tilde{C}_{x,y}^r|O_{r+1}^n(v(x)), \mathcal{F}_a)}\middle|\mathcal{F}_a\right]\right|$$

$$\tag{31}$$

Obviously $\mathbb{P}(\tilde{C}_{x,y}^r|O_{r+1}^n(v(x)), \mathcal{F}_a) \xrightarrow{P} 1$, also notice that $\|F_w\|_\infty$ has a uniform upper bound, then equation (31) is no larger than

$$\left|\underbrace{\mathbb{E}\left[(b_{v(x)}^t - \mathbb{E}[b_{v(x)}^t|\mathcal{F}_a])\mathbb{1}_{A_x^r}\mathbb{1}_{B_{x,y}^r}(\tilde{b}_{\tilde{v}(y)}^t - \mathbb{E}[\tilde{b}_{\tilde{v}(y)}^t|\mathcal{F}_a])\mathbb{1}_{\tilde{A}_y^r}\mathbb{1}_{\tilde{C}_{x,y}^r}\middle|\mathcal{F}_a\right]}_{\Delta}\right|+$$
$$4L_F^2\mathbb{E}\left[\left|1 - \mathbb{P}(\tilde{C}_{x,y}^r|O_{r+1}^n(v(x)), \mathcal{F}_a)^{-1}\right| \wedge 1 \mid \mathcal{F}_a\right]. \tag{32}$$

Notice that

$$\left|\Delta - \mathbb{E}\left[(b_{v(x)}^t - \mathbb{E}[b_{v(x)}^t|\mathcal{F}_a])\mathbb{1}_{A_x^r}\mathbb{1}_{B_{x,y}^r}(\tilde{b}_{\tilde{v}(y)}^t - \mathbb{E}[\tilde{b}_{\tilde{v}(y)}^t|\mathcal{F}_a])\middle|\mathcal{F}_a\right]\right| \le$$
$$4L_F^2\mathbb{P}((\tilde{A}_y^r)^c|\mathcal{F}_a) + 4L_F^2\mathbb{P}((\tilde{C}_{x,y}^r)^c|\mathcal{F}_a)$$

Conditioning on $\mathcal{F}_a$, $(\tilde{b}_{\tilde{v}(y)}^t - \mathbb{E}[\tilde{b}_{\tilde{v}(y)}^t|\mathcal{F}_a])$ is conditionally independent of $(b_{v(x)}^t - \mathbb{E}[b_{v(x)}^t|\mathcal{F}_a])\mathbb{1}_{A_x^r}\mathbb{1}_{B_{x,y}^r}$. Combining this and equations (29), (32), we have

$$|\mathbb{E}[(b_x^t - \mathbb{E}[\bar{b}^t|\mathcal{F}_a])(b_y^t - \mathbb{E}[\bar{b}^t|\mathcal{F}_a])]|$$
$$\le 4L_F^2\mathbb{P}((B_{x,y}^r)^c) + 4L_F^2\mathbb{P}((A_y^r)^c) + 4L_F^2\mathbb{P}((A_x^r)^c) + 4L_F^2\mathbb{P}((\tilde{A}_y^r)^c) + 4L_F^2\mathbb{P}((\tilde{C}_{x,y}^r)^c)+$$
$$4L_F^2\mathbb{E}\left[\left|1 - \mathbb{P}(\tilde{C}_{x,y}^r|O_{r+1}^n(v(x)), \mathcal{F}_a)^{-1}\right| \wedge 1\right].$$

According to the proof of Theorem 1 we have

$$\limsup_{n\to\infty} \sup_{x,y\in[n]} \left\{4L_F^2\mathbb{P}((A_x^r)^c) + 4L_F^2\mathbb{P}((A_y^r)^c) + 4L_F^2\mathbb{P}((\tilde{A}_y^r)^c)\right\} = 0.$$

Furthermore, we claim without proof that

$$\limsup_{n\to\infty} \sup_{x,y\in[n]} \left\{4L_F^2\mathbb{P}((B_{x,y}^r)^c) + 4L_F^2\mathbb{P}((\tilde{C}_{x,y}^r)^c)\right\} = 0, \qquad,$$
$$\limsup_{n\to\infty} \sup_{x,y\in[n]} 4L_F^2\mathbb{E}\left[\left|1 - \mathbb{P}(\tilde{C}_{x,y}^r|O_{r+1}^n(v(x)), \mathcal{F}_a)^{-1}\right| \wedge 1\right] = 0$$

Notice that these asymptotic results do not depend on $x, y$ or $t$, therefore,

$$\limsup_{n\to\infty} \sup_{\lceil \delta n\rceil+1\leq t\leq n} \sup_{1\leq x<y\leq t} \left| \mathbb{E}\left[ (b_{v(x)}^t - \mathbb{E}[b_{v(x)}^t|\mathcal{F}_a])(b_{v(y)}^t - \mathbb{E}[b_{v(y)}^t|\mathcal{F}_a]) \right] \right| = 0.$$

Then we conclude that equation (28) holds, and Lemma 8 follows from a discretization argument. Furthermore, from the proof of Lemma 8 we can deduce the following corollary:

**Corollary 3.** *Assume $(\tau, \boldsymbol{G}) \sim \mathrm{StSSBM}(n, k, a, b)$. Let $\mathcal{A}$ be any algorithm such that $\mathcal{A}(i; G(n)) \in [k]$ is a function of $\mathcal{G}_i^n$ and $\mathcal{F}_a$. Then we have as $n \to \infty$,*

$$\sup_{\mathcal{A}} \mathrm{Var}\left[ \frac{1}{n}\sum_{i=1}^n \mathbb{1}\{\mathcal{A}(i; G(n)) = \tau(i)\} \mid \mathcal{F}_a \right] \xrightarrow{P} 0.$$

*Proof.* To prove this corollary, we first show for all $x, y \in [n]$,

$$C(x, y) := \sup_{\mathcal{A}} | \mathbb{P}\left( \mathcal{A}(v(x); G(n)) = \tau(v(x)), \mathcal{A}(v(y); G(n)) = \tau(v(y)) \mid \mathcal{F}_a \right) -$$

$$\mathbb{P}\left( \mathcal{A}(v(x); G(n)) = \tau(v(x)) \mid \mathcal{F}_a \right) \mathbb{P}\left( \mathcal{A}(v(y); G(n)) = \tau(v(y)) \mid \mathcal{F}_a \right) | \xrightarrow{P} 0. \tag{33}$$

Consider the graph generating procedure described in the proof of Lemma 8 and we use the notations defined there. Furthermore, we define the following quantities(here we use tildes to represent objects in $G_{new}$):

$$\Delta_1 := \mathbb{E}\left[ \mathbb{1}\{\mathcal{A}(v(x); G(n)) = \tau(v(x))\}\mathbb{1}_{A_x^r}\mathbb{1}_{B_{x,y}^r}\mathbb{1}\{\mathcal{A}(v(y); G(n)) = \tau(v(y))\}\mathbb{1}_{A_y^r} \mid \mathcal{F}_a \right],$$

$$\Delta_2 := \mathbb{E}\left[ \mathbb{1}\{\mathcal{A}(v(x); G(n)) = \tau(v(x))\}\mathbb{1}_{A_x^r}\mathbb{1}_{B_{x,y}^r}\mathbb{1}\{\mathcal{A}(\tilde{v}(y); \tilde{G}(n)) = \tau(\tilde{v}(y))\}\mathbb{1}_{\tilde{A}_y^r}\mathbb{1}_{\tilde{C}_{x,y}^r} \mid \mathcal{F}_a \right].$$

Similar to the proof of Lemma 8, we conclude that the following formulas hold:

$$|\mathbb{P}\left( \mathcal{A}(v(x); G(n)) = \tau(v(x)), \mathcal{A}(v(y); G(n)) = \tau(v(y)) \mid \mathcal{F}_a \right) - \Delta_1|$$
$$\leq \mathbb{P}((A_x^r)^c \mid \mathcal{F}_a) + \mathbb{P}((A_y^r)^c \mid \mathcal{F}_a) + \mathbb{P}((B_{x,y}^r)^c \mid \mathcal{F}_a),$$

$$\Delta_1 = \mathbb{E}[\mathbb{1}\{\mathcal{A}(v(x); G(n)) = \tau(v(x))\}\mathbb{1}_{A_x^r}\mathbb{1}_{B_{x,y}^r}$$
$$\mathbb{E}[\mathbb{1}\{\mathcal{A}(\tilde{v}(y); \tilde{G}(n)) = \tau(\tilde{v}(y))\}\mathbb{1}_{\tilde{A}_y^r} \mid O_{r+1}^n(v(x)), \tilde{C}_{x,y}^r, \mathcal{F}_a] \mid \mathcal{F}_a]$$
$$= \mathbb{E}[\mathbb{1}\{\mathcal{A}(v(x); G(n)) = \tau(v(x))\}\mathbb{1}_{A_x^r}\mathbb{1}_{B_{x,y}^r}$$
$$\frac{\mathbb{E}\left[ \mathbb{1}\{\mathcal{A}(\tilde{v}(y); \tilde{G}(n)) = \tau(\tilde{v}(y))\}\mathbb{1}_{\tilde{A}_y^r}\mathbb{1}_{\tilde{C}_{x,y}^r} \mid O_{r+1}^n(v(x)), \mathcal{F}_a \right]}{\mathbb{P}\left( \tilde{C}_{x,y}^r \mid O_{r+1}^n(v(x)), \mathcal{F}_a \right)} \mid \mathcal{F}_a],$$

$$|\Delta_1 - \Delta_2| \leq \mathbb{E}\left[ \left| \mathbb{P}\left( \tilde{C}_{x,y}^r \mid O_{r+1}^n(v(x)), \mathcal{F}_a \right)^{-1} - 1 \right| \wedge 1 \mid \mathcal{F}_a \right],$$

$$|\Delta_2 - \mathbb{P}\left( \mathcal{A}(v(x); G(n)) = \tau(v(x)) \mid \mathcal{F}_a \right) \mathbb{P}\left( \mathcal{A}(v(y); G(n)) = \tau(v(y)) \mid \mathcal{F}_a \right)|$$
$$\leq \mathbb{P}((A_x^r)^c \mid \mathcal{F}_a) + \mathbb{P}((B_{x,y}^r)^c \mid \mathcal{F}_a) + \mathbb{P}((\tilde{A}_y^r)^c \mid \mathcal{F}_a) + \mathbb{P}((\tilde{C}_{x,y}^r)^c \mid \mathcal{F}_a).$$

Combining the above equations we have

$$C(x, y) \leq 2\mathbb{P}((A_x^r)^c \mid \mathcal{F}_a) + 2\mathbb{P}((B_{x,y}^r)^c \mid \mathcal{F}_a) + 2\mathbb{P}((\tilde{A}_y^r)^c \mid \mathcal{F}_a) + \mathbb{P}((\tilde{C}_{x,y}^r)^c \mid \mathcal{F}_a)$$
$$+ \mathbb{E}\left[ \left| \mathbb{P}\left( \tilde{C}_{x,y}^r \mid O_{r+1}^n(v(x)), \mathcal{F}_a \right)^{-1} - 1 \right| \wedge 1 \mid \mathcal{F}_a \right]. \tag{34}$$

The upper bound on the right hand side of equation (34) is independent of $\mathcal{A}$, and converges in probability to zero by Theorem 1. Thus we have finished proving (33). Furthermore, similar to the proof of Lemma 8, from (34) we can conclude that as $n \to \infty$,

$$\sup_{x,y\in[n]} \mathbb{E}[C(x, y)] \to 0,$$

thus we finishes the proof of this corollary by Markov's inequality.

$\square$

### H.7   Proof of Lemma 5

For any $\pi \in \mathfrak{S}_k$ and $\delta > 0$, using Corollary 3, we have

$$\frac{1}{n} \sum_{i=1}^{n} \mathbb{1}\{\hat{\tau}(b_i^n + \epsilon U_i) = \pi \circ \tau(i)\} = \frac{1}{n} \sum_{i=1}^{n} \mathbb{P}\left(\hat{\tau}(b_i^n + \epsilon U_i) = \pi \circ \tau(i) \mid \mathcal{F}_a\right) + o_p(1).$$

If we can show the following equation for all $\pi \in \mathfrak{S}_k$, then we finishes the proof of Lemma 5.

$$\limsup_{\delta \to 0^+} \limsup_{n \to \infty} \mathbb{E}\left[\left|\frac{1}{n} \sum_{i=1}^{n} \mathbb{P}\left(\hat{\tau}(b_i^n + \epsilon U_i) = \pi \circ \tau(i) \mid \mathcal{F}_a\right) - \frac{1}{k}\right|\right] = 0.$$

Since

$$\mathbb{E}\left[\left|\frac{1}{n} \sum_{i=1}^{n} \mathbb{P}\left(\hat{\tau}(b_i^n + \epsilon U_i) = \pi \circ \tau(i) \mid \mathcal{F}_a\right) - \frac{1}{k}\right|\right] \leq \frac{1}{n} \sum_{i=1}^{n} \sum_{h=1}^{k} \mathbb{E}\left[\left|\mathbb{P}\left(\tau(i) = h \mid \mathcal{F}_a, b_i^n\right) - \frac{1}{k}\right|\right],$$

then we only need to show for all $1 \leq i \leq n$ and all $h \in [k]$,

$$\limsup_{\delta \to 0^+} \limsup_{n \to \infty} \mathbb{E}\left[\left|\mathbb{P}\left(\tau(i) = h \mid \mathcal{F}_a, b_i^n\right) - \frac{1}{k}\right|\right] = 0. \tag{35}$$

By Theorem 1, for any $\epsilon > 0$, there exists $r_\epsilon \in \mathbb{N}^+$, such that $\limsup_{n \to \infty} \mathbb{P}(\mathcal{G}_v^n$ is not a subgraph of $\mathsf{B}_{r_\epsilon}^n(v)) \leq \epsilon$ for any $v \in [n]$. Then for large enough $n$, after excluding irrelevant quantities then applying conditional Jensen's inequality, we have

$$\mathbb{E}\left[\left|\mathbb{P}\left(\tau(i) = h \mid b_i^n, \mathcal{F}_a\right) - \frac{1}{k}\right|\right]$$
$$\leq \mathbb{P}\left(\mathcal{G}_i^n \text{ is not a subgraph of } \mathsf{B}_{r_\epsilon}^n(i)\right)$$
$$+ \mathbb{E}\left[\left|\mathbb{P}\left(\tau(i) = h \mid \mathcal{F}_a, \mathsf{B}_{r_\epsilon}^n(i)\right) - \frac{1}{k}\right| \mathbb{1}\left\{V(\lceil \delta n \rceil) \cap V_{r_\epsilon}^n(i) = \emptyset\right\}\right]$$
$$+ \mathbb{P}(V(\lceil \delta n \rceil) \cap V_{r_\epsilon}^n(i) \neq \emptyset) \tag{36}$$

On the event $V(\lceil \delta n \rceil) \cap V_{r_\epsilon}^n(v) = \emptyset$, for $h_1, h_2 \in [k]$ with $h_1 \neq h_2$, we have

$$\frac{\mathbb{P}\left(\tau(i) = h_1 \mid \mathcal{F}_a, \mathsf{B}_{r_\epsilon}^n(i)\right)}{\mathbb{P}(\tau(i) = h_2 \mid \mathcal{F}_a, \mathsf{B}_{r_\epsilon}^n(i))} = \frac{\mathbb{P}\left(\tau(i) = h_1 \mid \mathsf{B}_{r_\epsilon}^n(i), \{n_h : h \in [k]\}, \tau(V(\lceil \delta n \rceil))\right)}{\mathbb{P}\left(\tau(i) = h_2 \mid \mathsf{B}_{r_\epsilon}^n(i), \{n_h : h \in [k]\}, \tau(V(\lceil \delta n \rceil))\right)}$$

Therefore, the right hand side of equation (36) is equal to

$$\mathbb{E}\left[\left|\mathbb{P}\left(\tau(i) = h \mid \mathsf{B}_{r_\epsilon}^n(i), \{n_h : h \in [k]\}, \tau(V(\lceil \delta n \rceil))\right) - \frac{1}{k}\right| \mathbb{1}\left\{V(\lceil \delta n \rceil) \cap V_{r_\epsilon}^n(i) = \emptyset\right\}\right]$$
$$+ \mathbb{P}\left(\mathcal{G}_i^n \text{ is not a subgraph of } \mathsf{B}_{r_\epsilon}^n(i)\right) + \mathbb{P}(V(\lceil \delta n \rceil) \cap V_{r_\epsilon}^n(i) \neq \emptyset).$$

By Proposition 2 in [MNS15],

$$\lim_{\delta \to 0^+} \limsup_{n \to \infty} \mathbb{E}\left[\left|\mathbb{P}\left(\tau(i) = h \mid \mathsf{B}_{r_\epsilon}^n(i), \{n_h : h \in [k]\}, \tau(V(\lceil \delta n \rceil))\right) - \frac{1}{k}\right|\right] = 0.$$

Furthermore, as $\delta \to 0^+$, $\limsup_{n \to \infty} \mathbb{P}(V(\lceil \delta n \rceil) \cap V_{r_\epsilon}^n(i) \neq \emptyset) \to 0$, therefore,

$$\limsup_{\delta \to 0^+} \limsup_{n \to \infty} \mathbb{E}\left[\left|\mathbb{P}\left(\tau(i) = h \mid \mathcal{F}_a, b_i^n\right) - \frac{1}{k}\right|\right] \leq \epsilon.$$

Since $\epsilon$ is arbitrary, we conclude that equation (35) holds, this finishes the proof of this lemma.