# OpenReview forum: "Streaming Belief Propagation for Community Detection"
_NeurIPS.cc/2021/Conference — NeurIPS 2021 Poster_

### Official Review · Reviewer_N5WW · 2021-07-02

**Rating:** 8
**Confidence:** 4

**Summary:**

This paper tackles a streaming version of community detection, where the vertices of a graph generated with the classical stochastic block model are revealed sequentially, in a random order.

The authors restrict themselves to what they call $R$-local algorithms, wherein the new information brought by the arrival of a vertex is only propagated at distance at most $R$ of this revealed vertex.

In the absence of side information (i.e. additional information correlated with the ground truth community of each vertex), they show that this class of algorithms is unable to yield meaningful reconstruction when $n$ goes to infinity. In contrast, when side information is present, they provide an algorithm (StreamBP) for community detection, and show that this algorithm matches the performance of the classical belief propagation algorithm (OfflineBP) on the whole graph.

Those results are complemented by experiments on both synthetic and real-world datasets, comparing the StreamBP and OfflineBP algorithms, as well as several alternatives : simple voting algorithms, and a modified version of StreamBP (StreamBP*), modified to better deal with real-world datasets while enjoying the same theoretical guarantees as the original algorithm.

**Limitations And Societal Impact:**

Yes

**Main Review:**

This paper establishes strong results on the defined streaming SBM, which is a natural extension of the SBM to dynamic data. This model is justified by the need to study efficient algorithms that work on evolving networks, which is a fairly recent line of work but is gaining more and more traction.

The authors present a very comprehensive view of the topic : they show a strong impossibility result in the absence of side information, and their StreamBP algorithm matches the performance of classical BP while operating in the $R$-local framework. Since classical BP is conjectured to achieve optimal reconstruction in the SBM, this makes it a strong positive result.

The exposition of the article is quite clear, although it assumes some familiarity with BP algorithms : equations (5) and (6) are tersely justified, and the generalization to non-symmetric SBM may not be "immediate" to non-expert readers. Similarly, the StreamBP* could use a bit more justification, maybe in the form of a small appendix section.

The main technical argument of this paper is Theorem 1, which states that $R$-local algorithms in the sense of streaming data are "local" on another sense: each vertex only gets information from vertices that are at a bounded distance from him. This entails directly the impossiblity result, and allows using the classical Galton-Watson tree coupling arguments for the study of the StreamBP algorithm, which uses classical arguments from thereon.

Both proposed algorithms are quite thoroughly tested, on a large variation of synthetic and real-world data. A very interesting phenomenon, which (as far as I could read) is glossed over in the paper, is the fact that StreamBP is more efficient that OfflineBP, especially for small/medium R; this could be worth ponting out/explaining. Those experiments are complemented by another wealth of graphs in the appendix, where in particular the authors study the variation of performance with respect to the estimates of the parameter $a$: the need for for perfect information of the model parameters being one of the drawbacks of BP algorithms, this is a particularly interesting study.

Finally, for lack of space, the authors relegated the study of what they call "Local streaming algorithms with summary statistics" to the supplementary material. This is an extension of $R$-local algorithms where vertices are allowed to share restricted information outside of their $R$-neighbourhood. They show that Corollary 1 also holds in this case, i.e. no such algorithm can achieve non-trivial reconstruction. This is a nice extension of the local algorithms studied previously, and the gist of the (technical) proofs of this section is explained clearly.

Minor remarks:
- Lines 128 - 129: I found the expression "information from [...] is pulled" quite unclear; maybe consider replacing it by "pulled together" or "merged"
- Above line 269: I didn't find the definition for $\hat{\mathbb E}}$, is it simply an averaging over several runs ?

**Time Spent Reviewing:**

5

---

> ### Author Response · Authors · 2021-08-10
> **Response to Reviewer N5WW**
>
> We thank the reviewer for careful reading and helpful comments, and provide some explanations below.
>
> “This paper establishes strong results on the defined streaming SBM, which is a natural extension of the SBM to dynamic data. This model is justified by the need to study efficient algorithms that work on evolving networks, which is a fairly recent line of work but is gaining more and more traction.”
>
> “The authors present a very comprehensive view of the topic : they show a strong impossibility result in the absence of side information, and their StreamBP algorithm matches the performance of classical BP while operating in the -local framework. Since classical BP is conjectured to achieve optimal reconstruction in the SBM, this makes it a strong positive result.”
>
> Thank you for the comments. We would like to point out that the optimality of BP in sparse SBM settings is actually proved in [1].
>
> “The exposition of the article is quite clear, although it assumes some familiarity with BP algorithms : equations (5) and (6) are tersely justified, and the generalization to non-symmetric SBM may not be "immediate" to non-expert readers. Similarly, the StreamBP* could use a bit more justification, maybe in the form of a small appendix section.”
>
> That’s a good point, we will add more explanations in our future versions, and we thank the reviewer for pointing this out.
>
> “The main technical argument of this paper is Theorem 1, which states that -local algorithms in the sense of streaming data are "local" on another sense: each vertex only gets information from vertices that are at a bounded distance from him. This entails directly the impossiblity result, and allows using the classical Galton-Watson tree coupling arguments for the study of the StreamBP algorithm, which uses classical arguments from thereon.
>
> Both proposed algorithms are quite thoroughly tested, on a large variation of synthetic and real-world data. A very interesting phenomenon, which (as far as I could read) is glossed over in the paper, is the fact that StreamBP is more efficient that OfflineBP, especially for small/medium R; this could be worth ponting out/explaining. Those experiments are complemented by another wealth of graphs in the appendix, where in particular the authors study the variation of performance with respect to the estimates of the parameter : the need for for perfect information of the model parameters being one of the drawbacks of BP algorithms, this is a particularly interesting study.”
>
> The phenomenon that StreamBP outperforms OfflineBP in simulation can be explained using the proof of Theorem 2: according to the proof, $R$-local StreamBP is gathering information from a neighborhood that contains the radius R-neighborhood of a node to classify it, thus outperforms radius $R$ Offline BP which uses exactly the information from its radius $R$ neighborhood. Such advantage vanishes as $R$ goes to infinity, especially under the setting of Corollary 2.
>
> “Finally, for lack of space, the authors relegated the study of what they call "Local streaming algorithms with summary statistics" to the supplementary material. This is an extension of -local algorithms where vertices are allowed to share restricted information outside of their -neighbourhood. They show that Corollary 1 also holds in this case, i.e. no such algorithm can achieve non-trivial reconstruction. This is a nice extension of the local algorithms studied previously, and the gist of the (technical) proofs of this section is explained clearly.”
>
> Thank you for the careful reading into the appendix. Currently we only analyze local algorithms with summary statistics being the average of statistics attached to each node. Hopefully we can extend our analysis to other types of summary statistics as well.
>
> "Minor remarks:
>
> Lines 128 - 129: I found the expression "information from [...] is pulled" quite unclear; maybe consider replacing it by "pulled together" or "merged""
>
> That is a good point, we can make this change in our future version.
>
> "Above line 269: I didn't find the definition for \hat{\mathbb E}}, is it simply an averaging over several runs ?"
>
> Yes, that is the correct understanding. We will clarify this in our future version.
>
> Please let us know if you have any other feedback and we will do our best to address those issues!
>
>
>
>
>
>
>
>
> [1] Mossel, E., & Xu, J. (2016, January). Local algorithms for block models with side information. In   Proceedings of the 2016 ACM Conference on Innovations in Theoretical Computer Science (pp. 71-80).

---

### Official Review · Reviewer_WcZo · 2021-07-08

**Rating:** 6
**Confidence:** 4

**Summary:**

The authors consider a streaming stochastic block model and propose a streaming variant of belief propagation for this dynamical scenario. They show that the proposed version asymptotically performs as good as offline belief propagation.

**Limitations And Societal Impact:**

Partly.

Yes, the proposed algorithm  performs asymptotically the same as offline BP. Yet it does this in fewer iterations, which might make it more applicable to certain scenarios for which offline BP does not scale well enough. Clustering in particular might have detrimental societal effects when applied to social networks (polarization of society, targeted advertising,...)

**Main Review:**

This paper tackles the problem of community detection in dynamical networks. The problem is studied in the context of the streaming stochastic block model with the assumptions of (i) bounded degree graphs and (ii) available side information.

They propose a streaming version of belief propagation and a more robust version that reduces the effect of longer cycles by considering only "bounded" information.

Theoretical guarantees are derived that show "optimality" of streamingBP in that it asymptotically achieves the same accuracy as the offline version. The algorithms are also evaluated on a range of synthetic and (limited) real-world scenarios.

I have reviewed an earlier version of this paper and I must admit that the authors did a good job in overhauling the presentation of the work. In the current form, the paper is much clearer and more accessible. Nonetheless, I am certainly surprised that the authors did not address certain aspects that were central points of criticism in the last round of reviews.  In case of acceptance, I would highly encourage the authors to take this points into account. These are:
* First, the discussion of the experimental results
  * The results in Section 5.3 (and in Appendix A/B) show certain aspects regarding the behavior of the algorithms that are not discussed sufficiently. In particular, we can see that StreamBP often does better than its offline version (for a fixed radius $R$). I found this surprising and the paper does not discuss this consistently. E.g., in the Intro it is stated that StreamBP does not outperform offline BP whereas Theorem 2 states that StreamBP performs at least as good as offline BP. I assume that this discrepancy stems from the difference of comparing the algorithm for a fixed radius vs. unbounded radius. If so, this should definitely be clarified!

  * There is some erratic behavior of StreamBp e.g., in Fig. 4a and 4c or in Fig. 7. The brief discussion in Sec. 5.1 does not explain the precise reasons for this behavior or which of the underlying assumptions is violated. Clearly, the optimality of Theorem 2 does not apply in this setting.

  * Also why is it that StreamBP* usually performs worse than StreamBP (and strictly worse than OracleBP) but does a very good job in precisely those scenarios where StreamBP fails?

I would suggest that the authors summarize the main insights of the experiments (also for the additional experiments in the Appdx.) in a couple of sentences (see next point)

* Second, the conclusion (or rather the lack of it)
The lack of conclusion is definitely not optimal as the paper abruptly ends after the experiments. It would definitely help to discuss how the experiments connect to the theory and how they substantiate the claims.


* Finally, missing references
The studied problem is closely related to community detection in temporal networks. Besides a brief mention of a single paper, the authors fail to reference this line of work and to distinguish their work from the recent progress in temporal networks. I suggest the following references again:
  *  Aslak, Ulf, Martin Rosvall, and Sune Lehmann. "Constrained information flows in temporal networks reveal intermittent communities." Physical Review E 97.6 (2018): 062312.
  * Zhu, Peican, et al. "Community detection in temporal networks via a spreading process." EPL (Europhysics Letters) 126.4 (2019): 48001.
  * He, Jialin, and Duanbing Chen. "A fast algorithm for community detection in temporal network." Physica A: Statistical Mechanics and its Applications 429 (2015): 87-94.
  * Gao, Xubo, et al. "Temporal network pattern identification by community modelling." Scientific reports 10.1 (2020): 1-12.

### Minor Remarks
* Typo in line 45: The ' after the stop should be removed
* The sentence before (2) lacks some object. We define... Let us define...
* The normalization proposed in Sec. 5.1 is pretty standard. Why is it not used for StreamBP?


**Time Spent Reviewing:**

6

---

> ### Author Response · Authors · 2021-08-10
> **Response to Reviewer WcZo**
>
> We thank the reviewer for careful reading and helpful comments, and provide some explanations below.
>
> “This paper tackles the problem of community detection in dynamical networks. The problem is studied in the context of the streaming stochastic block model with the assumptions of (i) bounded degree graphs and (ii) available side information.”
>
> Actually side information is not required in all parts of our study. For instance, Theorem 1 is proved under the setting where no side information is available.
>
> “They propose a streaming version of belief propagation and a more robust version that reduces the effect of longer cycles by considering only "bounded" information. Theoretical guarantees are derived that show ‘optimality’ of streamingBP in that it asymptotically achieves the same accuracy as the offline version. The algorithms are also evaluated on a range of synthetic and (limited) real-world scenarios.”
>
> We would like to point out that under the SBM, belief propagation (BP) is conjectured to be optimal in the offline setting among all polynomial-time algorithms. There is very strong rigorous and empirical evidence for this (see [1]). Furthermore, in Corollary 2, we showed that under our setting streaming BP with large radius achieves optimal estimation accuracy among all algorithms.
>
> “I have reviewed an earlier version of this paper and I must admit that the authors did a good job in overhauling the presentation of the work. In the current form, the paper is much clearer and more accessible.”
>
> We appreciate your effort in helping us improve the paper quality!
>
> “The results in Section 5.3 (and in Appendix A/B) show certain aspects regarding the behavior of the algorithms that are not discussed sufficiently. In particular, we can see that StreamBP often does better than its offline version (for a fixed radius ). I found this surprising and the paper does not discuss this consistently. E.g., in the Intro it is stated that StreamBP does not outperform offline BP whereas Theorem 2 states that StreamBP performs at least as good as offline BP. I assume that this discrepancy stems from the difference of comparing the algorithm for a fixed radius vs. unbounded radius. If so, this should definitely be clarified!”
>
> We apologize for not making this clear enough. Your understanding is correct. In simulation StreamBP outperforms offline BP because we fix the radius. If we let n goes to infinity then the radius goes to infinity, then according to Theorem 2 and Corollary 2, offline BP and StreamBP achieve asymptotically equivalent estimation accuracy (which is also the asymptotically optimal estimation accuracy achieved by any algorithm under our setting).
>
> “There is some erratic behavior of StreamBp e.g., in Fig. 4a and 4c or in Fig. 7. The brief discussion in Sec. 5.1 does not explain the precise reasons for this behavior or which of the underlying assumptions is violated. Clearly, the optimality of Theorem 2 does not apply in this setting.”
>
> This is due to $n$ not being large enough.
>
> The erratic behavior is caused by cycles in the fixed radius neighborhood of a node. As $n \rightarrow \infty$, theoretically speaking the neighborhood will not contain cycles.
>
>  However, due to computational limitations, currently we are not running simulations on larger graphs, thus there will be cycles in the neighborhood.
>
> As shown by the proof of Theorem 2, streamBP will be gathering information from a larger neighborhood compared to its offline counterpart with the same radius. Therefore, asymptotically speaking, for every fixed radius, as $n \rightarrow \infty$, streamBP should outperform offline BP.
>
> However, In the finite sample setting, this means that the performance of streamBP is more likely to be degraded by the appearance of cycles (since a larger neighborhood contains more cycles), which in theory will not be a problem as $n \rightarrow \infty$.
>
> “Also why is it that StreamBP* usually performs worse than StreamBP (and strictly worse than OracleBP) but does a very good job in precisely those scenarios where StreamBP fails?”
>
> StreamBP*can be understood as putting an upper bound on the size of the neighborhood from which the algorithm gathers information.
>
> Therefore, in cases the algorithm does not encounter cycles at all (this is the case for large $n$), StreamBP* will perform worse because it is using information from a smaller neighborhood compared to StreamBP.
>
> However, in cases where the algorithm does encounter cycles (recall that cycles will degrade the performance of any belief propagation), larger size neighborhood implies more cycles, therefore StreamBP* encounters less cycles while StreamBP encounters more, which explains why StreamBP* does a good job while StreamBP fails.
>
> Thank you for pointing these out, we will add a few sentences summarizing the insights of experiments in our future version.
>
> “Second, the conclusion (or rather the lack of it) The lack of conclusion is definitely not optimal as the paper abruptly ends after the experiments. It would definitely help to discuss how the experiments connect to the theory and how they substantiate the claims.”
>
> We choose a paper structure which summarizes our contributions at the end of introduction instead of adding a conclusion section. But if space permits, we will add a conclusion section after simulation to summarize insights since now it seems necessary to do so.
>
> “Finally, missing references The studied problem is closely related to community detection in temporal networks. Besides a brief mention of a single paper, the authors fail to reference this line of work and to distinguish their work from the recent progress in temporal networks. I suggest the following references again:
>
> Aslak, Ulf, Martin Rosvall, and Sune Lehmann. "Constrained information flows in temporal networks reveal intermittent communities." Physical Review E 97.6 (2018): 062312.
>
> Zhu, Peican, et al. "Community detection in temporal networks via a spreading process." EPL (Europhysics Letters) 126.4 (2019): 48001.
>
> He, Jialin, and Duanbing Chen. "A fast algorithm for community detection in temporal network." Physica A: Statistical Mechanics and its Applications 429 (2015): 87-94.
>
> Gao, Xubo, et al. "Temporal network pattern identification by community modelling." Scientific reports 10.1 (2020): 1-12.”
>
> Thank you for the references again! We didn’t include these papers in the current version because:
>
> (1) We realize that these papers are more focused on analyzing changing community structure, while in our setting we consider the expansion of datasets over time with static community structure.
>
> (2) We regard our work as a theoretical paper characterizing the optimal performance of clustering under a simple model. We also propose an algorithm achieving the optimum in our setting. Therefore, our literature review is more focused on mathematical literature which supports our theory.
>
> However, we agree that adding a few sentences pointing to this line of research would be a good idea, and we thank the reviewer for sharing these papers.
>
> “Typo in line 45: The ' after the stop should be removed”
>
> That’s the second quotation mark paired with the other one in the line above, and we should add it before the stop, thanks for pointing this out.
>
> "The sentence before (2) lacks some object. We define... Let us define…"
>
> Thanks, we will change the sentence accordingly.
>
> "The normalization proposed in Sec. 5.1 is pretty standard. Why is it not used for StreamBP?"
>
> If the reviewer means constraining the messages, then we need it here because in the finite sample case where $n$ is not large enough, this technique can be used to overcome some erratic behaviors.  This step is not needed in the ideal case where $n$ goes to infinity, in which case StreamBP is provably optimal (see Corollary 2).
>
> Please let us know if you have any other feedback and we will do our best to address those issues!
>
>
> [1] Decelle, A., Krzakala, F., Moore, C., & Zdeborová, L. (2011). Asymptotic analysis of the stochastic block model for modular networks and its algorithmic applications. Physical Review E, 84(6), 066106.

---

### Official Review · Reviewer_QSCc · 2021-07-16

**Rating:** 5
**Confidence:** 3

**Summary:**

A streaming Stochastic Block Model (SBM) is proposed as null model to analyze streaming community detection algorithms on growing graphs.
A online algorithm based on Belief Propagation (BP) is proposed to estimate community membership in this setting. It is proven to achieve the same accuracy as its offline variant BP in a simplified setting.


**Limitations And Societal Impact:**

A discussion at the end of the paper is completely missing. At least the limitations introduced by focusing only on simple SBMs with two communities and the missing parameter estimation procedure should be addressed.


**Main Review:**

Community detection based on relational data is a common problem and streaming algorithms that scale to large networks could be highly valuable.
The provided theoretical insights and proposed algorithms could make a valuable contribution to the field. Yet, the limitations are not clearly discussed and important related information is scattered between different sections, which makes it difficult to assess the validity of some of the claims.
My biggest concern is that theoretical insights and algorithms are limited to a simplistic Stochastic Block Model that consists of only two communities, for which the connection probabilities are assumed to be known. This is quite detached from possible application scenarios.
Detailed comments follow below.

Strengths:
+ Proposal of a null model for streaming community detection algorithms, which is based on the Stochastic Block Model (SBM)
+ Proposal of streaming versions of BP, STREAMBP and STREAMBP*, that achieve optimal estimation accuracy (in relation to global BP) in case of the simple two community SBM.
+ Theoretical insights:
  * proof that, in the absence of side information, no local streaming algorithm can achieve non-trivial community membership reconstruction
  * STREAMBP achieves the same accuracy as offline BP in case of available side information (assuming a simple SBM with only two communities)
+ Conjectures of optimality for STREAMBP follow naturally from the theoretical insights.
+ Experiments on synthetic and real world data provide evidence for the utility of the proposed algorithms.

Weaknesses:
- The network growth model (streaming SBM) does not seem to be appropriate in most real world applications:
* Only one node arrives randomly at each time point.
* This node will be disconnected from the current graph G_{t-1} with high probability because it will likely be connected to other nodes in the large full graph, which have not been arrived yet. This is in stark contrast to the common intuition of network growth models that create connected components (e.g. via preferential attachment or other attachment mechanisms).
* The motivation to model temporal networks does not seem to fit the growth model, as the parameters of the SBM are all static. Nodes just arrive in a random order. The time order does not even depend on a process.
* The SBM is neither degree corrected or allows for multiple community memberships. This makes it an unrealistic description of most real world graphs.
- The streaming SBM might still be interesting to study online algorithms. Yet, I would expect that also streaming algorithms perform better if they explore not uniformly at random arriving nodes but nodes that are connected to the currently available graph.
- The availability of side information in the introduced form seems to be a lot to ask for in applications.
- Is the definition of V^t_v correct on page 4? It is unclear to me why the neighborhoods should not be symmetric for symmetric graphs G(n). V^t_v seem to form connected components that contain v(t).
- Theoretical results apply only to the simple SBM with two groups with intra-community connection probability a and inter-community detection probability b.
- Also the algorithm STREAMBP is only introduced for this simplistic case.
- The definition of the BP update (Eq. (5)) seems to assume the knowledge of the probabilities a and b? How can this be? Don't the need to be estimated as well?! Shouldn't proofs also cover possible errors in their estimation?!
This problem seems to be acknowledged in the experiments section but it is not clear how the estimates are obtained without the knowledge of the communities Vi to start with.
- No code has been shared.
- A discussion is missing.

Points of minor critique:
- The notation delta over = should be introduced.
- It is surprising that the presence of cycles that hampers STREAMBP (see Sec. 5.1) does not create difficulties in the proof. What is the intuition behind that?
- A detailed explanation how multiple communities (k>2) are handled in the experiments is missing.


**Time Spent Reviewing:**

3

---

> ### Author Response · Authors · 2021-08-10
> **Response to Reviewer QSCc**
>
> We thank the reviewer for careful reading and helpful comments, and provide some explanations below.
>
> “Community detection ...... My biggest concern is that theoretical insights and algorithms are limited to a simplistic Stochastic Block Model that consists of only two communities, for which the connection probabilities are assumed to be known. This is quite detached from possible application scenarios. Detailed comments follow below.”
>
> Thanks for the comments. We would like to point out that:
>
>  (a) A precise characterization of the information-theoretic limit for clustering in the sparse graph regime has only appeared recently, and almost uniquely for the stochastic block model (see lines 28-33). Hence, if we want to prove optimality of our approach, this is a reasonable model to focus on. In this paper we study a dynamic variant of SBM.
>
> Furthermore, we are not aware of any other model under which optimality has been proved or can be proved given the current status of mathematical research.
> For instance, [1] and [2] are exemplar papers of this type, and the algorithms they proposed to achieve the optimal recovery also assume known connection probabilities.
> In cases where ground-truth labels are available for a subset of nodes, one can estimate the connection probabilities using the same reasoning as we did in Section 5.3. In cases where these labels are not available, we can use the algorithm in [3], Theorem 3 to estimate connection probabilities. Furthermore, in Appendix A we showed that the performance of our algorithms is robust to the choice of a and b.
>
> (b) Extending our proofs to cases where the number of communities is larger than two is straightforward. We used the case of two communities to make proofs more readable.
>
> “Strengths:
> …...
> Conjectures of optimality for STREAMBP follow naturally from the theoretical insights.
> …...”
>
> The optimality for StreamBP with a large radius is not a conjecture, and we would like to highlight that we have rigorously proved it (see Corollary 2).
>
> “The network growth model (streaming SBM) does not seem to be appropriate in most real world applications: Only one node arrives randomly at each time point. ”
>
> We would like to point out that since in our setting the graph is sparse, the probability that any x nodes interacting with each other with $x = O(1)$ will be $o(1)$. Therefore, it is very likely that our results can be extended to cases where $O(1)$ nodes arrive at each time point, and that will be one of our future research directions.
>
> “This node will be disconnected from the current graph $G_{t-1}$ with high probability because it will likely be connected to other nodes in the large full graph, which have not been arrived yet. This is in stark contrast to the common intuition of network growth models that create connected components (e.g. via preferential attachment or other attachment mechanisms).”
>
> Since we are working under a sparse network (which is the case for many network real datasets), when $t$ is small, it is indeed very likely that $v(t + 1)$ will be disconnected to $G(t)$. However, when both $n,t$ are large, the number of connections between $v(t + 1)$ and $G(t)$ will be approximately Poisson($Ct / n$) with C being a fixed constant depending only on connection probabilities. This means that for arbitrarily small $\epsilon > 0$, after $n\epsilon$ nodes are revealed, any new node will not be isolated with positive probability.
>
> We also point out that a vertex with degree s will not be isolated with probability $s / (s + 1)$ since it’s only isolated if it appears earlier than all its neighbors. So when the average degree is large, we would expect such isolation happens only for a small proportion of vertices.
>
> “The motivation to model temporal networks does not seem to fit the growth model, as the parameters of the SBM are all static. Nodes just arrive in a random order. The time order does not even depend on a process. The SBM is neither degree corrected or allows for multiple community memberships. This makes it an unrealistic description of most real world graphs.”
>
> See argument (a). Unfortunately at the current stage strong assumptions are required for understanding the information-theoretic limit of efficient clustering with local algorithms on dynamic graphs. This paper is also a first step towards analyzing these more complicated models.
>
> “The streaming SBM might still be interesting to study online algorithms. Yet, I would expect that also streaming algorithms perform better if they explore not uniformly at random arriving nodes but nodes that are connected to the currently available graph.”
>
> According to Corollary 2, under mild regularity conditions, streaming BP with large enough radius achieves optimal estimation accuracy among all algorithms. This result can be extended to cases where the number of communities is greater or equal to three.
>
> However, we agree that exploring connected nodes more frequently might be helpful for improving finite sample performance, and we thank the reviewer for pointing this out.
>
> “The availability of side information in the introduced form seems to be a lot to ask for in applications.”
>
> We would like to point out that (i) Side information is assumed to be nonzero only in Theorem 2, and Theorem 1 is proved without side information. (ii) In many applications an oracle of this type is provided by a classification algorithm that uses data available at each node (e.g. posts in a social network). (iii) The problem with side information is standard in the mathematical literature and there is a tight relationship between the two settings [1,4].
>
> “Is the definition of V^t_v correct on page 4? It is unclear to me why the neighborhoods should not be symmetric for symmetric graphs G(n). V^t_v seem to form connected components that contain v(t).”
>
> The definition is correct. $V^t_v$ is not the connected component that contains $v(t)$. You can think of
>
> $V^t_v =$ {$u: \mbox{ if we want to send something from u to v using only local streaming algorithms, then at time t it is possible to achieve this based on past dynamics}$},
>
> which is not symmetric.
>
> For example, consider a simple graph $a-b-c-d-e$ (the only four edges in this graph being $(a,b), (b,c), (c,d), (d,e)$), with nodes revealed in order $a,b,c,d,e$, and $R = 1$. Then according to our definition, $a \in V_e^5$, while $e \notin V_a^5$.
>
> “Theoretical results apply only to the simple SBM with two groups with intra-community connection probability a and inter-community detection probability b.”
>
> See arguments (a) and (b) above.
>
> “Also the algorithm STREAMBP is only introduced for this simplistic case.”
>
> StreamBP is introduced for cases including $k \geq 3$, while only proved for the simplistic case $k = 2$. See also argument (b) above, extension is actually straightforward.
>
> We would also like to point out that the case with intra-community connection probability a and inter-community connection probability b is the most difficult one, as in this case, nodes in different communities have “similar” behavior. Extension of StreamBP to other cases is straightforward using the standard rule of belief propagation, but the theory needs to be treated differently (actually it will be much simpler). Due to space limits we only introduce the current version here. We may add a few remarks and include it in the appendix in our future versions.
>
> “The definition of the BP update (Eq. (5)) seems to assume the knowledge of the probabilities a and b? How can this be? Don't the need to be estimated as well?! Shouldn't proofs also cover possible errors in their estimation?! This problem seems to be acknowledged in the experiments section but it is not clear how the estimates are obtained without the knowledge of the communities Vi to start with.”
>
> In Appendix A we showed that the performance of the algorithms is robust to the choice of a and b. Also see argument (a) above.
>
> “No code has been shared.”
>
> We apologize for that, code will be released soon.
>
> “A discussion is missing.”
>
> We chose a paper structure which summarizes our contributions at the end of the introduction section instead of adding a conclusion section. See for instance [1,2,3,4] for papers written in this format. But we will add a conclusion section after simulation if it is regarded as necessary.
>
> “The notation delta over = should be introduced.”
>
> Thank you for pointing this out, we will add it in our future versions.
>
> “It is surprising that the presence of cycles that hampers STREAMBP (see Sec. 5.1) does not create difficulties in the proof. What is the intuition behind that?”
>
> Since we are working on sparse graphs, the probability that cycles appear in any local neighborhood will converge to zero as the number of vertices goes to infinity. See Lemma 1 in appendix D for details.
>
> “A detailed explanation how multiple communities (k>2) are handled in the experiments is missing.”
>
> Neither Algorithm 1 nor Algorithm 2 is actually restricted to k = 2, so application should be straightforward.
>
> Please let us know if you have any other feedback and we will do our best to address those issues!
>
> [1] Mossel, E., & Xu, J. (2016, January). Local algorithms for block models with side information. In   Proceedings of the 2016 ACM Conference on Innovations in Theoretical Computer Science (pp. 71-80).
>
> [2] Mossel, E., Neeman, J., & Sly, A. (2014, May). Belief propagation, robust reconstruction and optimal recovery of block models. In Conference on Learning Theory (pp. 356-370). PMLR.
>
> [3] Mossel, E., Neeman, J., & Sly, A. (2015). Reconstruction and estimation in the planted partition model. Probability Theory and Related Fields, 162(3), 431-461.
>
> [4] Kanade, V., Mossel, E., & Schramm, T. (2016). Global and local information in clustering labeled block models. IEEE Transactions on Information Theory, 62(10), 5906-5917.

---

> > ### Comment · Reviewer_QSCc · 2021-08-27
> > **Post-rebuttal evaluation**
> >
> > I thank the authors for their detailed response and addressing some of my concerns. I will still keep my score for the following main reason: The used estimates of a and b are not available online, which makes the whole online argument and algorithmic improvement relatively pointless.
> >
> > Two remaining points of minor critique are:
> > 1) The streaming SBM model does not seem to be a realistic random graph model but useful for the analysis of some streaming algorithms. Of course, this has only relevance for finite sample performance. For that reason, it is also not that surprising that a BP version is optimal for infinite samples, since the streaming aspect does not influence the results crucially.
> > 2) The BP updates have only been defined for the situation with 2 groups and Algorithms 1 and 2 reference this BP update. A more general definition could add merit to the paper.

---

> > > ### Author Response · Authors · 2021-08-29
> > > **Response to reviewer QSCc**
> > >
> > > We thank the reviewer for the comments.
> > >
> > > At time t, the graph is distributed according to a SBM with edge probabilities a/n, b/n. Given that noisy side information is available on the vertices, there are easy approaches that can be implemented online. For instance, match the following three empirical
> > > quantities to the corresponding expectations: (1) Average degree; (2) Number of edges with endpoints having the same noisy label; (3) Number of length 2 paths with all the noisy labels equal. These are local quantities (only depend on the radius 2 neighborhood of each vertex), and can be accurately estimated online. Indeed it is sufficient to look at steps between t-k and k with k = o(t), to get estimates with accuracy O(1/\sqrt{t}). This is just one approach that is easy to describe. Estimating a and b is not a challenge.
> > >
> > > We would like to point out that Algorithm 1 and 2 are both defined for multiple group numbers. We didn't put any constraint on $k$ for these two algorithms, which refers to the number of groups.

---

> > > > ### Comment · Reviewer_QSCc · 2021-09-02
> > > > **Estimation of a and be needs to be included in the analysis**
> > > >
> > > > I thank the authors for their quick response. However, the estimation of the edge probabilities a/n, b/n has to be coupled with the estimation of group memberships (as it not only determined by the side information). Initially, the estimate can be quite off
> > > > and it needs to be shown that the message passing updates recover from potentially wrong early assessments.
> > > > Regardless of whether it is easy to do, I believe this needs to be included in the proof and the analysis. Otherwise, the theory for the proposed algorithm is incomplete. For that reason, I believe that the paper is not ready for acceptance yet.

---

### Official Review · Reviewer_Y4Hw · 2021-07-19

**Rating:** 5
**Confidence:** 4

**Summary:**

The paper proposes a Belief-Propagation algorithm to effectively solve the problem of online community detection, i.e., in cases where the complete information about the network
is not available to a system. Instead, more and more information becomes available to the system as time progresses.


**Main Review:**

The paper proposes a Belief-Propagation algorithm to effectively solve the problem of online community detection, i.e., in cases where the complete information about the network
is not available to a system. Instead, more and more information becomes available to the system as time progresses.

Sections 2-4 are quite well written. In Section 2, it is understood that the vertex and edge sets at time t, namely V(t) and E(t) are subsets of those at time t+1,
i.e. V(t) \subset V(t+1) and so on.

However, it's not quite clear from the experiment description how was the data being supplied to the Streaming BP model proposed.
Since experiments are conducted on complete graphs, I believe the information about the graph was partially fed in.
More details about the settings need to be included about how the initial graph G(1) was chosen. In practice, G(t) growing radially it may actually happen
that a separate connected component of G becomes available, i.e., there is minimal or no connection between V(t+1)-V(t) and V(t), or in other words |V(t+1)-V(t) \cap V(t)| could be small.
How would the streaming BP algorithm do if the partial information being available is of that nature?

Moreover, it's a bit strange to see why weren't more recent approaches for community detection being applied as baselines, e.g. the graph convolutional approaches like SAGE (“Inductive representation learning on large graphs,” in NIPS, 2017), or say even shallow dense vector based approach like retraining node2vec/deepwalk on each G(t) as and when they become available and classifying the test set nodes threafter.

I think the paper needs to compare their streaming-BP results with some other incremental retraining approaches like employing graph convolutional networks etc.


**Time Spent Reviewing:**

3 hours

---

> ### Author Response · Authors · 2021-08-10
> **Response to Reviewer Y4Hw**
>
> We thank the reviewer for careful reading and helpful comments, and provide some explanations below.
>
> “Sections 2-4 are quite well written. In Section 2, it is understood that the vertex and edge sets at time $t$, namely $V(t)$ and $E(t)$ are subsets of those at time $t+1$, i.e. $V(t) \subset V(t+1)$ and so on.
>
> However, it's not quite clear from the experiment description how was the data being supplied to the Streaming BP model proposed. Since experiments are conducted on complete graphs, I believe the information about the graph was partially fed in. More details about the settings need to be included about how the initial graph $G(1)$ was chosen.”
>
> We apologize for not making this clear enough. On both synthetic and real world datasets, we generate the graph sequence by choosing a uniformly random permutation over all vertices to simulate the dynamic process. To be specific, on the real world dataset, we randomly pick an order for the vertices, and assume we receive the nodes following that order.
>
> This is in accordance with our modeling assumption in lines 42-43 and 95-96. We can add this clarification in our future versions, and appreciate the reviewer for pointing this out.
>
> “ In practice, $G(t)$ growing radially it may actually happen that a separate connected component of $G$ becomes available, i.e., there is minimal or no connection between $V(t+1)-V(t)$ and $V(t)$, or in other words $|V(t+1)-V(t) \cap V(t)|$ could be small. How would the streaming BP algorithm do if the partial information being available is of that nature?””
>
> If the number of edges between $v(t+1)$ and $V(t)$ is small, or even zero, we can still run streaming BP. In that case, at the point $v(t + 1)$ arrives, it will receive little (or no) information other than its own side information.
>
> However, as time progresses, very likely there will be other vertices being connected to $v(t + 1)$. When such connections occur, information from other vertices will be propagated to vertex $v(t + 1)$ according to our algorithm, thus helping clustering $v(t + 1)$.
>
> We would like to point out that the situation the reviewer described will happen for some vertices, but only a small proportion of them. Since we are working under a sparse network, when $t$ is small, it is indeed very likely that $v(t + 1)$ will be disconnected to $G(t)$. However, when both $n,t$ are large, the number of connections between $v(t + 1)$ and $G(t)$ will be approximately Poisson(Ct / n) with C being a fixed constant depending only on connection probabilities. This means that for arbitrarily small $\epsilon > 0$, after $n\epsilon$ nodes are revealed, any new node will not be isolated with positive probability. We also point out that a vertex with degree s will not be isolated with probability s / (s + 1) since it’s only isolated if it appears earlier than all its neighbors. So when the average degree is large, we would expect such isolation happens only for a small proportion of vertices.
>
>
> Furthermore, we have justified the performance of streaming BP in our paper, both theoretically and experimentally. Therefore the isolating situation the reviewer described will not harm the performance of the algorithm.
>
> “Moreover, it's a bit strange to see why weren't more recent approaches for community detection being applied as baselines, e.g. the graph convolutional approaches like SAGE (“Inductive representation learning on large graphs,” in NIPS, 2017), or say even shallow dense vector based approach like retraining node2vec/deepwalk on each G(t) as and when they become available and classifying the test set nodes threafter.
>
> I think the paper needs to compare their streaming-BP results with some other incremental retraining approaches like employing graph convolutional networks etc.’’
>
> Under the SBM, offline belief propagation (BP) is conjectured to be optimal among all polynomial-time algorithms. There is very strong rigorous and empirical evidence for this (see [1]). As a consequence, asymptotically speaking, none of SAGE, node2vec, and deepwalk will outperform offline BP with a large enough radius. A second consequence is that in an asymptotic sense, the accuracy of offline BP with large radius is an upper bound on all polynomial time online algorithms (see [2]). Hence, we are comparing our approach to the strongest possible benchmark.
>
> We add that the proposed approaches (SAGE, node2vec, deepwalk) seem to require optimizing a loss function over the whole graph (or training a neural network using samples from all nodes) at some stage. This is not a local operation in the classification we adopt, and hence the comparison is not fair.
>
> However, we agree that such a numerical question (comparing SAGE, node2vec, and deepwalk  to optimal approaches for SBM) could be an interesting research direction.
>
> Please let us know if you have any other feedback and we will do our best to address those issues!
>
>
>
>
> [1] Decelle, A., Krzakala, F., Moore, C., & Zdeborová, L. (2011). Asymptotic analysis of the stochastic block model for modular networks and its algorithmic applications. Physical Review E, 84(6), 066106.
>
> [2] Mossel, E., & Xu, J. (2016, January). Local algorithms for block models with side information. In   Proceedings of the 2016 ACM Conference on Innovations in Theoretical Computer Science (pp. 71-80).

---

### Decision · Program_Chairs · 2021-09-28

**Decision:**

Accept (Poster)

**Comment:**

The paper considers streaming/dynamic version of the stochastic block model, and provides theoretical guarantee. However, as is evident from multiple reviews, the paper writing is suboptimal. The dynamic version of SBM can be defined in multiple ways. Therefore, the authors need to do a better job in motivating the usefulness of their model, discuss the parameter estimation, and provide a more thorough experimental analysis.

**Consistency Experiment:**

NeurIPS has a long history of experimentation. In 2014, NeurIPS ran an experiment in which 10% of submissions were reviewed by two independent committees to quantify the randomness in the review process. This year, we repeated a variant of this experiment to see how the quality of the review process has changed over time.  This paper was part of the experiment and was therefore assigned to two committees (consisting of reviewers, an Area Chair, and a Senior Area Chair) that reached independent decisions.  If both committees made the same recommendation, this recommendation was followed. If a single committee recommended acceptance, the paper was accepted (with the exception of a few cases in which the other committee identified what we considered a fatal flaw, e.g., an error in a key result).

This copy’s committee reached the following decision: **Reject**

The other committee assigned to the paper recommended **Accept (Poster)**.  You can find the other set of reviews, along with any follow up discussion with the authors here:
https://openreview.net/forum?id=3-F0-Zpcrno